# Policy Optimization for Markov Games: Unified Framework and Faster Convergence

**Runyu Zhang**[*]
Harvard University
runyuzhang@fas.harvard.edu

**Qinghua Liu**[*]
Princeton University
qinghual@princeton.edu

**Huan Wang**
Salesforce Research
huan.wang@salesforce.com

**Caiming Xiong**
Salesforce Research
cxiong@salesforce.com

**Na Li**
Harvard University
nali@seas.harvard.edu

**Yu Bai**
Salesforce Research
yu.bai@salesforce.com

## Abstract

This paper studies policy optimization algorithms for multi-agent reinforcement learning. We begin by proposing an algorithm framework for two-player zero-sum Markov Games in the full-information setting, where each iteration consists of a policy update step at each state using a certain matrix game algorithm, and a value update step with a certain learning rate. This framework unifies many existing and new policy optimization algorithms. We show that the *state-wise average policy* of this algorithm converges to an approximate Nash equilibrium (NE) of the game, as long as the matrix game algorithms achieve low weighted regret at each state, with respect to weights determined by the speed of the value updates. Next, we show that this framework instantiated with the Optimistic Follow-The-Regularized-Leader (OFTRL) algorithm at each state (and smooth value updates) can find an $\widetilde{\mathcal{O}}(T^{-5/6})$ approximate NE in $T$ iterations, and a similar algorithm with slightly modified value update rule achieves a faster $\widetilde{\mathcal{O}}(T^{-1})$ convergence rate. These improve over the current best $\widetilde{\mathcal{O}}(T^{-1/2})$ rate of symmetric policy optimization type algorithms. We also extend this algorithm to multi-player general-sum Markov Games and show an $\widetilde{\mathcal{O}}(T^{-3/4})$ convergence rate to Coarse Correlated Equilibria (CCE). Finally, we provide a numerical example to verify our theory and investigate the importance of smooth value updates, and find that using "eager" value updates instead (equivalent to the independent natural policy gradient algorithm) may significantly slow down the convergence, even on a simple game with $H = 2$ layers.

## 1 Introduction

Policy optimization, i.e. algorithms that learn to make sequential decisions by local search on the agent's policy directly, is a widely used class of algorithms in reinforcement learning [40, 44, 45]. Policy optimization algorithms are particularly advantageous in the multi-agent reinforcement learning (MARL) setting (e.g. compared with value-based counterparts), due to their typically lower representational cost and better scalability in both training and execution. A variety of policy optimization algorithms such as Independent PPO [14], MAPPO [56], QMix [42] have been proposed to solve real-world MARL problems [4, 39, 43]. These algorithms share a same high-level structure with iterative *value updates* (for certain value estimates) and *policy updates* (often independently with each agent) using information from the value estimates and/or true rewards.

---

[*]The two authors contributed equally to this work.

36th Conference on Neural Information Processing Systems (NeurIPS 2022).

While policy optimization for MARL has been studied theoretically in a growing body of work, there are still gaps between algorithms used in practice and provably-efficient algorithms studied in theory—Algorithms in practice generally follow two natural design principles: *symmetric updates* among all agents, and *simultaneous learning* of values and policies [59, 56]. By contrast, policy optimization algorithms studied in theory often diverge from these principles and incorporate some tweaks, such as (i) asymmetric updates, where one agent takes a much smaller learning rate than the others (two time-scale) [11] or waits until the other agents learn an approximate best response [62]; and (ii) batch-like learning, where policies are optimized to sufficient precision with respect to the current value estimate before the next value update [8]. There is so far a lacking of systematic studies on the performance of the more vanilla policy optimization algorithms following the above two principles, even under the setting where full-information feedback from the game is available.

Towards bridging these gaps, this paper studies policy optimization algorithms for Markov games, with a focus on algorithms with symmetric updates and simultaneous learning of values and policies. Our contributions can be summarized as follows:

- We propose an algorithm framework for two-player zero-sum Markov games in the full-information setting (Section 3). This framework unifies many existing and new policy optimization algorithms such as Nash V-Learning, Gradient Descent/Ascent, as well as seemingly disparate algorithms such as Nash Q-Learning (Section 3.2). We prove that the *state-wise average policy* outputted by the above algorithm is an approximate Nash Equilibrium (NE), so long as suitable per-state weighted regrets are bounded (Section 3.1). This generic result can be instantiated in a modular fashion to derive convergence guarantees for the many examples above.

- We instantiate our framework to show that a new algorithm based on Optimistic Follow-The-Regularized-Leader (OFTRL) and smooth value updates finds an $\widetilde{\mathcal{O}}(T^{-5/6})$ approximate NE in $T$ iterations (Section 4). This improves over the current best rate of $\widetilde{\mathcal{O}}(T^{-1/2})$ achieved by symmetric policy optimization type algorithms. In addition, we also propose a slightly modified OFTRL algorithm that further improves the rate to $\widetilde{\mathcal{O}}(T^{-1})$, which matches with the known best rate for all policy optimization type algorithm.

- We additionally extend the above OFTRL algorithm to multi-player general-sum Markov games and show an $\widetilde{\mathcal{O}}(T^{-3/4})$ convergence rate to Coarse Correlated Equilibria (CCE), which is also the first rate faster than $\widetilde{\mathcal{O}}(T^{-1/2})$ for policy optimization in general-sum Markov games (Section 4.1).

- We perform simulations on a carefully constructed zero-sum Markov game with $H = 2$ layers to verify our convergence guarantees. The numerical tests further suggest the importance of *smooth value updates*: the Independent Natural Policy Gradient algorithm (as one instantiation of our algorithm framework with "eager" value updates) appears to converge much slower (Section 5).

## 1.1 Related work

**Two-player zero-sum MGs** Markov games (MGs) [30] (also known as Stochastic Games [47]) is a widely studied model for multi-agent reinforcement learning. In the most basic setting of two-player zero-sum MGs, algorithms for computing the NE have been extensively studied in both the full-information setting [31, 20, 19] and the sample-based/online setting [6, 52, 22, 48, 58, 2, 55, 3, 33, 25, 21, 57, 10, 34]. Our algorithm framework incorporates (the full-information version of) several algorithms in this line of work.

**Policy optimization for zero-sum MGs** Policy optimization for single-agent Markov Decision Processes has been extensively in a recent line of work, e.g. [1, 5, 32, 46, 37, 35, 7, 15, 54] and the many references therein. For two-player zero-sum MGs, the Nash V-Learning algorithm of Bai et al. [3] (originally proposed for the sample-based online setting) can be viewed as an independent policy optimization algorithm, and can be adapted to the full-information setting with $\widetilde{\mathcal{O}}(T^{-1/2})$ convergence rate. Daskalakis et al. [11] prove that the independent policy gradient algorithm with an asymmetric two time-scale learning rate can learn the NE (for one player only) with polynomial iteration/sample complexity. Zhao et al. [62] show that another asymmetric algorithm that simulates a policy gradient/best response dynamics converges to NE with $\widetilde{\mathcal{O}}(1/T)$ rate. Cen et al. [8] use a symmetric optimistic (extragradient) subroutine for matrix games to learn zero-sum MGs in a *layer-wise* fashion (the matrix games at each state are learned to sufficient precision before the

backup, more like a Value Iteration type algorithm), and also derive an $\widetilde{\mathcal{O}}(1/T)$ convergence rate. The closest to our work is Wei et al. [53] which proves that Optimistic Gradient Descent/Ascent (OGDA), combined with smooth value updates at all layers simultaneously, converges to an NE with $\widetilde{\mathcal{O}}(T^{-1/2})$ rate for both the average duality gap and the last iterate. The $\widetilde{\mathcal{O}}(T^{-5/6})$ rate of our OFTRL algorithm improves over [53] and is the first such faster rate for symmetric, policy optimization type algorithms. The $\widetilde{\mathcal{O}}(1/T)$ rate of our modified OFTRL algorithm matches with rates in [62, 7], while still maintaining symmetric update and simultaneous learning of values and policies.

**Multi-player general-sum MGs** A recent line of work shows that a generalization of the V-learning algorithm to the multi-player general-sum setting can learn Coarse Correlated Equilibria (CCE) [49, 24, 36] and Correlated Equilibria (CE) [49, 24]. The algorithmic designs in these works are specially tailored to the sample-based setting, where the best possible rate is $\widetilde{\mathcal{O}}(T^{-1/2})$.[2] In contrast, this paper considers the full-information setting and proposes new algorithms achieving the faster $\widetilde{\mathcal{O}}(T^{-3/4})$ for learning CCE in general-sum MGs. The complexity of computing or learning a Markovian or stationary Markovian CCE has been studied in [13, 26]. Another recent line of work considers learning NE in Markov Potential Games [60, 29, 49, 16, 61], which can be seen as a cooperative-type subclass of general-sum MGs.

**Optimistic algorithms in normal-form games** Technically, our accelerated rates build on the recent line of work on faster rates for optimistic no-regret algorithms in normal-form games [50, 41, 9, 12]. Specifically, our $\tilde{\mathcal{O}}(T^{-5/6})$ rate for two-player zero-sum MGs builds upon a first-order smoothness analysis of [9], our improved $\widetilde{\mathcal{O}}(T^{-1})$ rate in the same setting (achieved by the modified OFTRL algorithm) leverages analysis of [41, 50] on bounding the summed regret over the two players, and our $\tilde{\mathcal{O}}(T^{-3/4})$ rate for multi-player general-sum MGs follows from the RVU-property [50, Definition 3]. Our incorporation of these techniques involves non-trivial new components such as *weighted* first-order smoothness bounds and handling changing game rewards.

## 2 Preliminaries

We consider the tabular episodic (finite-horizon) two-player-zero-sum Markov games (MGs), which can be denoted as $\mathcal{M}(H, \mathcal{S}, \mathcal{A}, \mathcal{B}, \mathbb{P}, r)$, where $H$ is the horizon length; $\mathcal{S}$ is the state space with $|\mathcal{S}| = S$; $\mathcal{A}, \mathcal{B}$ are the action space of the *max-player* and *min-player* respectively, with $|\mathcal{A}| = A, |\mathcal{B}| = B$; $\mathbb{P} = \{\mathbb{P}_h\}_{h=1}^{H}$ is the transition probabilities, where each $\mathbb{P}_h(s'|s, a, b)$ gives the probability of transition to state $s'$ from state-action $(s, a, b)$; $r = \{r_h\}_{h=1}^{H}$ are the reward functions, such that $r_h(s, a, b)$ is reward[3] of the max-player and $-r_h(s, a, b)$ is the reward of the min-player at time step $h$ and state-action $(s, a, b)$. In each episode, the MG starts with a deterministic initial state $s_1$. Then at each time step $1 \le h \le H$, both players observes the state $s_h$, the max-player takes an action $a_h \in \mathcal{A}$, and the min-player takes an action $b_h \in \mathcal{B}$. Then, both players receive their rewards $r_h(s_h, a_h, b_h)$ and $-r_h(s_h, a_h, b_h)$, respectively, and the system transits to the next state $s_{h+1} \sim \mathbb{P}_h(\cdot|s_h, a_h, b_h)$.

**Policies & value functions** A (Markov) policy $\mu$ of the max-player is a collection of policies $\mu = \{\mu_h : \mathcal{S} \to \Delta_{\mathcal{A}}\}_{h=1}^{H}$, where each $\mu_h(\cdot|s_h) \in \Delta_{\mathcal{A}}$ specifies the probability of taking action $a_h$ at $(h, s_h)$. Similarly, a (Markov) policy $\nu$ of the min-player is defined as $\nu = \{\nu_h : \mathcal{S} \to \Delta_{\mathcal{B}}\}$. For any policy $(\mu, \nu)$ (not necessarily Markov), we use $V_h^{\mu,\nu} : \mathcal{S} \to \mathbb{R}$ and $Q_h^{\mu,\nu} : \mathcal{S} \times \mathcal{A} \times \mathcal{B} \to \mathbb{R}$ to denote the value function and Q-function at time step $h$, respectively, i.e.

$$V_h^{\mu,\nu}(s) := \mathbb{E}_{\mu,\nu}\left[\sum_{h'=h}^{H} r_{h'}(s_{h'}, a_{h'}, b_{h'}) \mid s_h = s\right], \tag{1}$$

$$Q_h^{\mu,\nu}(s, a, b) := \mathbb{E}_{\mu,\nu}\left[\sum_{h'=h}^{H} r_{h'}(s_{h'}, a_{h'}, b_{h'}) \mid s_h = s, a_h = a, b_h = b\right]. \tag{2}$$

For notational simplicity, we use the following abbreviation: $[\mathbb{P}_h V](s, a, b) := \mathbb{E}_{s' \sim \mathbb{P}_h(\cdot|s,a,b)} V(s')$ for any value function $V$. By definition of the value functions and Q-functions, we have the following Bellman equations

$$Q_h^{\mu,\nu}(s, a, b) = \left(r_h + \mathbb{P}_h V_{h+1}^{\mu,\nu}\right)(s, a, b),$$

---

[2]Even if we specialize their algorithms to the full-information setting, the attained rates are no better than $\widetilde{\mathcal{O}}(T^{-1/2})$ because the bandit subroutines they deployed in V-learning converge no faster than $\Omega(T^{-1/2})$ even in the simplest setting of full-information matrix games.

[3]This assumes deterministic rewards; our results can be generalized directly to the case of stochastic rewards.

$$V_h^{\mu,\nu}(s,a,b) = \mathbb{E}_{a\sim\mu_h(\cdot|s),b\sim\nu_h(\cdot|s)}[Q_h^{\mu,\nu}(s,a,b)] = \langle Q_h^{\mu,\nu}(s,\cdot,\cdot), \mu(\cdot|s) \times \nu(\cdot|s) \rangle.$$

The goal for the max-player is to maximize the value function, whereas the goal for the min-player is to minimize the value function.

**Best response & Nash equilibrium** For any Markov policy $\mu$ of a max-player, there exists a best response for the min-player, which can be taken as a Markov policy $\nu^\dagger(\mu)$ such that $V_h^{\mu,\nu^\dagger(\mu)}(s) = \inf_\nu V_h^{\mu,\nu}(s)$ for all $(s,h) \in \mathcal{S} \times [H]$. For simplicity we define $V_h^{\mu,\dagger} := V_h^{\mu,\nu^\dagger(\mu)}$. By symmetry, we can also define $\mu^\dagger(\nu)$ and $V_h^{\dagger,\nu}$. It is known (e.g. [17]) that there exist Markov policies $(\mu^\star, \nu^\star)$ that perform optimally against best responses. These policies are also equivalent to Nash Equilibria (NEs) of the game, where no player can gain by switching to a different policy unilaterally. It can also be shown that any NE $(\mu^\star, \nu^\star)$ satisfies the following minimax equation
$$\sup_\mu \inf_\nu V_h^{\mu,\nu}(s) = V_h^{\mu^\star,\nu^\star}(s) = \inf_\nu \sup_\mu V_h^{\mu,\nu}(s).$$

Thus, while the NE policy $(\mu^\star, \nu^\star)$ may not be unique, all of them share the same value functions, which we denote as $V_h^\star := V_h^{\mu^\star,\nu^\star}$. The Q-function $Q_h^\star$ can be defined similarly. In this paper, our main goal is to find an approximate NE, which is formally defined below.

**Definition 1** ($\varepsilon$-approximate Nash Equilibrium). *For any $\varepsilon \geq 0$, a policy $(\mu,\nu)$ is an $\varepsilon$-approximate Nash Equilibrium ($\varepsilon$-NE) if* $\mathrm{NEGap}(\mu,\nu) := V_1^{\dagger,\nu}(s_1) - V_1^{\mu,\dagger}(s_1) \leq \varepsilon$.

**Full-information setting** We consider finding an approximate NE under the full-information setting, where we can query the exact value of $r_h + \mathbb{P}_h V_{h+1} \in \mathbb{R}^{\mathcal{S} \times \mathcal{A} \times \mathcal{B}}$ for any layer $h \in [H]$ and $V$ function $V_{h+1} \in \mathbb{R}^{\mathcal{S}}$. Further, the majority of algorithms are policy optimization algorithms that only query $(r_h + \mathbb{P}_h V_{h+1})\nu_h \in \mathbb{R}^{\mathcal{S} \times \mathcal{A}}$ and $(r_h + \mathbb{P}_h V_{h+1})^\top \mu_h \in \mathbb{R}^{\mathcal{S} \times \mathcal{B}}$ for policies $\mu_h, \nu_h$. (See Appendix I.1 for additional discussions.)

**Additional notation** For any $(h,s) \in [H] \times \mathcal{S}$ and Q function $Q_h : \mathcal{S} \times \mathcal{A} \times \mathcal{B} \to \mathbb{R}$, we define shorthand $[(\mu_h)^\top Q_h \nu_h](s) := \langle Q_h(s,\cdot,\cdot), \mu_h(\cdot|s) \times \nu_h(\cdot|s) \rangle$ for any policy $(\mu,\nu)$. Similarly, we let $[Q_h\nu_h](s,\cdot) := \mathbb{E}_{b\sim\nu_h(\cdot|s)}[Q_h(s,\cdot,b)] \in \mathbb{R}^A$, and $[Q_h^\top\mu_h](s,\cdot) := \mathbb{E}_{a\sim\mu_h(\cdot|s)}[Q_h(s,a,\cdot)] \in \mathbb{R}^B$. We use $A \vee B := \max\{A,B\}$.

## 3 An algorithm framework for zero-sum Markov games

We begin by presenting an algorithm framework that unifies many existing and new algorithms for two-player zero-sum Markov Games, and its performance guarantee that could be specialized to yield concrete convergence results for many specific algorithms.

Our algorithm framework, described in Algorithm 1, consists of two main components: the *policy update* step computing policies $(\mu^t, \nu^t)$, and the *value update* step computing the Q estimate $Q_h^t$'s.

**Policy update via matrix game algorithms** In the policy update step (Line 5), for each $(h,s)$, the two players update policies $(\mu_h^t(\cdot|s), \nu_h^t(\cdot|s))$ at $(h,s)$ using some matrix game algorithm MatrixGameAlg which takes as input all past Q matrices and all past policies of both players. The MatrixGameAlg offers a flexible interface that allows many choices such as the matrix NE subroutine over the most recent Q matrix MatrixNE($Q_h^{t-1}(s,\cdot,\cdot)$), or any independent no-regret algorithm (for both players), such as Follow-The-Regularized-Leader (FTRL) (10) or projected Gradient Descent-Ascent (11) considered in the examples later.

**Value update with learning rate** $\{\beta_t\}$ For any $(h,s,a,b)$, the value update step (Line 6) updates $Q_h^t(s,a,b)$ by the newest value function $r_h + \mathbb{P}_h[(\mu_{h+1}^t)^\top Q_{h+1}^t \nu_{h+1}^t]$ propagated from layer $h+1$, using a sequence of learning rates $\{\beta_t\}_{t\geq 1}$ which we assume to be within $[0,1]$ (with $\beta_1 := 1$). $\{\beta_t\}$ controls the speed of the value update, with two important special cases:

(1) *Eager* value updates, where we set $\beta_t = 1$ so that $Q_h^t$ performs *policy evaluation* of the current policy $(\mu^t, \nu^t)$, that is, $Q_h^t = Q_h^{\mu^t,\nu^t}$.

(2) *Smooth* (incremental) value updates, where we choose $\beta_t \to 0$ as $t \to \infty$. In this case, the $Q_h^t$ moves slower (resembling a *critic* in Actor-Critic like algorithms), and becomes a weighted average of all past updates. A standard choice that is frequently used is from [23] (and many subsequent work),
$$\beta_t = \alpha_t := (H+1)/(H+t). \tag{3}$$

---

**Algorithm 1** Algorithm framework for two-player zero-sum Markov Games

---

1: **Require:** Learning rate $\{\beta_t\}_{t\geq 1} \subset [0,1]$ (with $\beta_1 = 1$); Algorithm MatrixGameAlg.
2: **Initialize:** $Q_h^0(s,a,b) \leftarrow H - h + 1$ for all $(h,s,a,b) \in [H] \times \mathcal{S} \times \mathcal{A} \times \mathcal{B}$.
3: **for** $t = 1, \ldots, T$ **do**
4:    **for** $h = H, \ldots, 1$ **do**
5:      **Policy update:** Update policies for all $s \in \mathcal{S}$:

$$\left(\mu_h^t(\cdot|s), \nu_h^t(\cdot|s)\right) \leftarrow \mathsf{MatrixGameAlg}\left(\left\{Q_h^i(s,\cdot,\cdot)\right\}_{i=1}^{t-1}, \left\{\mu_h^i(\cdot|s)\right\}_{i=1}^{t-1}, \left\{\nu_h^i(\cdot|s)\right\}_{i=1}^{t-1}\right).$$

6:      **Value update:** Update Q-value for all $(s,a,b) \in \mathcal{S} \times \mathcal{A} \times \mathcal{B}$:

$$Q_h^t(s,a,b) \leftarrow (1-\beta_t)Q_h^{t-1}(s,a,b) + \beta_t\left(r_h + \mathbb{P}_h[(\mu_{h+1}^t)^\top Q_{h+1}^t \nu_{h+1}^t]\right)(s,a,b). \quad (4)$$

7: **Output**: State-wise average policy $(\widehat{\mu}^T, \widehat{\nu}^T)$, with $\beta_T^t$ defined in (6):

$$\widehat{\mu}_h^T(\cdot|s) \leftarrow \sum_{t=1}^T \beta_T^t \mu_h^t(\cdot|s), \quad \widehat{\nu}_h^T(\cdot|s) \leftarrow \sum_{t=1}^T \beta_T^t \nu_h^t(\cdot|s) \quad \text{for all } (h,s) \in [H] \times \mathcal{S}. \quad (5)$$

---

For any $\{\beta_t\}$, the update (4) implies that

$$Q_h^t(s,a,b) = \sum_{i=1}^t \beta_t^i \left(\left[r_h + \mathbb{P}_h[(\mu_{h+1}^i)^\top Q_{h+1}^i \nu_{h+1}^i]\right](s,a,b)\right),$$

where $\beta_t^i$'s are a group of weights summing to one ($\sum_{i=1}^t \beta_t^i = 1$) defined as

$$\beta_t^t = \beta_t; \qquad \beta_t^i = \prod_{j=i+1}^t (1-\beta_j)\beta_i, \ \text{ for } \ i \in [t-1]. \quad (6)$$

Note that with smooth value updates ($\beta_t < 1$), $Q_h^t$ is not necessarily the Q-function of any policy. Upon finishing, the algorithm outputs the *state-wise average policy* $(\widehat{\mu}^T, \widehat{\nu}^T)$ defined in (5), where each $\widehat{\mu}_h^T(\cdot|s)$ is the weighted average of $\mu_h^t(\cdot|s)$ using weights $\{\beta_T^t\}_{t=1}^T$ (and similarly for $\widehat{\nu}^T$), which we remark can be implemented efficiently using moving averages (cf. Appendix C.1).

**Symmetric & simultaneous learning, (de)centralization** We remark that Algorithm 1 by definition performs simultaneous learning (of policies and values) at all layers, and also yields symmetric (policy) updates if MatrixGameAlg is a symmetric algorithm with respect to $\mu$ and $\nu$. Also, although Algorithm 1 appears to be a *centralized* algorithm as it maintains Q values in (4), this does not preclude possibilities that the algorithm can be executed in a *decentralized* fashion. This can happen e.g. when the Q-update (4) can be rewritten as an equivalent V-update (cf. Example 1 & 2).

### 3.1 Theoretical guarantee

We are now ready to state the main theoretical guarantee of Algorithm 1, which states that the $(\widehat{\mu}^T, \widehat{\nu}^T)$ is an approximate NE, as long as the algorithm achieves low *per-state weighted regrets* w.r.t. weights $\{\beta_t^i\}_{i=1}^t$, defined as

$$\mathrm{reg}_{h,\mu}^t(s) := \max_{\mu^\dagger \in \Delta_\mathcal{A}} \sum_{i=1}^t \beta_t^i \left\langle \mu^\dagger - \mu_h^i(\cdot|s), [Q_h^i \nu_h^i](s,\cdot)\right\rangle,$$
$$\mathrm{reg}_{h,\nu}^t(s) := \max_{\nu^\dagger \in \Delta_\mathcal{B}} \sum_{i=1}^t \beta_t^i \left\langle \nu_h^i(\cdot|s) - \nu^\dagger, [(Q_h^i)^\top \mu_h^i](s,\cdot)\right\rangle, \quad (7)$$
$$\mathrm{reg}_h^t := \max_{s \in \mathcal{S}} \max\{\mathrm{reg}_{h,\mu}^t(s), \mathrm{reg}_{h,\nu}^t(s)\}.$$

**Theorem 2** (Main guarantee of Algorithm 1). *Suppose that the per-state regrets can be upper-bounded as $\mathrm{reg}_h^t \leq \overline{\mathrm{reg}}_h^t$ for all $(h,t) \in [H] \times [T]$, where $\overline{\mathrm{reg}}_h^t$ is non-increasing in $t$: $\overline{\mathrm{reg}}_h^t \geq \overline{\mathrm{reg}}_h^{t+1}$ for all $t \geq 1$. Then, the output policy $(\widehat{\mu}^T, \widehat{\nu}^T)$ of Algorithm 1 satisfies*

$$\mathrm{NEGap}(\widehat{\mu}^T, \widehat{\nu}^T) \leq C\left[H \max_{h \in [H]} \overline{\mathrm{reg}}_h^T + H^2 c_\beta^H \log T \cdot \frac{1}{T}\sum_{t=1}^T \max_{h \in [H]} \overline{\mathrm{reg}}_h^t\right] \quad (8)$$

*for all $T \geq 2$ and some absolute constant $C > 0$, where $c_\beta$ is a constant depending on $\{\beta_t\}_{t\geq 1}$:*

$$c_\beta := \sup_{j \geq 1} \sum_{t=j}^\infty \beta_t^j \geq 1. \quad (9)$$

*Specifically, $c_\beta = \left(1 + \frac{1}{H}\right)$ if $\beta_t = \alpha_t = \frac{H+1}{H+t}$, and $c_\beta = 1$ if $\beta_t = 1$.*

Bound (8) is typically dominated by the second term on the right hand side, suggesting that the NEGap can be bounded by the average weighted regret $\widetilde{\mathcal{O}}\Big(\frac{1}{T}\sum_{t=1}^{T}\max_{h}\overline{\mathrm{reg}}_{h}^{t}\Big)$, if $c_{\beta}^{H}=O(1)$. Theorem 2 serves as a modular tool for analyzing a broad class of algorithms: As long as this average regret is sublinear in $T$ (including—but not limited to—choosing MatrixGameAlg as uncoupled no-regret algorithms), the output policy will be an approximate NE. We emphasize though that this result is not yet end-to-end, as each $\mathrm{reg}_{h}^{t}$ is a weighted regret w.r.t. *the particular set of weights* $\big\{\beta_{t}^{i}\big\}_{i=1}^{t}$, minimizing which may require careful algorithm designs and/or case-by-case analyses. We provide some concrete examples in Section 3.2 to demonstrate the usefulness of Theorem 2.

We remark that the state-wise average policy considered in Theorem 2 is an average policy that is also *Markovian* by definition, which is different from existing work which considers either the (Markovian) last iterate [53] or non-Markovian average policies (e.g. [3]). However, this guarantee relies on full-information feedback (so that per-state regret bounds are available), and it remains an open question how such guarantees could be generalized to sample-based settings.

**Proof overview**    The proof of Theorem 2 follows by (1) bounding $\mathrm{NEGap}(\widehat{\mu}^{T},\widehat{\nu}^{T})$ in terms of per-state regrets w.r.t. the *Nash value functions* $Q_{h}^{\star}$'s by performance difference arguments (Lemma C.1); (2) recursively bounding the value estimation error $\delta_{h}^{t}:=\|Q_{h}^{t}-Q_{h}^{\star}\|_{\infty}$ (Lemma C.2) which yields the constant $c_{\beta}$; and (3) combining the above to translate the regret from $Q_{h}^{\star}$'s to $Q_{h}^{t}$'s (which we assume to be bounded by $\overline{\mathrm{reg}}_{h}^{t}$) and obtain the theorem. The full proof can be found in Appendix C.

## 3.2   Examples

We now demonstrate the generality of Algorithm 1 and Theorem 2 by showing that they subsume many existing algorithms (and yield new algorithms) for two-player-zero-sum Markov games, and provide new guarantees with the particular output policy (5).

**Example 1** (Nash V-Learning [3], full-information version)**:**  The full algorithm (Algorithm 5) can be found in Appendix D.1. The algorithm is a special case of Algorithm 1 with $\beta_{t}=\alpha_{t}=(H+1)/(H+t)$, and MatrixGameAlg chosen as the weighted FTRL algorithm

$$\mu_{h}^{t}(a|s)\propto_{a}\exp\Bigg(\frac{\eta}{w_{t-1}}\sum_{i=1}^{t-1}w_{i}\big[Q_{h}^{i}\nu_{h}^{i}\big](s,a)\Bigg),\ \nu_{h}^{t}(b|s)\propto_{b}\exp\Bigg(-\frac{\eta}{w_{t-1}}\sum_{i=1}^{t-1}w_{i}\big[\big(Q_{h}^{i}\big)^{\top}\mu_{h}^{i}\big](s,b)\Bigg),\tag{10}$$

where $w_{t}:=\alpha_{t}^{t}/\alpha_{t}^{1}$. Combining Theorem 2 with the standard regret bound of weighted FTRL, this algorithm achieves $\mathrm{NEGap}(\widehat{\mu}^{T},\widehat{\nu}^{T})\leq\widetilde{\mathcal{O}}(H^{7/2}/\sqrt{T})$ choosing $\eta\asymp 1/\sqrt{T}$ (Proposition D.2).

Additionally, although the original Nash V-learning algorithm [3] updates the V values (which makes the algorithm implementable in a decentralized fashion) instead of the Q values used in Algorithm 1, these two forms are actually equivalent in the full-information setting (Proposition D.1). ◊

Compared with the $\widetilde{\mathcal{O}}(\sqrt{H^{5}S\max\{A,B\}/T})$ guarantee of (the non-Markovian output policy of) Nash V-Learning in the sample-based online setting [3, 51, 24], our rate achieves better (logarithmic) $S,A,B$ dependence due to our full-information setting, and worse $H$ dependence which happens as our output policy is the (Markovian) state-wise average policies, whose guarantee (Theorem 2) follows from a different analysis.

**Example 2** (GDA-Critic)**:**  This algorithm is a special case of Algorithm 1 with $\beta_{t}=\alpha_{t}=(H+1)/(H+t)$, and MatrixGameAlg as projected gradient descent/ascent (GDA), i.e.,

$$\mu_{h}^{t}(\cdot|s)\leftarrow\mathcal{P}_{\Delta_{\mathcal{A}}}\big(\mu_{h}^{t-1}(\cdot|s)+\eta\big[Q_{h}^{t-1}\nu_{h}^{t-1}\big](s)\big),\ \ \nu_{h}^{t}(\cdot|s)\leftarrow\mathcal{P}_{\Delta_{\mathcal{B}}}\big(\nu_{h}^{t-1}(\cdot|s)-\eta\big(\big[Q_{h}^{t-1}\big]^{\top}\mu_{h}^{t-1}\big)(s)\big).\tag{11}$$

Similar as Nash V-Learning, GDA-Critic also admits an equivalent form with V value updates (full description in Algorithm 6). As GDA achieves weighted regret bounds with any monotone weights including $\big\{\alpha_{t}^{i}\big\}_{i=1}^{t}$ (Lemma B.1), we can invoke Theorem 2 to show that this algorithm achieves $\mathrm{NEGap}(\widehat{\mu}^{T},\widehat{\nu}^{T})\leq\widetilde{\mathcal{O}}(H^{7/2}(A\vee B)^{1/2}/\sqrt{T})$ if we choose $\eta\asymp 1/\sqrt{T}$ (Proposition D.4).

The GDA-critic algorithm is also similar to the OGDA-MG algorithm of Wei et al. [53], except that we use the (non-optimistic) vanilla version of GDA. To our best knowledge, the above algorithm and guarantee are not known. We remark that even ignoring difference between GDA and OGDA, the

above guarantee cannot be obtained by direct adaptation of the results of [53] which focus on either the average duality gap and/or last-iterate convergence. ◊

Besides the above examples, Algorithm 1 also incorporates the following algorithms which are typically not categorized as policy optimization algorithms (see Appendix I.2 for a discussion of the categorization).

**Example 3** (Nash Q-Learning [20, 3], full-information version)**:** This algorithm is a special case of Algorithm 1 with $\beta_t = \alpha_t = (H+1)/(H+t)$ and MatrixGameAlg as the matrix Nash subroutine

$$(\mu_h^t(\cdot|s), \nu_h^t(\cdot|s)) \leftarrow \mathsf{MatrixNE}(Q_h^{t-1}(s,\cdot,\cdot)) := \arg\left(\min_{\mu\in\Delta_{\mathcal{A}}} \max_{\nu\in\Delta_{\mathcal{B}}} \mu^\top Q_h^{t-1}(s,\cdot,\cdot)\nu\right).$$

(Full description in Algorithm 7.) Although $\mathsf{MatrixNE}(Q_h^{t-1}(s,\cdot,\cdot))$ is not by default a no-regret algorithm, using the fact that $\left\|Q_h^t - Q_h^{t-1}\right\|_\infty$ is small (due to the small $\alpha_t$) we can show that it is close to a (hypothetical) "Be-The-Leader" style algorithm that computes the matrix NE of the current Q matrix $Q_h^t$ which achieves $\leq 0$ regret (Lemma D.3). Combining this with Theorem 2 shows that this algorithm achieves $\mathrm{NEGap}(\widehat{\mu}^T, \widehat{\nu}^T) \leq \widetilde{\mathcal{O}}(H^4/T)$ (Proposition D.5). ◊

**Example 4** (Nash Policy Iteration (Nash-PI))**:** This classical algorithm (Algorithm 8) performs iterative policy evaluation and policy improvement (also similar to Nash Value Iteration [47, 2, 33]):

$$(\mu_h^{t+1}(\cdot|s), \nu_h^{t+1}(\cdot|s)) \leftarrow \mathsf{MatrixNE}(Q_h^{\mu^t,\nu^t}(s,\cdot,\cdot)). \tag{12}$$

This is also a special case of Algorithm 1 with $\beta_t = 1$ and MatrixGameAlg set as MatrixNE. It is a standard result that this algorithm converges exactly (achieving zero NE gap) in $H$ steps, and this fact can be obtained using our framework as well (Proposition D.6). ◊

## 4 Fast convergence of optimistic FTRL

In this section, we instantiate Algorithm 1 by choosing MatrixGameAlg as the Optimistic Follow-The-Regularized-Leader (OFTRL) algorithm. OFTRL is also an uncoupled no-regret algorithm that is known to enjoy faster convergence than standard FTRL under additional loss smoothness assumptions [41, 50, 9, 12]. We show that, using OFTRL, Algorithm 1 enjoys faster convergence than the $\widetilde{\mathcal{O}}(1/\sqrt{T})$ rate of using FTRL or GDA (cf. Example 1 & 2).

Concretely, we use the following weighted OFTRL algorithm at each $(h, s, t)$:

$$\begin{aligned}
\mu_h^t(a|s) &\propto_a \exp\left((\eta/w_t) \cdot \left[\sum_{i=1}^{t-1} w_i(Q_h^i \nu_h^i)(s,a) + w_{t-1}(Q_h^{t-1}\nu_h^{t-1})(s,a)\right]\right), \\
\nu_h^t(b|s) &\propto_b \exp\left(-(\eta/w_t) \cdot \left[\sum_{i=1}^{t-1} w_i((Q_h^i)^\top \mu_h^i)(s,b) + w_{t-1}((Q_h^{t-1})^\top \mu_h^{t-1})(s,b)\right]\right),
\end{aligned} \tag{13}$$

where $w_t$ is the same weights as defined in Example 1, and we choose $\beta_t = \alpha_t = (H+1)/(H+t)$.

**Theorem 3** (Fast convergence of OFTRL in zero-sum Markov Games)**.** *Suppose Algorithm 1 is instantiated with $\beta_t = \alpha_t = (H+1)/(H+t)$ and MatrixGameAlg to be the OFTRL algorithm* (13) *with any $\eta \leq 1/H$ (full description in Algorithm 9). Then the per-state regret can be bounded as follows for some absolute constant $C > 0$:*

$$\mathrm{reg}_h^t \leq \overline{\mathrm{reg}}_h^t := C\left[\frac{H^2\log(A\vee B)}{\eta t} + \eta^5 H^6\right] \quad \text{for all } (h,t) \in [H]\times[T]. \tag{14}$$

*Further, choosing $\eta = \mathrm{poly}(H, \log(A\vee B), \log T) \cdot T^{-1/6}$, the output (state-wise average) policy $(\widehat{\mu}^T, \widehat{\nu}^T)$ achieves approximate NE guarantee*

$$\mathrm{NEGap}(\widehat{\mu}^T, \widehat{\nu}^T) \leq \mathcal{O}\left(\mathrm{poly}(H, \log(A\vee B), \log T) \cdot T^{-5/6}\right). \tag{15}$$

To our best knowledge, the $\widetilde{\mathcal{O}}(T^{-5/6})$ rate asserted in Theorem 3 is the first rate faster than the standard $\widetilde{\mathcal{O}}(1/\sqrt{T})$ for symmetric, policy optimization type algorithms in two-player zero-sum Markov games. The closest existing result to this is of Wei et al. [53], who analyze the OGDA algorithm with smooth value updates and show a $\widetilde{\mathcal{O}}(1/\sqrt{T})$ convergence of both the average NEGap

and the NEGap of the last-iterate. However, these only imply at most a $\widetilde{\mathcal{O}}(1/\sqrt{T})$ rate for the average policies, and not our faster rate[4]. Cen et al. [8], Zhao et al. [62] show $\widetilde{\mathcal{O}}(1/T)$ convergence of algorithms with optimistic gradient-based policy updates, which are however very different styles of algorithms that either performs *layer-wise learning* (the matrix games at each state are learned to sufficient precision before the backup) similar as Value Iteration (the matrix games at each state are learned to sufficient precision before the backup) [8], or uses strongly *asymmetric* updates that simulate a policy gradient-best response dynamics [62]. By contrast, our Algorithm 9 (as well as its modified version in Algorithm 10 with $\widetilde{\mathcal{O}}(T^{-1})$ rate) runs symmetric no-regret dynamics for both players, simultaneously at all layers.

**Proof overview** The proof of Theorem 3 (deferred to Appendix E) builds upon the recent line of work on fast convergence of optimistic algorithms [41, 50, 9], in particular the work of Chen and Peng [9] which shows an $\widetilde{\mathcal{O}}(T^{-5/6})$ convergence rate of OFTRL for two-player normal-form games. Our regret bound (14) generalizes this result non-trivially by additionally handling (1) The *weighted* regret, which requires bounding the weighted stability of the OFTRL iterates by a new analysis of the potential functions (Lemma B.4), and (2) The errors induced by *changing game matrices*, as $Q_h^t(s, \cdot, \cdot)$ changes over $t$. Plugging (14) into Theorem 2 yields the policy guarantee (15).

**Modified OFTRL algorithm with $\widetilde{\mathcal{O}}(T^{-1})$ rate** We further slightly modify Algorithm 9 to design a new OFTRL style algorithm with $\widetilde{\mathcal{O}}(T^{-1})$ convergence rate (Algorithm 10 and Theorem F.1), which improves over the $\widetilde{\mathcal{O}}(T^{-5/6})$ of Theorem 3 and matches the known best convergence rate for policy optimization type algorithms in two-player zero-sum Markov games. Algorithm 10 still uses OFTRL in its policy update step, and the main difference from Algorithm 9 is in its value update step: Rather than maintaining a single $Q_h^t$, the two players now each maintain their own value estimate $\overline{Q}_h^t, \underline{Q}_h^t$ which are still updated in an incremental fashion similar to (though not strictly speaking an instantiation of) the update rule (4) in our main algorithm framework. Details of the algorithm as well as the proofs are deferred to Appendix F.

## 4.1 Extension to multi-player general-sum Markov games

Our fast convergence result can be extended to the more general setting of *multi-player general-sum Markov games*. Concretely, we consider general-sum Markov games with $m \geq 2$ players, $S$ states, $H$ steps, where the $i$-th player has action space $\mathcal{A}_i$ with $A_{\max} := \max_{i \in [m]} |\mathcal{A}_i|$ and her own reward function. The goal is to find a correlated policy over all players that is an approximate Coarse Correlated Equilibrium (CCE) of the game (see Appendix G.1 for the detailed setup).

We show that the OFTRL algorithm works for general-sum Markov games as well, with a fast $\widetilde{\mathcal{O}}(T^{-3/4})$ convergence to CCE. The formal statement and proof is in Theorem G.1 & Appendix G.4.

**Theorem 4** (Fast convergence of OFTRL in general-sum Markov Games; Informal version of Theorem G.1). *For $m$-player general-sum Markov Games, running the OFTRL algorithm (Algorithm 12) for $T$ rounds, the output (correlated) policy $\widehat{\pi}$ is an $\varepsilon$-approximate CCE, where*

$$\varepsilon \leq \mathcal{O}\big(\mathrm{poly}(H, \log A_{\max}, \log T) \cdot (m-1)^{1/2} \cdot T^{-3/4}\big).$$

A baseline result for this problem would be $\widetilde{\mathcal{O}}(T^{-1/2})$, which may be obtained directly by adapting existing proofs of the V-Learning algorithm [49, 24] to the full-information setting. Our Theorem 4 shows that a faster $\widetilde{\mathcal{O}}(T^{-3/4})$ rate is available by using the OFTRL algorithm, which to our best knowledge is the first such result for policy optimization in general-sum Markov games. We also remark that the output policy $\widehat{\pi}$ above is not a state-wise average policy as in the zero-sum setting, but rather a mixture policy that is in general non-Markov (cf. Algorithm 13), which is similar as (and slightly simpler than) the "certified policies" used in existing work [3, 49, 24]. The proof of Theorem 4 builds upon the RVU property of OFTRL [50] and additionally handles changing game rewards, similar as in Theorem 3. A proof sketch and comparison with the $\widetilde{\mathcal{O}}(T^{-5/6})$ analysis of the zero-sum case can be found in Appendix G.2.

---

[4]See also [18] for another example where last-iterates are *provably* slower than averages.

# 5 Simulations

We perform numerical studies on the various policy optimization algorithms. Our goal is two-fold: (1) Verify the convergence guarantees in our theorems and examples; (2) Test some other important special cases of Algorithm 1 that may not yet admit a provable guarantee.

To this end, we consider three algorithms covered by the framework in Algorithm 1:

1. **FTRL** (Nash V-Learning) with smooth value updates $\beta_t = \alpha_t$ (Example 1 & Algorithm 5). Here the output policy $(\widehat{\mu}^T, \widehat{\nu}^T)$ are the state-wise averages with weights $\{\alpha_T^i\}_{i=1}^T$, and achieves $\mathrm{NEGap}(\widehat{\mu}^T, \widehat{\nu}^T) \lesssim T^{-1/2}$ if we choose $\eta \asymp T^{-1/2}$ (Proposition D.2).

2. **OFTRL** with smooth value updates $\beta_t = \alpha_t$ (Algorithm 9). Here the output policy $(\widehat{\mu}^T, \widehat{\nu}^T)$ are the state-wise averages with weights $\{\alpha_T^i\}_{i=1}^T$, and achieves $\mathrm{NEGap}(\widehat{\mu}^T, \widehat{\nu}^T) \lesssim T^{-5/6}$ if we choose $\eta \asymp T^{-1/6}$ (Theorem 3). We also consider the more aggressive choice $\eta = 1$.

3. **INPG** (Independent Natural Policy Gradients). This algorithm is an instantiation of Algorithm 1 (cf. Appendix H.3 for formal justifications) with *eager* value updates ($\beta_t = 1$), and MatrixGameAlg chosen as standard unweighted FTRL (a.k.a. Hedge) for all $(h, s, t)$:

$$\mu_h^t(a|s) \propto_a \mu_h^{t-1}(a|s) \exp\big(\eta\big[Q_h^{t-1}\nu_h^{t-1}\big](s)\big), \ \nu_h^t(b|s) \propto_b \nu_h^{t-1}(b|s) \exp\Big(-\eta\Big[\big(Q_h^{t-1}\big)^\top\mu_h^{t-1}\Big](s)\Big).$$

For this algorithm, we choose two standard learning rates: $\eta = 1$, and $\eta = T^{-1/2}$, and use the *vanilla (state-wise) average* as the output policies (since the last-iterate is known to be cyclic):

$$\widehat{\mu}_h^T(\cdot|s) = \tfrac{1}{T}\textstyle\sum_{t=1}^T \mu_h^t(\cdot|s), \ \widehat{\nu}_h^T(\cdot|s) = \tfrac{1}{T}\sum_{t=1}^T \nu_h^t(\cdot|s) \ \text{ for all } (h, s) \in [H] \times \mathcal{S}.$$

The main motivation for considering INPG is that it is a natural generalization of both the widely-studied NPG algorithm for single-agent RL, and the standard Hedge algorithm for zero-sum matrix games. In both cases the algorithm admits favorable convergence guarantees: NPG converges with rate $\mathcal{O}(T^{-1})$ [1, 27, 38, 7] (in both last iterate and averaging) using $\eta = O(1)$; Hedge converges with rate $\mathcal{O}(T^{-1/2})$ in zero-sum matrix games (e.g. [41]) using $\eta \asymp T^{-1/2}$. However, to our best knowledge, the convergence of INPG for zero-sum Markov games is unclear, and it is commented by Wei et al. [53, Section 5] that eager value updates ($\beta_t = 1$) could cause the value function of the $(h+1)$th layer to oscillate, which make learning unstable or even biased within the $h$-th layer.

**A two-layer numerical example** We design a simple zero-sum Markov game with two layers and small state/action spaces ($H = 2$, $S = 4$, $A = 2$; see Appendix H.1 for the detailed description). The main feature of this game is that the reward in the first layer is much lower magnitude than that of the second layer (the scale is roughly $|r_1(s, \cdot, \cdot)| \approx 0.1|r_2(s, \cdot, \cdot)|$), which may exaggerate the aforementioned unstable effect. We also choose a careful initialization $(\mu^1, \nu^1)$ which is non-uniform (and modify the FTRL / OFTRL algorithms to start at this initialization, cf. Appendix H.1) but with all entries bounded in $[0.15, 0.85]$. We test all three algorithms above on this game, with this initialization, $T \in \{10^3, 3 \times 10^3, 10^4, \dots, 10^7\}$, and $\eta$ chosen correspondingly as described above.

**Results** Figure 1a plots the $\mathrm{NEGap}$ of the final output policies, one for each {algorithm, $(T, \eta)$}. Observe that FTRL converges with rate roughly $T^{-.570} \lesssim T^{-1/2}$, and OFTRL with $\eta = T^{-1/6}$ converges with rate $T^{-.835} \approx T^{-5/6}$, both corroborating our theory. Further, OFTRL with $\eta = 1$ appears to converge with rate $T^{-1}$; showing this may be an interesting open theoretical question.

On the other hand, the INPG algorithm appears to be much slower: The $\eta = 1$ version does not seem to converge, whereas the convergence of $\eta = T^{-1/2}$ version is not clear but at least substantially slower than $T^{-1/2}$ ($T^{-.308}$ given by the linear fit) .

To further understand the behavior of INPG, we visualize its *layer-wise* $\mathrm{NEGap}$'s for $h \in \{1, 2\}$ (on our example), defined as the $\mathrm{NEGap}$ of the $h$-th layer's policies with respect to $Q_h^\star$:

$$\mathrm{NEGap\text{-}Layer\text{-}}h(\mu, \nu) := \max_s \Big(\max_{\mu_h^\dagger}\Big[(\mu_h^\dagger)^\top Q_h^\star \nu_h\Big](s) - \min_{\nu_h^\dagger}\Big[\mu_h^\top Q_h^\star \nu_h^\dagger\Big](s)\Big), \ h = 1, 2.$$

Note that $\mathrm{NEGap\text{-}Layer\text{-}}1$ is a lower bound of $\mathrm{NEGap}(\mu, \nu)$ (cf. Appendix H.3) and thus needs to be minimized by any convergent algorithm. By contrast, on our example, $\mathrm{NEGap\text{-}Layer\text{-}}2$ is concerned with the last layer only, and can be minimized by any algorithm that works on matrix games.

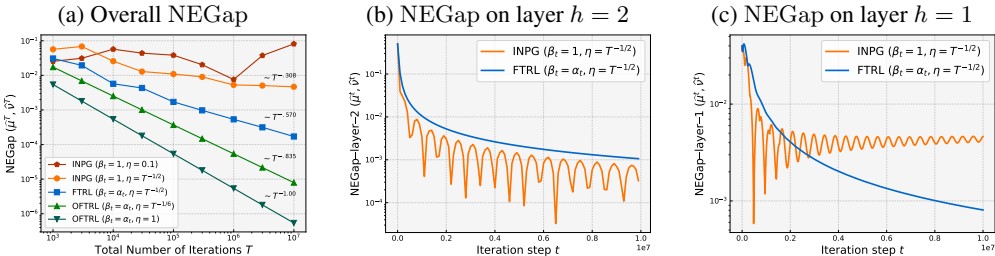

Figure 1: **(a)** NEGap of the final output policies ($y$-axis) against total # iterations $T$ ($x$-axis) on the two-layer example (cf. Appendix H.1) in log-log scale. **Each dot represents a different run with its own** $(T, \eta)$. The scalings of the form $\sim T^{-\alpha}$ are obtained via best linear fits in the log space. **(b,c)** Layer-wise NEGaps ($y$-axis, log-scale) against iteration count $t$ ($x$-axis) for {INPG, FTRL} on a **single run** with $T = 10^7$ and $\eta = T^{-1/2}$.

Figure 1b & 1c plot the layer-wise NEGap's of INPG against FTRL, on the single run with $T = 10^7$ and $\eta = T^{-1/2}$. As expected, the NEGap-Layer-2 converges nicely for both algorithms with similar rates (Figure 1b) albeit the oscillation of INPG, whereas their behavior on NEGap-Layer-1 is drastically different: FTRL still converges, whereas INPG seems to be oscillating around a non-zero bias (Figure 1c). This suggests that INPG may indeed be suffer from a non-vanishing bias in the first layer caused by the second layer's learning dynamics. (See Appendix H.2 for additional illustrations.) It would be an interesting open question to investigate the convergence of INPG theoretically.

## 6 Conclusion

This paper provides a unified framework for analyzing a large class of policy optimization algorithms for two-player zero-sum Markov games. Using our framework, we prove new fast convergence rates for the OFTRL algorithm with smooth value updates: $\widetilde{\mathcal{O}}(T^{-5/6})$ for learning Nash Equilibria two-player zero-sum Markov games, which can be further accelerated to $(T^{-1})$ by slightly modifying the framework; and $\widetilde{\mathcal{O}}(T^{-3/4})$ for learning Coarse Correlated Equilibria in multi-player general-sum Markov games. We further demonstrate the importance of smooth value updates on a simple numerical example. We believe our work opens up many other interesting directions, such as whether improved rates (e.g. $\widetilde{\mathcal{O}}(T^{-1})$) are available for the unmodified OFTRL algorithm, or further investigation of policy optimization algorithms with eager value updates (such as Independent Natural Policy Gradients). Finally, a limitation of this work is its focus on the full-information setting, and it is an important open question how to generalize our analyses to the sample-based setting.

## Acknowledgment

The authors would like to thank Chi Jin, Yuanhao Wang, Tiancheng Yu, Shicong Cen, and Song Mei for the valuable discussions. Runyu Zhang is supported by NSF AI institute: 2112085, ONR YIP: N00014-19-1-2217, NSF CNS: 2003111 and NSF CPS: 2038603.

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
