# A Technical tools

## A.1 Properties of $\alpha_t^i$

Throughout this section, the sequence $\{\beta_t^i\}_{i\in[t]}$ is defined through sequence $\{\beta_t\}_{t\geq 1}$ as in (6), and $\alpha_t^i$ is its special case with $\beta_t = \alpha_t$, where $\{\alpha_t\}_{t\geq 1}$ is defined in (3). We present some basic algebraic properties of $\alpha_t^i$ that will be used in later proofs.

**Lemma A.1.** *Given a sequence $\{\Delta_h^t\}_{h,t}$ defined by*

$$\begin{cases} \Delta_h^t = \sum_{i=1}^t \alpha_t^i \Delta_{h+1}^i + \beta_t, \\ \Delta_{H+1}^t = 0, \ \ for\ all\ t, \end{cases} \tag{16}$$

*where $\{\beta_t\}$ is non-increasing w.r.t. $t$. Then $\Delta_h^{t+1} \leq \Delta_h^t$ for all $(t,h) \in \mathbb{N} \times [H+1]$.*

*Proof.* We prove by doing backward induction on $h$. For the base case of induction, notice that the claim is true for $H+1$. Assume the claim is true for $h+1$. At step $h$, we have

$$\Delta_h^{t+1} = \sum_{i=1}^{t+1} \alpha_{t+1}^i \Delta_{h+1}^i + \beta_{t+1}$$

$$= (1-\alpha_{t+1}) \sum_{i=1}^t \alpha_t^i \Delta_{h+1}^i + \alpha_{t+1} \Delta_{h+1}^{t+1} + \beta_{t+1}$$

$$\leq (1-\alpha_{t+1}) \sum_{i=1}^t \alpha_t^i \Delta_{h+1}^i + \alpha_{t+1} \sum_{i=1}^t \alpha_t^i \Delta_{h+1}^i + \beta_t = \Delta_h^t,$$

where the inequality follows from the inductive hypothesis and $\beta_{t+1} \leq \beta_t$. $\qquad\square$

The following lemma is taken from [23].

**Lemma A.2.** *The sequence $\alpha_t^i$ satisfies the following:*

*(a) $\sum_{t=i}^\infty \alpha_t^i = 1 + 1/H$ for all $i \geq 1$.*

**Lemma A.3** (Convolution of $\beta_T^t$ with decaying sequences)**.** *From general $\{\beta_t\}_{t\geq 1}$ sequence, we have that*

$$X_T := \sum_{t=1}^T \frac{1}{t}\beta_T^t \leq \frac{2c_\beta \log(T)}{T}, \ \ for\ T \geq 2.$$

*Specifically if $\beta_t = \alpha_t$, the following holds for all $T \geq 1$:*

*(a) $A_T := \sum_{t=1}^T \alpha_T^t \cdot \frac{1}{t^2} \leq \frac{4}{T}$.*

*(b) $B_T := \sum_{t=1}^T \alpha_T^t \alpha_t \leq \frac{(H+1)^2}{H(H+T)}$*

*(c) $C_T := \sum_{t=1}^T \alpha_T^t \cdot \alpha_t^2 \leq \frac{4H}{T}$.*

*Proof.* We first prove for the inequality on $X_T$ for general $\{\beta_t\}_{t\geq 1}$ sequence. We start with showing that $X_{t+1} \leq X_t, \ \forall t \geq 1$.

$$X_{t+1} = \sum_{i=1}^{t+1} \frac{1}{i}\beta_{t+1}^i = \sum_{i=1}^t \frac{1}{i}\beta_{t+1}^i + \frac{\beta_{t+1}}{t+1}$$

$$= (1-\beta_{t+1}) \sum_{i=1}^t \frac{1}{i}\beta_t^i + \frac{\beta_{t+1}}{t+1} = (1-\beta_{t+1})X_t + \frac{\beta_{t+1}}{t+1}$$

$$\implies X_{t+1} - X_t = \beta_{t+1}\left(\frac{1}{t+1} - X_t\right).$$

Since

$$\frac{1}{t+1} - X_t = \frac{1}{t+1} - \sum_{i=1}^{t} \frac{1}{i}\beta_t^i = \sum_{i=1}^{t} \beta_t^i \left(\frac{1}{t+1} - \frac{1}{i}\right) \leq 0,$$

we have that $X_{t+1} - X_t \leq 0$. Thus

$$X_T \leq \frac{1}{T}\sum_{t=1}^{T} X_t = \frac{1}{T}\sum_{t=1}^{T}\sum_{i=1}^{t} \frac{1}{i}\beta_t^i = \frac{1}{T}\sum_{i=1}^{T} \frac{1}{i}\left(\sum_{t=i}^{T} \beta_t^i\right) \leq \frac{c_\beta}{T}\sum_{i=1}^{T} \frac{1}{i} \leq \frac{2c_\beta \log(T)}{T}.$$

Now we prove for the specific case where $\beta_t = \alpha_t$:

(a) Note that $A_1 = 1$ and we have the recursive relationship

$$A_{T+1} = (1 - \alpha_{T+1})A_T + \alpha_{T+1} \cdot \frac{1}{(T+1)^2}$$

by definition of the sequence $\alpha_T^t$. In particular this implies $A_{T+1} \leq A_T$, since $A_T$ is a weighted average of $1/t^2 \geq 1/(T+1)^2$. Therefore we have

$$A_T \leq \frac{1}{T}\sum_{t=1}^{T} A_t = \frac{1}{T}\sum_{t=1}^{T}\sum_{s=1}^{t} \alpha_t^s \cdot \frac{1}{s^2}$$

$$= \frac{1}{T}\sum_{s=1}^{T} \underbrace{\sum_{t=s}^{T} \alpha_t^s}_{\leq 1+1/H \leq 2} \cdot \frac{1}{s^2} \leq \frac{2}{T}\sum_{s=1}^{T} \frac{1}{s^2} \leq \frac{2}{T}\sum_{s=1}^{\infty} \frac{1}{s^2} \leq \frac{4}{T}.$$

Above, the step $\sum_{t=s}^{T} \alpha_t^s \leq \sum_{t=s}^{\infty} \alpha_t^s = 1 + 1/H$ follows from Lemma A.2.

(b) From the definition of $B_T$ we have that

$$B_{T+1} = \sum_{t=1}^{T+1} \alpha_{T+1}^t \alpha_t = \sum_{t=1}^{T}(1 - \alpha_{T+1})\alpha_T^t \alpha_t + \alpha_{T+1}^2 = (1 - \alpha_{T+1}^t)B_T + \alpha_{T+1}$$

$$\implies B_{T+1} = \frac{T}{H+T+1}B_T + \frac{(H+1)^2}{(H+T+1)^2} \leq \frac{T}{H+T+1}B_T + \frac{(H+1)^2}{(H+T+1)(H+T)}$$

$$\implies \left(B_{T+1} - \frac{(H+1)^2}{H(H+T+1)}\right) \leq \frac{T}{H+T+1}\left(B_T - \frac{(H+1)^2}{H(H+T)}\right).$$

Since $B_1 = \alpha_1^2 = 1 \leq \frac{(H+1)^2}{H(H+1)}$, we have that $B_T \leq \frac{(H+1)^2}{H(H+T)}$ via proof by induction.

(c) Since $\alpha_t \leq 1$, we have that $C_T \leq B_T$, thus by part (b)

$$C_T \leq B_T \leq \frac{(H+1)^2}{H(H+T)} \leq \frac{4H}{T}.$$

$\square$

Consider the sequence $\{w_t\}_{t \geq 1}$ defined by (cf. also Example 1)

$$w_t = \alpha_t^t/\alpha_t^1. \tag{17}$$

Note that we also have $w_t = \alpha_T^t/\alpha_T^1$ for any $T \geq t$.

**Lemma A.4** (Properties of $w_t$). *The following holds for all $t \geq 2$:*

*(a) $w_t/w_{t-1} = (H+t-1)/(t-1)$.*

*(b) $\left(\frac{1}{w_{t-1}} - \frac{1}{w_t}\right)\sum_{i=1}^{t-1} w_i = H/(H+1)$.*

*Proof.* (a) We have

$$\frac{w_t}{w_{t-1}} = \frac{\alpha_t}{\alpha_{t-1}(1-\alpha_t)} = \frac{(H+1)/(H+t)}{(H+1)/(H+t-1)\cdot(t-1)/(H+t)} = \frac{H+t-1}{t-1}.$$

(b) We have

$$\left(\frac{1}{w_{t-1}} - \frac{1}{w_t}\right)\sum_{i=1}^{t-1} w_i = \frac{1}{w_{t-1}}\left(1 - \frac{w_{t-1}}{w_t}\right)\sum_{i=1}^{t-1} w_i = \left(1 - \frac{w_{t-1}}{w_t}\right)\cdot\frac{1}{\alpha_{t-1}^{t-1}}$$

$$\overset{(i)}{=} \frac{H}{H+t-1}\cdot\frac{H+t-1}{H+1} = \frac{H}{H+1}.$$

Above, (i) used part (a).

$\square$

## A.2 Other technical lemmas

**Lemma A.5** (Smoothness of Exponential Weights). *Let $g_1, g_2 \in \mathbb{R}^n$ and*

$$x_1 = \underset{x\in\Delta_{[n]}}{\arg\max}\,\langle x, g_1\rangle - H(x),$$

$$x_2 = \underset{x\in\Delta_{[n]}}{\arg\max}\,\langle x, g_2\rangle - H(x),$$

*where $H(x) := \sum_{i=1}^n x_i\log x_i$ is the standard entropy functional. Then $\|x_1 - x_2\|_1 \le 2\|g_1 - g_2\|_\infty$.*

*Proof.* Since $H$ is 1-strongly convex in $\|\cdot\|_1$ (Pinsker's inequality), we have that

$$
\begin{aligned}
\frac{\|x_1 - x_2\|_1^2}{2} &\le H(x_1) - H(x_2) - \langle\nabla H(x_2), x_1 - x_2\rangle \\
&= H(x_1) + (\langle x_2, g_1\rangle - H(x_2)) - \langle x_2, g_1\rangle - \langle\nabla H(x_2), x_1 - x_2\rangle \\
&\le H(x_1) + (\langle x_1, g_1\rangle - H(x_1)) - \langle x_2, g_1\rangle - \langle\nabla H(x_2), x_1 - x_2\rangle \\
&= \langle x_1 - x_2, g_1\rangle - \langle\nabla H(x_2), x_1 - x_2\rangle \\
&= \langle x_1 - x_2, g_1 - g_2\rangle + \langle g_2 - \nabla H(x_2), x_1 - x_2\rangle \\
&\le \langle x_1 - x_2, g_1 - g_2\rangle \quad \text{(From the optimality of } x_2) \\
&\le \|x_1 - x_2\|_1\|g_1 - g_2\|_\infty,
\end{aligned}
$$

which completes the proof.

$\square$

# B  Bound for regret minimization algorithms

## B.1  Projected gradient descent

---
**Algorithm 2** Projected gradient descent

---
**Require:** Learning rate $\eta > 0$.
 1: **Initialize** $x_1 \in \mathcal{X} \subset \mathbb{R}^d$.
 2: **for** $t = 1, \ldots, T$ **do**
 3:    Receive loss $g_t \in \mathbb{R}^d$.
 4:    Compute update $y_{t+1} = x_t - \eta g_t$, $x_{t+1} = \mathcal{P}_{\mathcal{X}}(y_{t+1})$.

---

The following weighted regret bound for projected gradient descent is standard. For completeness we provide a proof here. For simplicity of notation, denote the diameter of $\mathcal{X} \subset \mathbb{R}^d$ by $R$ and $G = \max_{t\in[T]}\|g_t\|_2$.

**Lemma B.1** (Weighted regret bound for projected gradient descent). *For any weights $\{w_t\}_{t\ge 1} \in \mathbb{R}_{>0}$ with $w_t \le w_{t+1}$ for all $t \ge 1$, Algorithm 2 achieves*

$$\max_{z\in\mathcal{X}}\sum_{t=1}^T w_t\langle x_t - z, g_t\rangle \le \frac{w_T}{2\eta}R^2 + \frac{\eta\sum_{t=1}^T w_t\cdot G^2}{2}.$$

*Proof.* By following the standard GD analysis, we first have

$$
\begin{aligned}
\langle x_t - z, g_t \rangle =& \frac{1}{\eta} \langle x_t - z, x_t - y_{t+1} \rangle \\
=& \frac{1}{2\eta} \left[ \|x_t - z\|^2 + \|x_t - y_{t+1}\|^2 - \|z - y_{t+1}\|^2 \right] \\
\leq& \frac{1}{2\eta} \left[ \|x_t - z\|^2 + \|x_t - y_{t+1}\|^2 - \|z - x_{t+1}\|^2 \right] \\
=& \frac{1}{2\eta} \left[ \|x_t - z\|^2 - \|z - x_{t+1}\|^2 + \eta^2 \|g_t\|^2 \right],
\end{aligned}
$$

where the inequality follows from $z \in \mathcal{X}$ and $x_{t+1} = \mathcal{P}_{\mathcal{X}}(y_{t+1})$. By multiplying both sides with $w_t$ and taking summation over $t \in [T]$, we have

$$
\begin{aligned}
\sum_{t=1}^{T} w_t \langle x_t - z, g_t \rangle \leq& \frac{1}{2\eta} \sum_{t=1}^{T} w_t \left[ \|x_t - z\|^2 - \|z - x_{t+1}\|^2 + \eta^2 \|g_t\|^2 \right] \\
=& \frac{1}{2\eta} \sum_{t=1}^{T-1} (w_{t+1} - w_t) \|z - x_{t+1}\|^2 + w_1 \|x_1 - z\|^2 + \frac{\eta \sum_{t=1}^{T} w_t \cdot G^2}{2} \\
\overset{(i)}{\leq}& \frac{1}{2\eta} \sum_{t=1}^{T-1} (w_{t+1} - w_t) R^2 + w_1 R^2 + \frac{\eta G^2}{2} = \frac{1}{2\eta} w_T R^2 + \frac{\eta \sum_{t=1}^{T} w_t \cdot G^2}{2},
\end{aligned}
$$

Above, (i) follows as $w_{t+1} \geq w_t$. This completes the proof. $\square$

## B.2 Follow-The-Regularized Leader (FTRL)

In this subsection, we consider the following weighted FTRL algorithm over the probability simplex $\Delta_{[A]}$ with the standard (negative) entropy regularizer $\Phi(x) := \sum_{a \in [A]} x(a) \log x(a)$. Below the notation $x_t(a)$ denotes the $a$-th entry of $x_t$.

---

**Algorithm 3** Weighted FTRL with changing learning rate

---

**Require:** Learning rate $\eta > 0$; Weights $\{w_t\}_{t \geq 1} \subset \mathbb{R}_{>0}$.
 1: **Initialize** $x_1 \leftarrow \mathbf{1}_A / A$ to be the uniform distribution over $[A]$.
 2: **for** $t = 1, \dots, T$ **do**
 3:     Receive loss $g_t \in \mathbb{R}^A$.
 4:     Compute FTRL update

$$
x_{t+1} \leftarrow \arg\min_{x \in \Delta_{[A]}} \left\langle x, \sum_{s=1}^{t} w_s g_s \right\rangle + \frac{w_t}{\eta} \Phi(x). \tag{18}
$$

---

Note that (18) has a closed-form solution via exponential weights:

$$
x_{t+1}(a) \propto_a \exp \left( -\frac{\eta}{w_t} \sum_{s=1}^{t} w_s g_s(a) \right). \tag{19}
$$

**Lemma B.2** (Regret bound of weighted FTRL). *Suppose the weights are non-decreasing: $w_{t+1} \geq w_t$ for all $t \geq 1$, and $\max_t \|g_t\|_\infty \leq G$. Then Algorithm 3 achieves weighted regret bound*

$$
\max_{z \in \Delta_{[A]}} \sum_{t=1}^{T} w_t \langle x_t - z, g_t \rangle \leq \frac{w_T}{\eta} \log A + \frac{\eta G^2}{2} \sum_{t=1}^{T} w_t.
$$

*Proof.* Applying standard anytime FTRL analysis (see, e.g., Excercise 28.12 in [28]) with loss sequence $\{w_t g_t\}_{t \geq 1}$ and learning rate $\{\eta / w_t\}_{t \geq 1}$, we have that

$$
\sum_{t=1}^{T} w_t \langle x_t - z, g_t \rangle \leq \frac{w_T (\Phi(x_1) - \min_x \Phi(x))}{\eta} + \sum_{t=1}^{T} w_t \left( \langle x_t - x_{t+1}, g_t \rangle - \frac{\mathrm{KL}(x_{t+1} \| x_t)}{\eta} \right)
$$

$$\leq \frac{w_T \log A}{\eta} + \sum_{t=1}^{T} \left( w_t \langle x_t - x_{t+1}, g_t \rangle - \frac{w_t \|x_t - x_{t+1}\|_1^2}{2\eta} \right) \quad \text{(Pinsker's inequality)}$$

$$\leq \frac{w_T \log A}{\eta} + \sum_{t=1}^{T} \frac{w_t \eta}{2} \|g_t\|_\infty^2$$

$$\leq \frac{w_T \log A}{\eta} + \frac{\eta G^2}{2} \sum_{t=1}^{T} w_t,$$

which completes the proof. $\qquad \square$

## B.3 Optimistic Follow-The-Regularized-Leader (OFTRL)

We consider the following OFTRL algorithm on the probability simplex $\Delta_{[A]}$ with standard (negative) entropy regularizer $\Phi(x) = \sum_{a \in [A]} x(a) \log x(a)$.

---

**Algorithm 4** Anytime OFTRL

---

**Require:** Learning rate $\{\eta_t\}_{t \geq 1}$.
1: **Initialize** $x_1 \leftarrow \mathbf{1}_A / A$ to be the uniform distribution over $[A]$.
2: **for** $t = 1, \dots, T$ **do**
3:      Receive loss $g_t \in \mathbb{R}^A$.
4:      Compute a prediction vector $M_{t+1} \in \mathbb{R}^A$ using past observations.
5:      Compute OFTRL update

$$x_{t+1} \leftarrow \underset{x \in \Delta_{[A]}}{\arg\min} \, \eta_{t+1} \left\langle x, \sum_{s=1}^{t} g_s + M_{t+1} \right\rangle + \Phi(x). \tag{20}$$

---

Note that (20) has a closed-form solution via exponential weights:

$$x_{t+1}(a) \propto_a \exp\left( -\eta_{t+1} \left[ \sum_{s=1}^{t} g_s(a) + M_{t+1}(a) \right] \right). \tag{21}$$

The following regret bound for OFTRL follows similarly as standard OFTRL analysis, see, e.g. [41, Lemma 1]. For completeness, we provide a proof here.

**Lemma B.3** (Regret bound for OFTRL). *Suppose the learning rates are non-increasing: $\eta_t \geq \eta_{t+1}$ for all $t \geq 1$. Then Algorithm 4 achieves the following bound for all $x \in \mathcal{X}$:*

$$\sum_{t=1}^{T} \langle x_t - x, g_t \rangle \leq \frac{\log A}{\eta_T} + \sum_{t=1}^{T} \eta_t \|g_t - M_t\|_\infty^2 - \sum_{t=1}^{T-1} \frac{1}{8\eta_t} \|x_t - x_{t+1}\|_1^2.$$

*Proof.* Consider a fixed $T \geq 1$. Note that Algorithm 4 is equivalent to Algorithm (25) with regularizer $R_t(\cdot) := (\Phi(\cdot) + \log A)/\eta_{t+1} \geq 0$ for $t \geq 0$, and $R_T(\cdot) := R_{T-1}(\cdot)$ (Note that the shifting by $\log A$ does not affect the algorithm.)

We first decompose the regret into the following three terms

$$\sum_{t=1}^{T} \langle x_t - x, g_t \rangle = \sum_{t=1}^{T} \langle q_{t+1} - x, g_t \rangle + \sum_{t=1}^{T} \langle x_t - q_{t+1}, M_t \rangle + \sum_{t=1}^{T} \langle x_t - q_{t+1}, g_t - M_t \rangle,$$

where $\{q_t\}_{t \geq 1}$ is defined in (26). By Lemma B.5, we can upper bound the first two terms by $R_T(x) - \min_{x'} R_0(x') + S_T \leq R_T(x) + S_T$ and obtain

$$\sum_{t=1}^{T} \langle x_t - x, g_t \rangle$$

$$\leq R_T(x) + \sum_{t=1}^{T} \left( R_{t-1}(q_{t+1}) - R_t(q_{t+1}) \right) + \sum_{t=1}^{T} \left( \langle x_t - q_{t+1}, g_t - M_t \rangle - D_{R_{t-1}}(q_{t+1}, x_t) - D_{R_{t-1}}(x_t, q_t) \right)$$

$$\leq R_T(x) + \sum_{t=1}^{T} \left( \|x_t - q_{t+1}\|_1 \|g_t - M_t\|_\infty - \frac{1}{2\eta_t} \|q_{t+1} - x_t\|_1^2 - \frac{1}{2\eta_t} \|x_t - q_t\|_1^2 \right),$$

where the second inequality uses $R_{t-1} \leq R_t$ and $R_{t-1}$ is $1/\eta_t$ strongly-convex w.r.t. $\|\cdot\|_1$. Finally, we conclude the proof by applying Cauchy-Schwarz inequality:

$$\|x_t - q_{t+1}\|_1 \|g_t - M_t\|_\infty - \frac{1}{4\eta_t} \|q_{t+1} - x_t\|_1^2 \leq \eta_t \|g_t - M_t\|_\infty^2,$$

and triangle inequality

$$-\frac{1}{4\eta_t} \|q_{t+1} - x_t\|_1^2 - \frac{1}{4\eta_t} \|q_{t+1} - x_{t+1}\|_1^2 \leq -\frac{1}{8\eta_t} \|x_{t+1} - x_t\|_1^2,$$

and the bound $R_T(x) \leq \log A / \eta_{T+1} = \log A / \eta_T$ for any $x \in \Delta_{[A]}$. $\qquad\square$

The following lemma bounds the total variation of the iterates in terms of the smoothness of loss vectors and prediction vectors. This can be seen as a generalization of [9, Lemma 3.2] to the case with changing learning rate and arbitrary prediction vectors.

**Lemma B.4** (Bounding stability by the smoothness of loss). *Suppose the learning rates are non-increasing: $\eta_t \geq \eta_{t+1}$ for all $t \geq 1$. Then the OFTRL algorithm* (20) *satisfies (understanding $M_1 := 0$)*

$$\sum_{t=2}^{T} \frac{1}{2\eta_t} \|x_t - x_{t-1}\|_1^2 \leq \frac{\log A}{\eta_T} + \max_{x \in \Delta_{[d]}} \sum_{t=1}^{T-1} \langle x_t - x, g_t \rangle + \sum_{t=2}^{T} \|M_t - M_{t-1}\|_\infty + \|M_T\|_\infty \tag{22}$$

$$\leq \frac{2\log A}{\eta_T} + \sum_{t=1}^{T-1} \eta_t \|g_t - M_t\|_\infty^2 + \sum_{t=2}^{T} \|M_t - M_{t-1}\|_\infty + \|M_T\|_\infty. \tag{23}$$

*In particular, choosing the prediction vector $M_t = g_{t-1}$ with $g_0 := 0$, and assume $\|g_t - g_{t-1}\|_\infty \leq G_t$ for all $t \geq 1$, we have*

$$\sum_{t=2}^{T} \frac{1}{2\eta_t} \|x_t - x_{t-1}\|_1^2 \leq \frac{2\log A}{\eta_T} + \sum_{t=1}^{T-1} \eta_t \|g_t - g_{t-1}\|_\infty^2 + \sum_{t=2}^{T} \|g_{t-1} - g_{t-2}\|_\infty + \|g_{T-1}\|_\infty$$

$$\leq \frac{2\log A}{\eta_T} + \sum_{t=1}^{T-1} (1 + \eta_t G_t) \|g_t - g_{t-1}\|_\infty + \|g_{T-1}\|_\infty. \tag{24}$$

*Proof.* We first prove (22). For any $t \geq 2$, the optimality condition of (20) for $x_t$ gives

$$\left\langle \sum_{s=1}^{t-1} g_s + M_t + \frac{\nabla \Phi(x_t)}{\eta_t}, x' - x_t \right\rangle \geq 0$$

for all $x' \in \Delta_{[A]}$. In particular, this holds for $x' = x_{t-1}$, from which we get

$$\frac{1}{2\eta_t} \|x_{t-1} - x_t\|_1^2 \overset{(i)}{\leq} \frac{1}{\eta_t} \mathrm{KL}(x_{t-1} \| x_t) \overset{(ii)}{=} \frac{\Phi(x_{t-1}) - \Phi(x_t)}{\eta_t} - \left\langle \frac{\nabla \Phi(x_t)}{\eta_t}, x_{t-1} - x_t \right\rangle$$

$$\leq \frac{\Phi(x_{t-1}) - \Phi(x_t)}{\eta_t} + \left\langle \sum_{s=1}^{t-1} g_s + M_t, x_{t-1} - x_t \right\rangle,$$

where (i) is by Pinsker's inequality, and (ii) is since the KL divergence is the Bregman divergence of $\Phi$. Summing the above over $t = 2, \ldots, T$ yields

$$\sum_{t=2}^{T} \frac{1}{2\eta_t} \|x_{t-1} - x_t\|_1^2$$

$$\leq -\frac{\Phi(x_T)}{\eta_T} + \sum_{t=2}^{T} \underbrace{\left( \frac{1}{\eta_t} - \frac{1}{\eta_{t-1}} \right)}_{\geq 0} \underbrace{\Phi(x_{t-1})}_{\leq 0} - \left\langle \sum_{s=1}^{T-1} g_s + M_T, x_T \right\rangle + \sum_{t=2}^{T} \langle g_{t-1} + M_t - M_{t-1}, x_{t-1} \rangle$$

$$\underbrace{}_{\leq \log A/\eta_T}$$

$$\leq \frac{\log A}{\eta_T} + \sum_{t=1}^{T-1} \langle x_t, g_t \rangle - \sum_{t=1}^{T-1} \langle x_T, g_t \rangle + \langle M_T, x_T \rangle + \sum_{t=2}^{T} \langle M_t - M_{t-1}, x_{t-1} \rangle$$

$$\leq \frac{\log A}{\eta_T} + \max_{x \in \Delta_{[d]}} \sum_{t=1}^{T-1} \langle x_t - x, g_t \rangle + \sum_{t=2}^{T} \|M_t - M_{t-1}\|_\infty + \|M_T\|_\infty .$$

This proves (22). Then, (23) follows by plugging in the regret bound given by Lemma B.3:

$$\max_{x \in \Delta_{[d]}} \sum_{t=1}^{T-1} \langle x_t - x, g_t \rangle \leq \frac{\log A}{\eta_{T-1}} + \sum_{t=1}^{T-1} \eta_t \|g_t - M_t\|_\infty^2 \leq \frac{\log A}{\eta_T} + \sum_{t=1}^{T-1} \eta_t \|g_t - M_t\|_\infty^2 .$$

Finally, (24) is a direct consequence of (23) by plugging in $M_t = g_{t-1}$ and $\|g_t - g_{t-1}\|_\infty \leq G_t$ for all $t \geq 1$. $\qquad\square$

### B.3.1 Auxiliary lemma for OFTRL with general regularizers

Consider an OFTRL algorithm with loss function $\{g_t\}_{t \geq 0} \subset \mathbb{R}^d$, parameter space $\mathcal{X} \subset \mathbb{R}^d$, and convex regularizers $R_t : \mathcal{X} \to \mathbb{R}$ for $t \geq 0$:

$$x_{t+1} \leftarrow \arg\min_{x \in \mathcal{X}} \left\langle x, \sum_{s=1}^{t} g_s + M_{t+1} \right\rangle + R_t(x). \tag{25}$$

Define auxiliary sequence

$$q_{t+1} = \arg\min_{x \in \mathcal{X}} \left\langle x, \sum_{s=1}^{t} g_s \right\rangle + R_t(x). \tag{26}$$

Recall the Bregman divergence associated with any convex regularizer $R : \mathcal{X} \to \mathbb{R}$ is given by

$$D_R(x,y) := R(x) - R(y) - \langle \nabla R(y), x - y \rangle \geq 0.$$

**Lemma B.5** (Auxiliary lemma for OFTRL with general regularizers). *Algorithm* (25) *achieves the following for any $T \geq 1$ and $x \in \mathcal{X}$:*

$$\sum_{t=1}^{T} \langle q_{t+1}, g_t \rangle + \sum_{t=1}^{T} \langle x_t - q_{t+1}, M_t \rangle \leq \sum_{t=1}^{T} \langle x, g_t \rangle + R_T(x) - \min_{x' \in \mathcal{X}} R_0(x') + S_T,$$

*where*

$$S_T := \sum_{t=1}^{T} \left( R_{t-1}(q_{t+1}) - R_t(q_{t+1}) \right) - \sum_{t=1}^{T} \left( D_{R_{t-1}}(q_{t+1}, x_t) + D_{R_{t-1}}(x_t, q_t) \right).$$

*Proof.* We prove the lemma by induction. The above relation holds trivially for $T = 0$. Assume the relation holds for $\tau = T - 1$. For $\tau = T$, we have

$$\sum_{t=1}^{T} \langle q_{t+1}, g_t \rangle + \sum_{t=1}^{T} \langle x_t - q_{t+1}, M_t \rangle$$

$$\leq \min_{x \in \mathcal{X}} \left[ \sum_{t=1}^{T-1} \langle x, g_t \rangle + R_{T-1}(x) \right] - \min_{x' \in \mathcal{X}} R_0(x') + S_{T-1} + \langle q_{T+1}, g_T \rangle + \langle x_T - q_{T+1}, M_T \rangle$$

$$= \sum_{t=1}^{T-1} \langle q_T, g_t \rangle + R_{T-1}(q_T) - \min_{x' \in \mathcal{X}} R_0(x')$$

$$+ S_{T-1} + \langle q_{T+1}, g_T \rangle + \langle x_T - q_{T+1}, M_T \rangle \quad \text{(definition of } q_T\text{)}$$

$$\leq \sum_{t=1}^{T-1} \langle x_T, g_t \rangle + R_{T-1}(x_T) - D_{R_{T-1}}(x_T, q_T) - \min_{x' \in \mathcal{X}} R_0(x')$$

$$+ S_{T-1} + \langle q_{T+1}, g_T \rangle + \langle x_T - q_{T+1}, M_T \rangle \quad \text{(optimality of } q_T\text{)}$$

$$= \min_{x \in \mathcal{X}} \left[ \langle x, \sum_{t=1}^{T-1} g_t + M_T \rangle + R_{T-1}(x) \right] - D_{R_{T-1}}(x_T, q_T) - \min_{x' \in \mathcal{X}} R_0(x')$$

$$+ S_{T-1} + \langle q_{T+1}, g_T - M_T \rangle \quad \text{(definition of } x_T\text{)}$$

$$\leq \langle q_{T+1}, \sum_{t=1}^{T-1} g_t + M_T \rangle + R_{T-1}(q_{T+1}) - D_{R_{T-1}}(q_{T+1}, x_T)$$

$$- D_{R_{T-1}}(x_T, q_T) - \min_{x' \in \mathcal{X}} R_0(x') + S_{T-1} + \langle q_{T+1}, g_T - M_T \rangle \quad \text{(optimality of } x_T\text{)}$$

$$= \min_{x \in \mathcal{X}} \langle x, \sum_{t=1}^{T} g_t \rangle + R_T(x) + (R_{T-1}(q_{T+1}) - R_T(q_{T+1}))$$

$$- D_{R_{T-1}}(q_{T+1}, x_T) - D_{R_{T-1}}(x_T, q_T) - \min_{x' \in \mathcal{X}} R_0(x') + S_{T-1} \quad \text{(definition of } q_{T+1}\text{)},$$

which completes the induction. $\qquad\square$

## C   Proofs for Section 3.1

In this section we prove Theorem 2. The proof relies on the following two lemmas.

**Lemma C.1** (Performance difference for Markov policies)**.** *In two-player zero-sum Markov games, suppose a Markov policy $(\mu, \nu)$ satisfies the following for all $h \in [H+1]$:*

$$\max_s \max_{\mu^\dagger \in \Delta_\mathcal{A}} \left( \left[ (\mu^\dagger)^\top Q_h^\star \nu_h \right](s) - V_h^\star(s) \right) \leq \varepsilon_h,$$

$$\max_s \max_{\nu^\dagger \in \Delta_\mathcal{B}} \left( V_h^\star(s) - \left[ \mu_h^\top Q_h^\star \nu^\dagger \right](s) \right) \leq \varepsilon_h.$$

*Then we have for all $h \in [H]$ that*

$$\max \left\{ \| V_h^{\dagger,\nu} - V_h^\star \|_\infty, \| V_h^{\mu,\dagger} - V_h^\star \|_\infty \right\} \leq \sum_{h'=h}^{H} \varepsilon_{h'}.$$

*Proof.* We prove by backward induction over $h$. The claim is trivial for $h = H + 1$. Suppose the claim holds for step $h + 1$. At step $h$,

$$\| V_h^{\dagger,\nu} - V_h^\star \|_\infty = \max_s \left| \max_{\mu^\dagger \in \Delta_\mathcal{A}} \left[ (\mu^\dagger)^\top Q_h^{\dagger,\nu} \nu_h \right](s) - V_h^\star(s) \right|$$

$$\leq \max_s \left| \max_{\mu^\dagger \in \Delta_\mathcal{A}} \left[ (\mu^\dagger)^\top Q_h^\star \nu_h \right](s) - V_h^\star(s) \right| + \| Q_h^{\dagger,\nu} - Q_h^\star \|_\infty$$

$$\leq \varepsilon_h + \| Q_h^{\dagger,\nu} - Q_h^\star \|_\infty.$$

Notice that

$$\| Q_h^{\dagger,\nu} - Q_h^\star \|_\infty \leq \max_{s,a,b} \left| \left( r_h + \mathbb{P}_h V_{h+1}^{\dagger,\nu} \right)(s,a,b) - \left( r_h + \mathbb{P}_h V_{h+1}^\star \right)(s,a,b) \right|$$

$$\leq \max_{s,a,b} \left| \mathbb{P}_h \left[ V_{h+1}^{\dagger,\nu} - V_{h+1}^\star \right](s,a,b) \right| \leq \| V_{h+1}^{\dagger,\nu} - V_{h+1}^\star \|_\infty \leq \sum_{h'=h+1}^{H} \varepsilon_{h'}. \quad \text{(by inductive hypothesis)}$$

Combining the two inequalities we get

$$\| V_h^{\dagger,\nu} - V_h^\star \|_\infty \leq \sum_{h'=h}^{H} \varepsilon_{h'}.$$

This proves the claim for $\|V_h^{\dagger,\nu} - V_h^\star\|_\infty$. The same argument also holds for $\|V_h^{\mu,\dagger} - V_h^\star\|_\infty$, which completes the proof. $\qquad\square$

Throughout the rest of this section, we define the following shorthand for the value estimation error:

$$\delta_h^t := \left\|Q_h^t - Q_h^\star\right\|_\infty = \max_{s,a,b} \left|Q_h^t(s,a,b) - Q_h^\star(s,a,b)\right|,$$

where $Q_h^t$ is the estimated value in Algorithm 1.

**Lemma C.2** (Recursion of value estimation)**.** *Algorithm 1 guarantees that for all $(t,h) \in [T] \times [H]$,*

$$\delta_h^t \leq \sum_{i=1}^t \beta_t^i \delta_{h+1}^i + \mathrm{reg}_{h+1}^t.$$

*Further, suppose that $\mathrm{reg}_h^t \leq \overline{\mathrm{reg}}_h^t$ for all $(h,t) \in [H] \times [T]$, where $\overline{\mathrm{reg}}_h^t$ is non-increasing in t: $\overline{\mathrm{reg}}_h^t \geq \overline{\mathrm{reg}}_h^{t+1}$ for all $t \geq 1$. Then we have*

$$\delta_h^t \leq H c_\beta^{H-1} \cdot \frac{1}{t} \sum_{i=1}^t \max_{h'} \overline{\mathrm{reg}}_{h'}^i,$$

*where $c_\beta$ is defined in* (9).

*Proof.* Fix $(h,s,a,b) \in [H] \times \mathcal{S} \times \mathcal{A} \times \mathcal{B}$. From the definition of $Q_h^\star$ we have that

$$
\begin{aligned}
Q_h^\star(s,a,b) &= r_h(s,a,b) + \max_{\mu_{h+1}} \min_{\nu_{h+1}} \mathbb{P}_h\left[\mu_{h+1}^\top Q_{h+1}^\star \nu_{h+1}\right](s,a,b) \\
&\leq r_h(s,a,b) + \max_{\mu_{h+1}} \mathbb{P}_h\left[\mu_{h+1}^\top Q_{h+1}^\star \left(\sum_{i=1}^t \beta_t^i \nu_{h+1}^i\right)\right](s,a,b) \\
&= r_h(s,a,b) + \max_{\mu_{h+1}} \sum_{i=1}^t \beta_t^i \mathbb{P}_h\left[\mu_{h+1}^\top Q_{h+1}^\star \nu_{h+1}^i\right](s,a,b) \\
&\leq r_h(s,a,b) + \max_{\mu_{h+1}} \sum_{i=1}^t \beta_t^i \left(\mathbb{P}_h\left[\mu_{h+1}^\top Q_{h+1}^i \nu_{h+1}^i\right](s,a,b) + \|Q_{h+1}^i - Q_{h+1}^\star\|_\infty\right) \\
&\leq r_h(s,a,b) + \sum_{i=1}^t \beta_t^i \mathbb{P}_h\left[(\mu_{h+1}^i)^\top Q_{h+1}^i \nu_{h+1}^i\right](s,a,b) + \sum_{i=1}^t \beta_t^i \delta_{h+1}^i + \mathrm{reg}_{h+1}^t \\
&= Q_h^t(s,a,b) + \sum_{i=1}^t \beta_t^i \delta_{h+1}^i + \mathrm{reg}_{h+1}^t.
\end{aligned}
$$

Above, the last equality is derived from the update rule (3), which implies that

$$Q_h^t(s,a,b) = \sum_{i=1}^t \beta_t^i \left(r_h + \mathbb{P}_h\left[(\mu_{h+1}^i)^\top Q_{h+1}^i \nu_{h+1}^i\right]\right)(s,a,b).$$

Therefore we have

$$Q_h^\star(s,a,b) - Q_h^t(s,a,b) \leq \sum_{i=1}^t \beta_t^i \delta_{h+1}^i + \mathrm{reg}_{h+1}^t, \quad \forall\, s,a,b.$$

Apply similar analysis to the min-player, we get

$$Q_h^t(s,a,b) - Q_h^\star(s,a,b) \leq \sum_{i=1}^t \beta_t^i \delta_{h+1}^i + \mathrm{reg}_{h+1}^t, \quad \forall\, s,a,b.$$

Thus we get

$$\delta_h^t \leq \sum_{i=1}^t \beta_t^i \delta_{h+1}^i + \mathrm{reg}_{h+1}^t,$$

which completes the proof of the first inequality in the Lemma. Now consider an auxiliary sequence $\{\Delta_h^t\}_{h,t}$ defined by

$$\begin{cases} \Delta_h^t = \sum_{i=1}^t \beta_t^i \Delta_{h+1}^i + \overline{\text{reg}}_{h+1}^t, \\ \Delta_{H+1}^t = 0, \quad \text{for all } t. \end{cases} \tag{27}$$

Where $\overline{\text{reg}}_h^t$ is the upperbound of $\text{reg}_h^t$ defined in Theorem 2. Observe that $\{\Delta_h^t\}_{h,t}$ satisfies the following properties

$$\begin{cases} \Delta_h^t \geq \delta_h^t & \text{(by definition)}, \\ \Delta_h^t \leq \Delta_h^{t-1} & \text{(by Lemma A.1)}. \end{cases} \tag{28}$$

Therefore, to control $\delta_h^t$, it suffices to bound $\Delta_h^t \leq \frac{1}{t} \sum_{i=1}^t \Delta_h^i$, which follows from the standard argument in [23]:

$$\begin{aligned}
\frac{1}{t} \sum_{i=1}^t \Delta_h^i &= \frac{1}{t} \sum_{i=1}^t \sum_{j=1}^i \beta_i^j \Delta_{h+1}^j + \frac{1}{t} \sum_{i=1}^t \overline{\text{reg}}_{h+1}^i \\
&\leq \frac{1}{t} \sum_{j=1}^t \left( \sum_{i=j}^t \beta_i^j \right) \Delta_{h+1}^j + \frac{1}{t} \sum_{i=1}^t \overline{\text{reg}}_{h+1}^i \\
&\leq c_\beta \cdot \frac{1}{t} \sum_{i=1}^t \Delta_{h+1}^i + \frac{1}{t} \sum_{i=1}^t \overline{\text{reg}}_{h+1}^i \\
&\leq c_\beta^2 \cdot \frac{1}{t} \sum_{i=1}^t \Delta_{h+2}^i + c_\beta \cdot \frac{1}{t} \sum_{i=1}^t \overline{\text{reg}}_{h+2}^i + \frac{1}{t} \sum_{i=1}^t \overline{\text{reg}}_{h+1}^i \\
&\leq \cdots \\
&\leq \left( \sum_{h'=h+1}^H c_\beta^{h'-h} \right) \cdot \frac{1}{t} \sum_{i=1}^t \max_{1 \leq h' \leq H} \overline{\text{reg}}_{h'}^i \\
&\leq H c_\beta^{H-1} \cdot \frac{1}{t} \sum_{i=1}^t \max_{1 \leq h' \leq H} \overline{\text{reg}}_{h'}^i.
\end{aligned}$$

Above, the last step used the fact that $c_\beta \geq 1$. This completes the proof of the second inequality in the Lemma. □

We are now ready to prove the main theorem.

*Proof of Theorem 2.* Fix any $(h,s) \in [H] \times \mathcal{S}$. We first give a bound for $\max_{\mu^\dagger \in \Delta_\mathcal{A}, \nu^\dagger \in \Delta_\mathcal{B}} \left[ (\mu^\dagger)^\top Q_h^\star \widehat{\nu}_h^T - (\widehat{\mu}_h^T)^\top Q_h^\star \nu^\dagger \right](s)$, i.e. the per-state duality gap of $(\widehat{\mu}^T, \widehat{\nu}^T)$ with respect to $Q_h^\star$. We have

$$\begin{aligned}
&\max_{\mu^\dagger \in \Delta_\mathcal{A}, \nu^\dagger \in \Delta_\mathcal{B}} \left[ (\mu^\dagger)^\top Q_h^\star \widehat{\nu}_h^T - (\widehat{\mu}_h^T)^\top Q_h^\star \nu^\dagger \right](s) \\
&= \max_{\mu^\dagger \in \Delta_\mathcal{A}, \nu^\dagger \in \Delta_\mathcal{B}} \sum_{t=1}^T \beta_T^t \left[ (\mu^\dagger)^\top Q_h^\star \nu_h^t - (\mu_h^t)^\top Q_h^\star \nu^\dagger \right](s) \\
&\leq \underbrace{\max_{\mu^\dagger \in \Delta_\mathcal{A}, \nu^\dagger \in \Delta_\mathcal{B}} \sum_{t=1}^T \beta_T^t \left[ (\mu^\dagger)^\top Q_h^t \nu_h^t - (\mu_h^t)^\top Q_h^t \nu^\dagger \right](s)}_{\text{reg}_{\mu,h}^T(s) + \text{reg}_{\nu,h}^T(s)} + 2 \sum_{t=1}^T \beta_T^t \delta_h^t \\
&\leq 2 \overline{\text{reg}}_h^T + 2 H c_\beta^{H-1} \sum_{t=1}^T \beta_T^t \cdot \frac{1}{t} \sum_{i=1}^t \max_{h'} \overline{\text{reg}}_{h'}^i \quad \text{(Lemma C.2)} \\
&\leq 2 \overline{\text{reg}}_h^T + 2 H c_\beta^{H-1} \left( \sum_{t=1}^T \frac{1}{t} \beta_T^t \right) \left( \sum_{i=1}^T \max_{h'} \overline{\text{reg}}_{h'}^i \right).
\end{aligned}$$

Apply Lemma A.3 into the above inequality, we get that

$$\max_{\mu^{\dagger}\in\Delta_{\mathcal{A}},\nu^{\dagger}\in\Delta_{\mathcal{B}}}\left[(\mu^{\dagger})^{\top}Q_h^{\star}\widehat{\nu}_h^T-\left(\widehat{\mu}_h^T\right)^{\top}Q_h^{\star}\nu^{\dagger}\right](s)\le 2\overline{\text{reg}}_h^T+4Hc_{\beta}^H\log T\cdot\frac{1}{T}\sum_{t=1}^{T}\max_{h'}\overline{\text{reg}}_{h'}^t.$$

Since

$$\max_{\mu^{\dagger}\in\Delta_{\mathcal{A}}}\left([(\mu^{\dagger})^{\top}Q_h^{\star}\widehat{\nu}_h^T](s)-V_h^{\star}(s)\right),\ \max_{\nu^{\dagger}\in\Delta_{\mathcal{B}}}\left(V_h^{\star}(s)-[(\widehat{\mu}_h^T)^{\top}Q_h^{\star}\nu^{\dagger}](s)\right)\le\max_{\mu^{\dagger}\in\Delta_{\mathcal{A}},\nu^{\dagger}\in\Delta_{\mathcal{B}}}\left[(\mu^{\dagger})^{\top}Q_h^{\star}\widehat{\nu}_h^T-(\widehat{\mu}_h^T)^{\top}Q_h^{\star}\nu^{\dagger}\right](s),$$

by applying Lemma C.1 and the preceding per-state duality gap bound, we have

$$\text{NEGap}(\widehat{\mu}^T,\widehat{\nu}^T)=\left(V_1^{\dagger,\widehat{\nu}^T}(s_1)-V_1^{\star}(s_1)\right)+\left(V_1^{\star}(s_1)-V_1^{\widehat{\mu}^T,\dagger}(s_1)\right)$$

$$\le 2\sum_{h=1}^{H}\max_{s}\max_{\mu^{\dagger}\in\Delta_{\mathcal{A}},\nu^{\dagger}\in\Delta_{\mathcal{B}}}\left[(\mu^{\dagger})^{\top}Q_h^{\star}\widehat{\nu}_h^T-\left(\widehat{\mu}_h^T\right)^{\top}Q_h^{\star}\nu^{\dagger}\right](s)$$

$$\le 2\sum_{h=1}^{H}\left(2\overline{\text{reg}}_h^T+4Hc_{\beta}^H\log T\cdot\frac{1}{T}\sum_{t=1}^{T}\max_{h'}\overline{\text{reg}}_{h'}^t\right)$$

$$\le 4H\max_{h}\overline{\text{reg}}_h^T+8H^2c_{\beta}^H\log T\cdot\frac{1}{T}\sum_{t=1}^{T}\max_{h'}\overline{\text{reg}}_{h'}^t.$$

This completes the proof. $\qquad\square$

### C.1 Implementation of state-wise average policy

Here we explain how to implement the state-wise average policy (5) efficiently via moving average. Recall that the weights $\{\beta_T^t\}_{t=1}^{T}$ are defined via $\{\beta_t\}_{t\ge 1}$ through (6). Therefore, for each $(h,s)$, we can maintain a set of moving averages

$$\overline{\mu}_h^t(\cdot|s):=(1-\beta_t)\overline{\mu}_h^{t-1}(\cdot|s)+\beta_t\mu_h^t(\cdot|s)$$

for all $t\ge 1$ during the execution of Algorithm 1. At time $T$, we output the moving average $\overline{\mu}_h^T(\cdot|s)$, which is exactly the desired state-wise average policy in (5):

$$\overline{\mu}_h^T(\cdot|s)=\sum_{t=1}^{T}\beta_T^t\mu_h^t(\cdot|s)=\widehat{\mu}_h^T(\cdot|s).$$

## D  Algorithm details and proofs for Section 3.2

### D.1 Nash V-Learning (full-information version)

The full description of Nash V-Learning (Example 1) using V updates is presented in Algorithm 5.

**Proposition D.1** (Equivalence between V update and Q update). *Nash V-learning (full-information version) in Algorithm 5 is equivalent to Algorithm 1 with the* MatrixGameAlg *as weighted FTRL* (10).

*Proof.* It suffices to show that, for the Q value defined in (4) and the V value defined in (30), the following holds for all $(h,s,a,b)$ and all $t\in[T]$:

$$Q_h^t(s,a,b)=\left[r_h+\mathbb{P}_hV_{h+1}^t\right](s,a,b). \tag{31}$$

Since $\alpha_1=1$, it is not hard to verify that

$$Q_h^1(s,a,b)=\left[r_h+\mathbb{P}_hV_{h+1}^1\right](s,a,b),$$

We now prove by induction on both $t$ and $h$. Given that $Q_h^1(s,a,b)=\left[r_h+\mathbb{P}_hV_{h+1}^1\right](s,a,b)$, it is not hard to verify that $Q_H^t(s,a,b)=r_H(s,a,b),\ \forall\,k\ge 0$. We assume that (31) holds for $(t-1,h)$ and $(t,h+1)$, then for $(t,h)$, from (3)

$$Q_h^t(s,a,b)=(1-\alpha_t)Q_h^{t-1}(s,a,b)+\alpha_t\left(r_h+\mathbb{P}_h[(\mu_{h+1}^t)^{\top}Q_{h+1}^t\nu_{h+1}^t]\right)(s,a,b)$$

---

**Algorithm 5** Nash V-learning (full-information version)

---

**Require:** Learning rate $\{\alpha_t\}_{t \geq 1}$ in (3) and corresponding $\{w_t\}_{t \geq 1}$ (cf. Example 1); $\eta > 0$.
**Initialize:** Set $V_h^0(s) = H - h + 1$ for all $(h, s) \in [H] \times \mathcal{S}$.
**for** $t = 1, \ldots, T$ **do**
  **for** $h = H, \ldots, 1$ **do**
    Update policy for all $s \in \mathcal{S}$ (understanding $w_0 := 1$):

$$\mu_h^t(a|s) \propto_a \exp\left(\frac{\eta}{w_{t-1}} \sum_{i=1}^{t-1} w_i \big[(r_h + \mathbb{P}_h V_{h+1}^i)\nu_h^i\big](s, a)\right)$$

$$\nu_h^t(a|s) \propto_b \exp\left(-\frac{\eta}{w_{t-1}} \sum_{i=1}^{t-1} w_i \Big[(r_h + \mathbb{P}_h V_{h+1}^i)^\top \mu_h^i\Big](s, b)\right). \tag{29}$$

    Update V value for all $s \in \mathcal{S}$:

$$V_h^t(s) \leftarrow (1 - \alpha_t)V_h^{t-1}(s) + \alpha_t\Big[(\mu_h^t)^\top (r_h + \mathbb{P}_h V_{h+1}^t)\nu_h^t\Big](s). \tag{30}$$

---

$$\leq (1 - \alpha_t)\big[r_h + \mathbb{P}_h V_{h+1}^{t-1}\big](s, a, b) + \alpha_t\big(r_h + \mathbb{P}_h\big[(\mu_{h+1}^t)^\top (r_{h+1} + \mathbb{P}_{h+1} V_{h+2})\nu_{h+1}^t\big]\big)(s, a, b)$$

(inductive hypothesis)

$$= \big[r_h + \mathbb{P}_h\big((1 - \alpha_t)V_{h+1}^{t-1} + \alpha_t(\mu_{h+1}^t)^\top (r_{h+1} + \mathbb{P}_{h+1} V_{h+2})\nu_{h+1}^t\big)\big](s, a, b)$$
$$= \big[r_h + \mathbb{P}_h V_{h+1}^t\big](s, a, b) \text{ (from (30))},$$

which completes the proof by induction. $\qquad\qquad\square$

**Lemma D.1** (Per-state regret bound for Nash V-learning). *Algorithm 5 achieves the following per-state regret bound:*

$$\mathrm{reg}_h^t \leq \frac{(H+1)\log(A \vee B)}{\eta t} + \frac{\eta H^2}{2}, \quad \forall h \in [H], t \geq 1.$$

*Proof.* Fix any $(h, s)$. Note that the update of $\{\mu_h^t(\cdot|s)\}_{t \geq 1}$ in (29) is equivalent to the weighted FTRL algorithm (Algorithm 3) with loss vectors $g_i = -\big[Q_h^i \nu_h^i\big](s)$. Thus by Lemma B.2 we get for any $t \geq 1$ that

$$\max_{\mu^\dagger \in \Delta_{\mathcal{A}}} \sum_{i=1}^t w_i \big\langle \mu^\dagger - \mu_h^i(\cdot|s), \big[Q_h^i \nu_h^i\big](s, \cdot)\big\rangle \leq \frac{w_t}{\eta} \log A + \frac{\eta H^2}{2} \sum_{i=1}^t w_i.$$

Further, recalling $\alpha_t^i = w_i \cdot \alpha_t^1$ for $1 \leq i \leq t$, we have

$$\mathrm{reg}_{h,\mu}^t \leq \alpha_t^1\left(\frac{w_t}{\eta} \log A + \frac{\eta H^2}{2} \sum_{i=1}^t w_i\right) = \frac{\alpha_t^1 w_t}{\eta} \log A + \frac{\eta H^2}{2} \sum_{i=1}^t \alpha_t^i$$

$$= \frac{\alpha_t}{\eta} \log A + \frac{\eta H^2}{2} \leq \frac{(H+1)\log A}{\eta t} + \frac{\eta H^2}{2}.$$

The similar bound also holds for $\mathrm{reg}_{h,\nu}^t$, and thus we have that

$$\mathrm{reg}_h^t \leq \frac{(H+1)\log(A \vee B)}{\eta t} + \frac{\eta H^2}{2}.$$

$\qquad\qquad\square$

**Proposition D.2** (Guarantee of Nash V-Learning). *Algorithm 5 achieves*

$$\mathrm{NEGap}(\widehat{\mu}^T, \widehat{\nu}^T) \leq 14\eta H^4 \log(T) + \frac{104 \log(A \vee B) \log(T)^2 H^3}{\eta T}.$$

*Specifically, choosing $\eta = \frac{4}{\sqrt{HT}}$, we have*

$$\mathrm{NEGap}(\widehat{\mu}^T, \widehat{\nu}^T) \le \frac{82 \log(A \vee B) \log(T)^2 H^{7/2}}{\sqrt{T}}.$$

*Proof.* From Lemma D.1, we can take $\overline{\mathrm{reg}}_h^t$ as $\overline{\mathrm{reg}}_h^t = \frac{(H+1)\log(A \vee B)}{\eta t} + \frac{\eta H^2}{2}$. Then from Theorem 2, we have

$$\mathrm{NEGap}(\widehat{\mu}^T, \widehat{\nu}^T) \le 4H \max_h \overline{\mathrm{reg}}_h^T + 8H^2 \left(1 + \frac{1}{H}\right)^H \log(T) \cdot \frac{1}{T} \sum_{t=1}^T \max_h \overline{\mathrm{reg}}_h^t$$

$$\le 4H \left(\frac{(H+1)\log(A \vee B)}{\eta T} + \frac{\eta H^2}{2}\right) + 24H^2 \log(T) \cdot \frac{1}{T} \sum_{t=1}^T \left(\frac{(H+1)\log(A \vee B)}{\eta t} + \frac{\eta H^2}{2}\right)$$

$$\le 14\eta H^4 \log(T) + 24H^2 \log(T) \cdot \frac{1}{T} \sum_{t=1}^T \frac{(H+1)\log(A \vee B)}{\eta t} + \frac{8H^2 \log(A \vee B)}{\eta T}$$

$$\le 14\eta H^4 \log(T) + \frac{104 \log(A \vee B) \log(T)^2 H^3}{\eta T}.$$

Thus, choosing $\eta = \frac{4}{\sqrt{HT}}$, we get

$$\mathrm{NEGap}(\widehat{\mu}^T, \widehat{\nu}^T) \le \frac{82 \log(A \vee B) \log(T)^2 H^{7/2}}{\sqrt{T}}.$$

$\square$

## D.2   GDA-Critic

The full description of GDA-Critic (Example 2) using V updates is presented in Algorithm 6.

---

**Algorithm 6** GDA-Critic

---

**Require:** Learning rate $\{\alpha_t\}_{t \ge 1}$ (defined in (3)), and $\eta > 0$.
**Initialize:** set $V_h^0(s) = H - h + 1$ and $\mu^0(\cdot|s), \nu^0(\cdot|s)$ to be uniform for all $(h, s) \in [H] \times \mathcal{S}$.
**for** $t = 1, \ldots, T$ **do**
  **for** $h = H, \ldots, 1$ **do**
    Update policy for all $s \in \mathcal{S}$:

$$\mu_h^t(\cdot|s) \leftarrow \mathcal{P}_{\Delta_{\mathcal{A}}}\left(\mu_h^{t-1}(\cdot|s) + \eta\left[(r_h + \mathbb{P}_h V_{h+1}^{t-1})\nu_h^{t-1}\right](s, \cdot, b)\right)$$

$$\nu_h^t(\cdot|s) \leftarrow \mathcal{P}_{\Delta_{\mathcal{B}}}\left(\nu_h^{t-1}(\cdot|s) - \eta\left[(r_h + \mathbb{P}_h V_{h+1}^{t-1})^\top \mu_h^{t-1}\right](s, a, \cdot)\right) \quad\quad (32)$$

    Update V value for all $s \in \mathcal{S}$:

$$V_h^t(s) \leftarrow (1 - \alpha_t)V_h^{t-1}(s) + \alpha_t\left[(\mu_h^t)^\top (r_h + \mathbb{P}_h V_{h+1}^t)\nu_h^t\right](s).$$

---

Similar as Proposition D.1 (with the same proof), the following equivalence between Q updates and V updates also holds for GDA-Critic.

**Proposition D.3** (Equivalence between Q updates and V updates for GDA-Critic). *Algorithm 6 is equivalent to our algorithm framework (Algorithm 1) with the* MatrixGameAlg *instantiated as* (11).

**Lemma D.2** (Per-state regret bound for GDA-Critic). *Algorithm 6 achieves the following per-state regret bound:*

$$\mathrm{reg}_h^t \le \frac{2(H+1)}{\eta t} + \frac{\eta(A \vee B)H^2}{2}.$$

*Proof.* Fix any $(h, s)$ and $t \geq 1$. We apply Lemma B.1 to the projected gradient descent (or ascent) update (32), with weights $w_i = \alpha_t^i$ and loss vectors $g_i$'s $-[Q_h^i \nu_h^i](s)$ or $[(Q_h^i)^\top \mu_h^i](s)$ respectively. For the gradient ascent update for $\mu_h^t(\cdot|s)$, we get

$$\mathrm{reg}_{h,\mu}^t(s) \leq \frac{\alpha_t^t}{2\eta} \cdot 4 + \frac{\eta\left(\sum_{i=1}^t \alpha_t^i\right) A H^2}{2} = \frac{2}{\eta}\frac{H+1}{H+t} + \frac{\eta A H^2}{2}$$

$$\mathrm{reg}_{h,\nu}^t(s) \leq \frac{\alpha_t^t}{2\eta} \cdot 4 + \frac{\eta\left(\sum_{t=1}^T \alpha_t^i\right) B H^2}{2} = \frac{2}{\eta}\frac{H+1}{H+t} + \frac{\eta B H^2}{2}$$

$$\implies \mathrm{reg}_h^t \leq \frac{2(H+1)}{\eta t} + \frac{\eta(A \vee B)H^2}{2}.$$

$\square$

**Proposition D.4** (Guarantee of GDA-Critic). *Algorithm 6 achieves*

$$\mathrm{NEGap}(\widehat{\mu}^T, \widehat{\nu}^T) \leq 14\eta(A \vee B)H^4 \log(T) + \frac{208\log(T)^2 H^3}{\eta T}.$$

*Specifically, picking $\eta = \frac{4}{\sqrt{(A \vee B)HT}}$ yields*

$$\mathrm{NEGap}(\widehat{\mu}^T, \widehat{\nu}^T) \leq \frac{108\log(T)^2\sqrt{A \vee B}H^{7/2}}{\sqrt{T}}.$$

*Proof.* From Lemma D.2, we can take $\overline{\mathrm{reg}}_h^t$ as $\frac{2(H+1)}{\eta t} + \frac{\eta(A \vee B)H^2}{2}$, then from Theorem 2

$$\mathrm{NEGap}(\widehat{\mu}^T, \widehat{\nu}^T) \leq 4H \max_h \overline{\mathrm{reg}}_h^T + 8H^2\left(1 + \frac{1}{H}\right)^H \log(T) \cdot \frac{1}{T}\sum_{t=1}^T \max_h \overline{\mathrm{reg}}_h^t$$

$$\leq 4H\left(\frac{2(H+1)}{\eta T} + \frac{\eta(A \vee B)H^2}{2}\right) + 24H^2 \log(T) \cdot \frac{1}{T}\sum_{t=1}^T \left(\frac{2(H+1)}{\eta t} + \frac{\eta(A \vee B)H^2}{2}\right)$$

$$\leq 14\eta(A \vee B)H^4 \log(T) + 48H^2 \log(T) \cdot \frac{1}{T}\sum_{t=1}^T \frac{(H+1)}{\eta t} + \frac{16H^2}{\eta T}$$

$$\leq 14\eta(A \vee B)H^4 \log(T) + \frac{208\log(T)^2 H^3}{\eta T}.$$

Thus, pick $\eta = \frac{4}{(A \vee B)\sqrt{HT}}$, we get

$$\mathrm{NEGap}(\widehat{\mu}^T, \widehat{\nu}^T) \leq \frac{108\log(T)^2\sqrt{A \vee B}H^{7/2}}{\sqrt{T}}.$$

$\square$

### D.3 Nash Q-Learning (full-information version)

The Nash Q-Learning algorithm (Example 3) is described in Algorithm 7.

**Lemma D.3** (Per-state regret bound for Nash Q-Learning). *Algorithm 7 achieves the following per-state regret bound:*

$$\mathrm{reg}_h^t \leq \frac{(H+1)^2}{H+t}, \quad \forall\, h \in [H],\ t \geq 1.$$

*Proof.* We have

$$\mathrm{reg}_{h,\mu}^t(s) = \max_{\mu^\dagger \in \Delta_{\mathcal{A}}} \sum_{i=1}^t \alpha_t^i \left\langle \mu^\dagger - \mu_h^i(\cdot|s), [Q_h^i \nu_h^i](s, \cdot)\right\rangle$$

**Algorithm 7** Nash Q-Learning

---

**Require:** Learning rate (For Nash Q-learning) $\{\beta_t = \alpha_t\}$;
**Initialize:** $Q_h^0(s, a, b) \leftarrow H - h + 1$ for all $(h, s, a, b)$.
**for** $k = 1, \ldots, K$ **do**
  **for** $h = H, \ldots, 1$ **do**
   Update policy for all $s \in \mathcal{S}$:

$$(\mu_h^t(\cdot|s), \nu_h^t(\cdot|s)) \leftarrow \mathsf{MatrixNE}(Q_h^{t-1}(s, \cdot, \cdot)). \tag{33}$$

   Update Q value for all $(s, a, b) \in \mathcal{S} \times \mathcal{A} \times \mathcal{B}$:

$$Q_h^t(s, a, b) \leftarrow (1 - \alpha_t)Q_h^{t-1}(s, a, b) + \alpha_t\big(r_h + \mathbb{P}_h[(\mu_{h+1}^t)^\top Q_{h+1}^t \nu_{h+1}^t]\big)(s, a, b).$$

---

$$
\begin{aligned}
&= \max_{\mu^\dagger \in \Delta_{\mathcal{A}}} \sum_{i=1}^t \alpha_t^i \left\langle \mu^\dagger - \mu_h^i(\cdot|s), \left[Q_h^{i-1}\nu_h^i\right](s, \cdot) \right\rangle + \sum_{i=1}^t \alpha_t^i \|Q_h^i - Q_h^{i-1}\|_\infty \\
&\leq \sum_{i=1}^t \alpha_t^i \underbrace{\max_{\mu^\dagger \in \Delta_{\mathcal{A}}} \left\langle \mu^\dagger - \mu_h^i(\cdot|s), \left[Q_h^{i-1}\nu_h^i\right](s, \cdot) \right\rangle}_{=0 \text{ from } (33)} + \sum_{i=1}^t \alpha_t^i \|Q_h^i - Q_h^{i-1}\|_\infty \\
&= \sum_{i=1}^t \alpha_t^i \|Q_h^i - Q_h^{i-1}\|_\infty.
\end{aligned}
$$

The same bound also holds for $\mathrm{reg}_{h,\nu}^t(s)$, thus

$$\mathrm{reg}_h^t \leq \sum_{i=1}^t \alpha_t^i \|Q_h^i - Q_h^{i-1}\|_\infty.$$

Since

$$Q_h^i(s, a, b) = (1 - \alpha_i)Q_h^{i-1}(s, a, b) + \alpha_i\big(r_h + \mathbb{P}_h[(\mu_{h+1}^i)^\top Q_{h+1}^i \nu_{h+1}^i]\big)(s, a, b)$$

$$\implies |Q_h^i(s, a, b) - Q_h^{i-1}(s, a, b)| \leq \alpha_i \|Q_h^{i-1} - \big((r_h + \mathbb{P}_h[(\mu_{h+1}^i)^\top Q_{h+1}^i \nu_{h+1}^i])\big)\|_\infty \leq \alpha_i H$$

$$\implies \|Q_h^i - Q_h^{i-1}\|_\infty \leq \alpha_i H,$$

substituting this into the above equations we have that

$$\mathrm{reg}_h^t \leq H \sum_{i=1}^t \alpha_t^i \alpha_i \leq \frac{(H+1)^2}{H+t} \quad \text{(From Lemma A.3 (b))},$$

which completes the proof. $\qquad\square$

**Proposition D.5** (Guarantee for Nash Q-Learning). *Algorithm 7 achieves*

$$\mathrm{NEGap}(\widehat{\mu}^T, \widehat{\nu}^T) \leq \frac{112 \log(T)^2 H^4}{T}.$$

*Proof.* From Theorem 2 and Lemma D.3 we have that

$$\mathrm{NEGap}(\widehat{\mu}^T, \widehat{\nu}^T) \leq 4H \max_h \overline{\mathrm{reg}}_h^T + 8H^2\left(1 + \frac{1}{H}\right)^H \log(T) \cdot \frac{1}{T}\sum_{t=1}^T \max_h \overline{\mathrm{reg}}_h^t$$

$$\leq 4H\frac{(H+1)^2}{H+T} + 24H^2 \log(T) \cdot \frac{1}{T}\sum_{t=1}^T \frac{(H+1)^2}{H+t} \leq \frac{112 \log(T)^2 H^4}{T}.$$

$\qquad\square$

---

**Algorithm 8** Nash Policy Iteration (Nash-PI)

---

**Initialize:** $Q_h^0(s, a, b) \leftarrow H - h + 1$ for all $(h, s, a, b)$.
**for** $t = 1, \dots, T$ **do**
  **for** $h = H, \dots, 1$ **do**
    Update policy for all $s \in \mathcal{S}$:

$$(\mu_h^t(\cdot|s), \nu_h^t(\cdot|s)) \leftarrow \mathsf{MatrixNE}(Q_h^{t-1}(s, \cdot, \cdot)).$$

    Update Q value for all $(s, a, b) \in \mathcal{S} \times \mathcal{A} \times \mathcal{B}$:

$$Q_h^t(s, a, b) \leftarrow \left[r_h + \mathbb{P}_h[(\mu_{h+1}^t)^\top Q_{h+1}^t \nu_{h+1}^t]\right](s, a, b). \tag{34}$$

---

### D.4 Nash Policy Iteration

The full description of Nash Policy Iteration (Nash-PI, Example 4) is presented in Algorithm 8.

Note that from (34), we have that $Q_h^k$ equals to $Q_h^{\mu^k \times \nu^k}$. Based on this observation, we have the following lemma.

**Lemma D.4** (Exact learning of Q functions). *For Algorithm 8, we have for any $h \in [H]$ and $t \geq H - h + 1$ that*

$$Q_h^t(s, a, b) = Q_h^\star(s, a, b), \ \ \forall (s, a, b) \in \mathcal{S} \times \mathcal{A} \times \mathcal{B}.$$

*Proof.* We prove this by backward induction over $h$. For $h = H$, we have that

$$Q_H^t(s, a, b) = r_H(s, a, b), \ \ \forall t \geq 1.$$

Assume that for $h + 1$, the condition holds, then for time horizon $h$ and iteration step $t \geq H - h + 1$, we have that

$$\begin{aligned}
Q_h^t(s, a, b) &= \left[r_h + \mathbb{P}_h[(\mu_{h+1}^t)^\top Q_{h+1}^t \nu_{h+1}^t]\right](s, a, b) \\
&= \left[r_h + \mathbb{P}_h[(\mu_{h+1}^t)^\top Q_{h+1}^\star \nu_{h+1}^t]\right](s, a, b).
\end{aligned}$$

Additionally, from the inductive hypothesis

$$(\mu_{h+1}^t(\cdot|s), \nu_{h+1}^t(\cdot|s)) = \mathsf{MatrixNE}(Q_{h+1}^{t-1}(s, \cdot, \cdot)) = \mathsf{MatrixNE}(Q_{h+1}^\star(s, \cdot, \cdot)),$$

we have that

$$[(\mu_{h+1}^t)^\top Q_{h+1}^\star \nu_{h+1}^t](s) = V_{h+1}^\star(s).$$

Thus

$$\begin{aligned}
Q_h^t(s, a, b) &= \left[r_h + \mathbb{P}_h[(\mu_{h+1}^t)^\top Q_{h+1}^\star \nu_{h+1}^t]\right](s, a, b) \\
&= \left[r_h + \mathbb{P}_h V_{h+1}^\star\right](s, a, b) = Q_h^\star(s, a, b),
\end{aligned}$$

which completes the proof. $\qquad\square$

**Proposition D.6** (Guarantee for Nash-PI). *Algorithm 8 achieves $\mathrm{NEGap}(\widehat{\mu}^T, \widehat{\nu}^T) = 0$ for $T \geq H$.*

*Proof.* For this proposition we will not proof by calling Theorem 2, but instead directly apply Lemma C.1, which is an auxiliary lemma for proving Theorem 2.

Note that Nash-PI corresponds is equivalent to using $\beta_t = 1$ in Algorithm 1, so that $\beta_t^i = \mathbf{1}\{i = t\}$. From Lemma D.4 we have that

$$Q_h^t = Q_h^\star, \ \ \forall t \geq H, h \in [H].$$

Thus for $t \geq H$,

$$\max_{\mu^\dagger \in \Delta_\mathcal{A}, \nu^\dagger \in \Delta_\mathcal{B}} \left[(\mu^\dagger)^\top Q_h^\star \widehat{\nu}_h^T - (\widehat{\mu}_h^T)^\top Q_h^\star \nu^\dagger\right](s)$$

---

**Algorithm 9** OFTRL for two-player zero-sum Markov Games

---

1: **Initialize:** $Q_h^0(s,a,b) \leftarrow H - h + 1$ for all $(h,s,a,b)$.
2: **for** $t = 1, \ldots, T$ **do**
3:     **for** $h = H, \ldots, 1$ **do**
4:         Update policies for all $s \in \mathcal{S}$ by OFTRL:

$$\mu_h^t(a|s) \propto_a \exp\left( (\eta/w_t) \cdot \left[ \sum_{i=1}^{t-1} w_i (Q_h^i \nu_h^i)(s,a) + w_{t-1}(Q_h^{t-1}\nu_h^{t-1})(s,a) \right] \right),$$

$$\nu_h^t(b|s) \propto_b \exp\left( -(\eta/w_t) \cdot \left[ \sum_{i=1}^{t-1} w_i ((Q_h^i)^\top \mu_h^i)(s,b) + w_{t-1}((Q_h^{t-1})^\top \mu_h^{t-1})(s,b) \right] \right).$$

5:         Update Q-value for all $(s,a,b) \in \mathcal{S} \times \mathcal{A} \times \mathcal{B}$:

$$Q_h^t(s,a,b) \leftarrow (1-\alpha_t)Q_h^{t-1}(s,a,b) + \alpha_t \big(r_h + \mathbb{P}_h[(\mu_{h+1}^t)^\top Q_{h+1}^t \nu_{h+1}^t]\big)(s,a,b). \quad (35)$$

6: Output state-wise average policy:

$$\widehat{\mu}_h^T(\cdot|s) \leftarrow \sum_{t=1}^{T} \alpha_T^t \mu_h^t(\cdot|s), \quad \widehat{\nu}_h^T(\cdot|s) \leftarrow \sum_{t=1}^{T} \alpha_T^t \nu_h^t(\cdot|s).$$

---

$$= \max_{\mu^\dagger \in \Delta_\mathcal{A}, \nu^\dagger \in \Delta_\mathcal{B}} \left[ (\mu^\dagger)^\top Q_h^T \nu_h^T - (\mu_h^T)^\top Q_h^T \nu^\dagger \right](s) + 2\sum_{t=1}^{T} \beta_T^t \delta_h^t$$

$$= 0, \quad \forall h.$$

Then applying Lemma C.1, we obtain

$$\text{NEGap}(\widehat{\mu}^T, \widehat{\nu}^T) \le \left( V_1^{\dagger, \widehat{\nu}^T}(s_1) - V_1^\star(s_1) \right) + \left( V_1^\star(s_1) - V_1^{\widehat{\mu}^T, \dagger}(s_1) \right)$$

$$\le \sum_{h=1}^{H} \max_{\mu^\dagger \in \Delta_\mathcal{A}, \nu^\dagger \in \Delta_\mathcal{B}} \left[ (\mu^\dagger)^\top Q_h^\star \widehat{\nu}_h^T - (\widehat{\mu}_h^T)^\top Q_h^\star \nu^\dagger \right](s) = 0, \quad \text{for } T \ge H.$$

This is the desired result. $\qquad\qquad\qquad\qquad\qquad\qquad\qquad\qquad\qquad\qquad\qquad\qquad\qquad\square$

## E   Proof of Theorem 3

In this section we prove Theorem 3. The full algorithm box of OFTRL for Markov Games is provided in Algorithm 9.

We aim to show that

$$\text{NEGap}(\widehat{\mu}^T, \widehat{\nu}^T)$$
$$\le \mathcal{O}\Big( H^{14/3}(\log(A \vee B))^{5/6}(\log T)^{11/6} \cdot T^{-5/6} + H^5 \log(A \vee B)(\log T)^2 \cdot T^{-1} \Big). \tag{36}$$

**Bounding per-state regret**    We first bound $\text{reg}_{\nu,h}^t(s)$, i.e. the per-state regret for the min-player, for any fixed $(h,s,t) \in [H] \times [S] \times [T]$. (The bound for $\text{reg}_{\mu,h}^t(s)$ follows similarly.) This is the main part of this proof.

Throughout this part, we will fix $(h,s)$ and omit these subscripts within the policies and Q functions, so that $\nu_h^t(\cdot|s)$ will be abbreviated as $\nu^t$ (and similarly for $\mu^t$ and $Q^t$). We will also overload $T \ge 1$ to be any positive integer (instead of the fixed total number of iterations).

Observe that the above update for $\nu^t$ is equivalent to the OFTRL algorithm (Algorithm 4) with loss vectors $g_t = w_t(Q^t)^\top \mu^t$ (understanding $g_0 = 0$ and $Q_h^0 = 0$), prediction vector $M_t = g_{t-1} = w_{t-1}(Q^{t-1})^\top \mu^{t-1}$, and learning rate $\eta_t = \eta/w_t$. Therefore we can apply the regret bound for

OFTRL in Lemma B.3 and obtain for any $T \geq 1$ that

$$\max_{\nu^\dagger \in \Delta_\mathcal{B}} \sum_{t=1}^T w_t \left\langle \nu^t - \nu^\dagger, (Q^t)^\top \mu^t \right\rangle = \max_{\nu^\dagger \in \Delta_\mathcal{B}} \sum_{t=1}^T \left\langle \nu^t - \nu^\dagger, g_t \right\rangle$$

$$\leq \frac{\log B}{\eta_T} + \sum_{t=1}^T \eta_t \|g_t - M_t\|_\infty^2 - \sum_{t=1}^{T-1} \frac{1}{8\eta_t} \|\nu^t - \nu^{t+1}\|_1^2 \tag{37}$$

$$= \frac{\log B \cdot w_T}{\eta} + \sum_{t=1}^T \frac{\eta}{w_t} \|w_t(Q^t)^\top \mu^t - w_{t-1}(Q^{t-1})^\top \mu^{t-1}\|_\infty^2 - \sum_{t=2}^T \frac{w_{t-1}}{8\eta} \|\nu^t - \nu^{t-1}\|_1^2.$$

We now relate the terms above to the stability of $\{\mu^t\}_{t\geq 1}$ (the other player's policies). Let

$$\Delta_t := \left\| w_t Q^t - w_{t-1} Q^{t-1} \right\|_\infty$$

for all $t \geq 1$ for shorthand, where $\|\cdot\|_\infty$ for a matrix denotes its infinity norm (i.e. entry-wise max absolute value). Then we have

$$\|w_t(Q^t)^\top \mu^t - w_{t-1}(Q^{t-1})^\top \mu^{t-1}\|_\infty^2$$

$$\leq 2 \left\| (w_t Q^t - w_{t-1} Q^{t-1})^\top \mu^{t-1} \right\|_\infty^2 + 2 \left\| (w_{t-1} Q^{t-1})^\top (\mu^t - \mu^{t-1}) \right\|_\infty^2$$

$$\leq 2\Delta_t^2 + 2w_{t-1}^2 \left\| Q^{t-1} \right\|_\infty^2 \left\| \mu^t - \mu^{t-1} \right\|_1^2$$

$$\leq 2\Delta_t^2 + 2w_{t-1}^2 H^2 \left\| \mu^t - \mu^{t-1} \right\|_1^2 \mathbf{1}\{t \geq 2\}.$$

By symmetry, the similar bound also holds for $\{\nu^t\}$, from which we obtain for any $t \geq 2$ that

$$-w_{t-1} \left\| \nu^t - \nu^{t-1} \right\|_1^2 \leq \frac{\Delta_t^2}{w_{t-1}H^2} - \frac{1}{2w_{t-1}H^2} \left\| w_t Q^t \nu^t - w_{t-1} Q^{t-1} \nu^{t-1} \right\|_\infty^2.$$

Plugging the above two bounds into (37), we get

$$\max_{\nu^\dagger \in \Delta_\mathcal{B}} \sum_{t=1}^T w_t \left\langle \nu^t - \nu^\dagger, (Q^t)^\top \mu^t \right\rangle$$

$$\leq \frac{\log B \cdot w_T}{\eta} + \sum_{t=1}^T \left[ \frac{2\eta}{w_t} \Delta_t^2 + \frac{2\eta H^2 w_{t-1}^2}{w_t} \left\| \mu^t - \mu^{t-1} \right\|_1^2 \mathbf{1}\{t \geq 2\} \right]$$

$$+ \sum_{t=2}^T \left[ \frac{\Delta_t^2}{8\eta H^2 w_{t-1}} - \frac{1}{16\eta H^2 w_{t-1}} \left\| w_t Q^t \nu^t - w_{t-1} Q^{t-1} \nu^{t-1} \right\|_\infty^2 \right]$$

$$\overset{(i)}{\leq} \frac{\log B \cdot w_T}{\eta} + \underbrace{\sum_{t=1}^T \left[ \frac{2\eta}{w_t} + \frac{1}{8\eta H^2 w_{t-1}} \mathbf{1}\{t \geq 2\} \right] \Delta_t^2}_{:=\mathrm{ERR}_T} + \underbrace{4\eta^2 H^2 \cdot \sum_{t=2}^T \frac{1}{2\eta/w_t} \left\| \mu^t - \mu^{t-1} \right\|_1^2}_{:=\mathrm{STAB}_t}$$

$$- \sum_{t=2}^T \frac{1}{16\eta H^2 w_{t-1}} \left\| w_t Q^t \nu^t - w_{t-1} Q^{t-1} \nu^{t-1} \right\|_\infty^2. \tag{38}$$

Above, (i) rearranges terms and used the fact that $w_{t-1} \leq w_t$.

The following lemma (proof deferred to Section E.1) bounds term $\mathrm{ERR}_T$.

**Lemma E.1** (Bound on $\mathrm{ERR}_T$). *Suppose $\eta \leq 1/H$. Then for any $T \geq 1$, we have*

$$\alpha_T^1 \cdot \mathrm{ERR}_T \leq \frac{192 H^2}{\eta T}.$$

To bound term $\mathrm{STAB}_T$, note that it is exactly the total distance (in squared $L_1$ norm) of the sequence $\{\mu^t\}_{t\geq 1}$, which itself follows an OFTRL algorithm with loss sequence $g_t' := -w_t Q^t \nu^t$, $M_t' := g_{t-1}'$, and $\eta_t = \eta/w_t$. Therefore we can apply the stability bound (24) in Lemma B.4 to obtain that

$$\mathrm{STAB}_T = 4\eta^2 H^2 \cdot \sum_{t=2}^T \frac{1}{2\eta_t} \left\| \mu^t - \mu^{t-1} \right\|_1^2$$

$$\leq 4\eta^2 H^2 \left( \frac{2\log A}{\eta_T} + \sum_{t=1}^{T-1} (1 + \eta_t G'_t) \left\| g'_t - g'_{t-1} \right\|_\infty + \left\| g'_{T-1} \right\|_\infty \right)$$

$$= 4\eta^2 H^2 \left( \frac{2w_T \log A}{\eta} + \sum_{t=1}^{T-1} (1 + \eta H) \left\| w_t Q^t \nu^t - w_{t-1} Q^{t-1} \nu^{t-1} \right\|_\infty + \left\| w_{T-1} Q^{T-1} \nu^{T-1} \right\|_\infty \right)$$

$$\overset{(i)}{\leq} 4\eta^2 H^2 \left( \frac{2w_T \log A}{\eta} + 2 \sum_{t=1}^{T-1} \left\| w_t Q^t \nu^t - w_{t-1} Q^{t-1} \nu^{t-1} \right\|_\infty + w_{T-1} H \right)$$

$$\overset{(ii)}{\leq} 4\eta^2 H^2 \left( \frac{4w_T \log A}{\eta} + 2 \sum_{t=1}^{T-1} \left\| w_t Q^t \nu^t - w_{t-1} Q^{t-1} \nu^{t-1} \right\|_\infty \right),$$

where here we take $G'_t = w_t H \geq \left\| g'_t - g'_{t-1} \right\|_\infty$, (i) holds whenever $\eta \leq 1/H$, and (ii) follows as $w_{T-1} H \leq w_T / \eta \leq 2w_T \log A / \eta$.

Plugging the above bounds into (38) yields that for any $T \geq 1$,

$$\mathrm{reg}^T_{\nu,h}(s) = \max_{\nu^\dagger \in \Delta_\mathcal{B}} \sum_{t=1}^T \underbrace{\alpha^t_T}_{\alpha^1_T \cdot w_t} \left\langle \nu^t - \nu^\dagger, (Q^t)^\top \mu^t \right\rangle = \alpha^1_T \max_{\nu^\dagger \in \Delta_\mathcal{B}} \sum_{t=1}^T w_t \left\langle \nu^t - \nu^\dagger, (Q^t)^\top \mu^t \right\rangle$$

$$\leq \frac{\log B \cdot (\alpha^1_T w_T)}{\eta} + \alpha^1_T \mathrm{ERR}_T + \alpha^1_T \left[ \mathrm{STAB}_T - \sum_{t=2}^T \frac{1}{16\eta H^2 w_{t-1}} \left\| w_t Q^t \nu^t - w_{t-1} Q^{t-1} \nu^{t-1} \right\|^2_\infty \right]$$

$$\leq \frac{\log B \cdot \alpha^T_T}{\eta} + \frac{192 H^2}{\eta T} + \alpha^1_T \left[ 16\eta H^2 w_T \log A \right.$$

$$\left. + 8\eta^2 H^2 \sum_{t=1}^{T-1} \left\| w_t Q^t \nu^t - w_{t-1} Q^{t-1} \nu^{t-1} \right\|_\infty - \sum_{t=2}^T \frac{1}{16\eta H^2 w_{t-1}} \left\| w_t Q^t \nu^t - w_{t-1} Q^{t-1} \nu^{t-1} \right\|^2_\infty \right]$$

$$\overset{(i)}{\leq} \frac{\log B \cdot \alpha^T_T}{\eta} + \frac{192 H^2}{\eta T} + \alpha^1_T \left[ 32 \underbrace{\eta H^2}_{\leq 1/\eta} w_T \log A \right.$$

$$\left. + \sum_{t=2}^{T-1} \left( 8\eta^2 H^2 \left\| w_t Q^t \nu^t - w_{t-1} Q^{t-1} \nu^{t-1} \right\|_\infty - \frac{1}{16\eta H^2 w_{t-1}} \left\| w_t Q^t \nu^t - w_{t-1} Q^{t-1} \nu^{t-1} \right\|^2_\infty \right) \right]$$

$$\overset{(ii)}{\leq} \frac{33 \log(A \vee B) \cdot \alpha^T_T}{\eta} + \frac{192 H^2}{\eta T} + \alpha^1_T \sum_{t=2}^{T-1} 256 \eta^5 H^6 w_{t-1}$$

$$\leq \frac{33 \log(A \vee B) \cdot \alpha^T_T}{\eta} + \frac{192 H^2}{\eta T} + 256 \eta^5 H^6 \underbrace{\sum_{t=2}^{T-1} \alpha^{t-1}_T}_{\leq 1}$$

$$\overset{(iii)}{\leq} C \left[ \frac{H^2 \log(A \vee B)}{\eta T} + \eta^5 H^6 \right].$$

Above, (i) used the fact that $8\eta^2 H^2 \left\| w_1 Q^1 \nu^1 \right\|_\infty \leq 8\eta^2 H^3 w_1 \leq 8\eta^2 H^3 w_T \leq 16\eta H^2 w_T \log A$, (ii) used the fact that $8\eta^2 H^2 z - z^2/(16\eta H^2 w_{t-1}) \leq 256\eta^5 H^6 w_{t-1}$ by the AM-GM inequality, and (iii) used the fact that $\alpha^T_T = \alpha_T = (H+1)/(H+T) \leq 2H/T$, where $C \leq 256$ is an absolute constant.

By symmetry, the same regret bound also holds for $\mathrm{reg}^T_{\mu,h}(s)$, which gives that for any $t \geq 1$

$$\mathrm{reg}^t_h := \max_{s \in \mathcal{S}} \max \left\{ \mathrm{reg}^t_{\mu,h}(s), \mathrm{reg}^t_{\nu,h}(s) \right\} \leq \underbrace{C \left[ \frac{H^2 \log(A \vee B)}{\eta t} + \eta^5 H^6 \right]}_{:=\overline{\mathrm{reg}}^t_h}.$$

Note that $\overline{\mathrm{reg}}^t_h$ is decreasing in $t$. This is the desired regret bound.

**Performance of output policy** As our algorithm chooses $\beta_t = \alpha_t = (H+1)/(H+t)$, we can invoke Theorem 2 with $c_\beta = 1 + 1/H \geq \sum_{t=j}^{\infty} \alpha_t^j$ (by Lemma A.2) so that $c_\beta^H = (1 + 1/H)^H \leq e \leq 3$. Further, by the above regret bound,

$$\max_{h \in [H]} \overline{\text{reg}}_h^t \leq C\left[\frac{H^2 \log(A \vee B)}{\eta t} + \eta^5 H^6\right].$$

Plugging this into Theorem 2 yields that the output policy $(\widehat{\mu}^T, \widehat{\nu}^T)$ satisfies

$$\text{NEGap}(\widehat{\mu}^T, \widehat{\nu}^T)$$

$$\leq \mathcal{O}\left(H \max_{h \in [H]} \overline{\text{reg}}_h^T + H^2 c_\beta^H \cdot \frac{\log T}{T} \sum_{t=1}^{T} \max_{h \in [H]} \overline{\text{reg}}_h^t\right)$$

$$\leq H \cdot \mathcal{O}\left(\frac{H^2 \log(A \vee B)}{\eta T} + \eta^5 H^6\right) + H^2 \frac{\log T}{T} \cdot \mathcal{O}\left(\frac{H^2 \log(A \vee B) \log T}{\eta} + \eta^5 H^6 T\right)$$

$$= \mathcal{O}\left(\frac{H^4 \log(A \vee B)(\log T)^2}{\eta T} + \eta^5 H^8 \log T\right).$$

Choosing $\eta = (\log T \log(A \vee B)/H^4 T)^{1/6} \wedge (1/H)$, we get

$$\text{NEGap}(\widehat{\mu}^T, \widehat{\nu}^T) \leq \mathcal{O}\left(H^{14/3}(\log(A \vee B))^{5/6}(\log T)^{11/6} \cdot T^{-5/6} + H^5 \log(A \vee B)(\log T)^2/T\right).$$

This proves (36) and thus Theorem 3.

### E.1 Proof of Lemma E.1

Recall our notation $Q^t := Q_h^t(s, \cdot, \cdot) \in [0, H]^{A \times B}$ for some fixed $(h, s) \in [H] \times \mathcal{S}$. We first note that, for any $t \geq 2$,

$$\left\|w_t Q^t - w_{t-1} Q^{t-1}\right\|_\infty^2 \leq 2\left\|w_t Q^t - w_{t-1} Q^t\right\|_\infty^2 + 2\left\|w_{t-1}(Q^t - Q^{t-1})\right\|_\infty^2$$

$$\leq 2(w_t - w_{t-1})^2 H^2 + 2w_{t-1}^2 \alpha_t^2 H^2$$

$$= 2w_{t-1}^2 H^2\left[\alpha_t^2 + \frac{H^2}{(t-1)^2}\right] \leq 2w_{t-1}^2 H^2 \cdot \frac{8H^2}{t^2} = 16w_{t-1}^2 H^4/t^2.$$

For $t = 1$, we have $\left\|w_t Q^t - w_{t-1} Q^{t-1}\right\|_\infty^2 \leq w_1^2 H^2 = H^2$. Substituting this into the expression of $\text{ERR}_T$ gives

$$\alpha_T^1 \text{ERR}_T$$

$$= \alpha_T^1 \sum_{t=1}^{T} \left(\frac{2\eta}{w_t} + \frac{1}{8\eta w_{t-1} H^2}\mathbf{1}\{t \geq 2\}\right) \cdot \left(H^2 \mathbf{1}\{t = 1\} + 16w_{t-1}^2 H^4/t^2 \cdot \mathbf{1}\{t \geq 2\}\right)$$

$$= 2\eta \alpha_T^1 H^2 + \alpha_T^1 \sum_{t=2}^{T} \left(\frac{2\eta w_{t-1}^2}{w_t} + \frac{w_{t-1}}{8\eta H^2}\right) \cdot \frac{16H^4}{t^2}$$

$$\overset{(i)}{\leq} 2\eta \alpha_T^1 H^2 + \sum_{t=2}^{T} \left(2\eta \alpha_T^t H^2 + \frac{\alpha_T^t}{8\eta}\right) \cdot \frac{16H^2}{t^2}$$

$$\overset{(ii)}{\leq} 2\eta \alpha_T^1 H^2 + \sum_{t=2}^{T} \alpha_T^t \cdot \frac{3}{\eta} \cdot \frac{16H^2}{t^2}$$

$$\overset{(iii)}{\leq} \frac{48H^2}{\eta} \sum_{t=1}^{T} \alpha_T^t \cdot \frac{1}{t^2} \overset{(iv)}{\leq} \frac{192H^2}{\eta T}.$$

Above, (i) used $w_{t-1} \leq w_t$ and $\alpha_T^1 w_t = \alpha_T^t$; (ii) used the fact that $2\eta H^2 \leq 2/\eta$ (as $\eta \leq 1/H$) and thus $2\eta H^2 + 1/(8\eta) \leq (2 + 1/8)/\eta \leq 3/\eta$; (iii) used the fact that $2\eta H^2 \leq 48H^2/\eta$ which also follows from $\eta \leq 1/H \leq 1$; (iv) used Lemma A.3(a). This is the desired result. $\qquad\square$

# F  A modified OFTRL algorithm with $\widetilde{\mathcal{O}}(T^{-1})$ rate

In this section we show that a slightly modified OFTRL algorithm (described Algorithm 10) achieves $\widetilde{\mathcal{O}}(T^{-1})$ convergence rate for finding NE in two-player zero-sum Markov Games, improving over the $\widetilde{\mathcal{O}}(T^{-5/6})$ of Algorithm 9.

---

**Algorithm 10** Modified OFTRL

---

1: **Initialize:** $\overline{Q}_h^0(s,a,b) \leftarrow H - h + 1, \underline{Q}_h^0 \leftarrow 0$ for all $(h,s,a,b)$.
2: **for** $t = 1, \ldots, T$ **do**
3:   **for** $h = H, \ldots, 1$ **do**
4:     Update policies for all $s \in \mathcal{S}$ by OFTRL:
$$\mu_h^t(a|s) \propto_a \exp\left( (\eta/w_t) \cdot \left[ \sum_{i=1}^{t-1} w_i (\overline{Q}_h^i \nu_h^i)(s,a) + w_{t-1}(\overline{Q}_h^{t-1}\nu_h^{t-1})(s,a) \right] \right);$$
$$\nu_h^t(b|s) \propto_b \exp\left( -(\eta/w_t) \cdot \left[ \sum_{i=1}^{t-1} w_i ((\underline{Q}_h^i)^\top \mu_h^i)(s,b) + w_{t-1}((\underline{Q}_h^{t-1})^\top \mu_h^{t-1})(s,b) \right] \right).$$
5:     Update Q-values for all $(s,a,b) \in \mathcal{S} \times \mathcal{A} \times \mathcal{B}$:
$$\overline{Q}_h^t(s,a,b) \leftarrow r_h(s,a,b) + \mathbb{P}_h \left[ \max_{\mu^\dagger \in \Delta_{\mathcal{A}}} \left\langle \mu^\dagger, \sum_{i=1}^t \alpha_t^i \overline{Q}_{h+1}^i \nu_{h+1}^i \right\rangle \right](s,a,b);$$
$$\underline{Q}_h^t(s,a,b) \leftarrow r_h(s,a,b) + \mathbb{P}_h \left[ \min_{\nu^\dagger \in \Delta_{\mathcal{B}}} \left\langle \nu^\dagger, \sum_{i=1}^t \alpha_t^i \left(\underline{Q}_{h+1}^i\right)^\top \mu_{h+1}^i \right\rangle \right](s,a,b). \tag{39}$$

6: Output state-wise average policy for all $(h,s)$:
$$\widehat{\mu}_h^T(\cdot|s) \leftarrow \sum_{t=1}^T \alpha_T^t \mu_h^t(\cdot|s), \quad \widehat{\nu}_h^T(\cdot|s) \leftarrow \sum_{t=1}^T \alpha_T^t \nu_h^t(\cdot|s).$$

---

Algorithm 10 keeps track of a series of $\overline{Q}_h^t, \underline{Q}_h^t$'s that are upper-bounds and lower-bounds of $Q_h^\star$ respectively. The policy update is similar to the update as the OFTRL algorithm (Algorithm 4), but here $\mu$ is performing OFTRL with respect to $\overline{Q}_h^t$'s while $\nu$ with respect to $\underline{Q}_h^t$'s. The value updates (39) are slightly different from the value update in our unified framework, however, we remark that it is still an incremental update because the terms inside the inner product $\sum_{i=1}^t \alpha_t^i \overline{Q}_{h+1}^i \nu_{h+1}^i, \sum_{i=1}^t \alpha_t^i \left(\underline{Q}_{h+1}^i\right)^\top \mu_{h+1}^i$ are incremental updates, which leads to that fact that $\overline{Q}_h^t, \underline{Q}_h^t$'s are also updating incrementally.. Further, the algorithm can be performed in a decentralized manner, which is stated in Algorithm 11. The convergence result is stated in Theorem F.1.

**Theorem F.1** (Convergence rate of modified OFTRL). *Algorithm 10 with $\eta = \frac{1}{16H}$ guarantees that*
$$\mathrm{NEGap}(\widehat{\mu}^T, \widehat{\nu}^T) \leq C \left[ \frac{H^4 \log(A \vee B)(\log T)^2}{T} \right],$$
*where $C$ is some absolute constant.*

## F.1  Proof of Theorem F.1

In this section, we consider the following definitions of regret, which is slightly different from the definition in (7):
$$\mathrm{reg}_{h,\mu}^t(s) := \max_{\mu^\dagger \in \Delta_{\mathcal{A}}} \sum_{i=1}^t \alpha_t^i \left\langle \mu^\dagger - \mu_h^i(\cdot|s), \left[\overline{Q}_h^i \nu_h^i\right](s, \cdot) \right\rangle,$$
$$\mathrm{reg}_{h,\nu}^t(s) := \max_{\nu^\dagger \in \Delta_{\mathcal{B}}} \sum_{i=1}^t \alpha_t^i \left\langle \nu_h^i(\cdot|s) - \nu^\dagger, \left[(\underline{Q}_h^i)^\top \mu_h^i\right](s, \cdot) \right\rangle,$$
$$\mathrm{reg}_{h,\mu+\nu}^t := \max_{s \in \mathcal{S}} \mathrm{reg}_{h,\mu}^t(s) + \mathrm{reg}_{h,\nu}^t(s).$$

We first prove that $\underline{Q}_h^t$ and $\overline{Q}_h^t$ upper and lower bounds $Q_h^\star$ respectively.

**Lemma F.1.**
$$\underline{Q}_h^t(s,a,b) \leq Q_h^\star(s,a,b) \leq \overline{Q}_h^t(s,a,b).$$

**Algorithm 11** Modified OFTRL (Equivalent V-form)

---

1: **Initialize:** $\overline{V}_h^1(s) \leftarrow H - h + 1, \underline{V}_h^1(s) \leftarrow 0$ for all $(h, s, a, b)$.
2: **for** $t = 1, \ldots, T$ **do**
3:    **for** $h = H, \ldots, 1$ **do**
4:       Update policies for all $s \in \mathcal{S}$ by OFTRL:
$$\mu_h^t(a|s) \propto_a \exp\left( (\eta/w_t) \cdot \left[ \sum_{i=1}^{t-1} w_i \overline{L}_h^i(s, a) + w_{t-1} \overline{L}_h^{t-1}(s, a) \right] \right)$$
$$\nu_h^t(b|s) \propto_b \exp\left( -(\eta/w_t) \cdot \left[ \sum_{i=1}^{t-1} w_i ((Q_h^i)^\top \mu_h^i)(s, b) + w_{t-1} ((Q_h^{t-1})^\top \mu_h^{t-1})(s, b) \right] \right).$$
5:       Update losses for all $(s, a) \in \mathcal{S} \times \mathcal{A}$:
$$\overline{L}_h^t(s, a) \leftarrow \left\langle r_h(s, a, \cdot) + \left[ \mathbb{P}_h \overline{V}_{h+1}^t \right](s, a, \cdot), \nu_h^t(\cdot|s) \right\rangle,$$
$$\underline{L}_h^t(s, a) \leftarrow \left\langle \left[ r_h(s, a, \cdot) + \left[ \mathbb{P}_h \underline{V}_{h+1}^t \right](s, a, \cdot) \right]^\top, \mu_h^t(\cdot|s) \right\rangle.$$
6:       Update V-value for all $s \in \mathcal{S}$:
$$\overline{V}_h^t(s) \leftarrow \max_{\mu^\dagger \in \Delta_{\mathcal{A}}} \left\langle \mu^\dagger, \sum_{i=1}^t \alpha_t^i \overline{L}_h^i(s, \cdot) \right\rangle, \quad \underline{V}_h^t(s) \leftarrow \min_{\nu^\dagger \in \Delta_{\mathcal{B}}} \left\langle \nu^\dagger, \sum_{i=1}^t \alpha_t^i \underline{L}_h^i(s, \cdot) \right\rangle. \tag{40}$$

7:  Output state-wise average policy for all $(h, s)$:
$$\widehat{\mu}_h^T(\cdot|s) \leftarrow \sum_{t=1}^T \alpha_T^t \mu_h^t(\cdot|s), \quad \widehat{\nu}_h^T(\cdot|s) \leftarrow \sum_{t=1}^T \alpha_T^t \nu_h^t(\cdot|s).$$

---

*Proof.* We prove by induction on $(h, t)$. Given the initialization, for $t = 0$ the condition holds. Since $\overline{Q}_{H+1}^t, \underline{Q}_{H+1}^t = 0$, we have that for $h = H + 1$ the condition holds. Assume that the condition hold for $(i, h + 1), i \leq t$, then

$$\overline{Q}_h^t(s, a, b) = r_h(s, a, b) + \mathbb{P}_h \left[ \max_{\mu^\dagger \in \Delta_{\mathcal{A}}} \left\langle \mu^\dagger, \sum_{i=1}^t \alpha_t^i \overline{Q}_{h+1}^i \nu_{h+1}^i \right\rangle \right](s, a, b)$$

$$\geq r_h(s, a, b) + \mathbb{P}_h \left[ \max_{\mu^\dagger \in \Delta_{\mathcal{A}}} \left\langle \mu^\dagger, Q_{h+1}^\star \left( \sum_{i=1}^t \alpha_t^i \nu_{h+1}^i \right) \right\rangle \right](s, a, b)$$

$$\geq r_h(s, a, b) + \mathbb{P}_h \left[ \max_{\mu^\dagger \in \Delta_{\mathcal{A}}} \min_{\nu^\dagger \in \Delta_{\mathcal{B}}} \left\langle \mu^\dagger, Q_{h+1}^\star \nu^\dagger \right\rangle \right](s, a, b)$$

$$= Q_h^\star(s, a, b).$$

Using similar strategy, we can also show that $\underline{Q}_h^t(s, a, b) \leq Q_h^\star(s, a, b)$, which implies that the condition hold for $(t, h)$, and thus finishes the proof by induction. $\square$

Throughout the rest of this section, we define the following shorthand for the gap between $\overline{Q}_h^t, \underline{Q}_h^t$ defined in (39):
$$\delta_h^t := \|\overline{Q}_h^t - \underline{Q}_h^t\|_\infty = \max_{s, a, b} \left[ \overline{Q}_h^t(s, a, b) - \underline{Q}_h^t(s, a, b) \right],$$

**Lemma F.2** (Recursion of $\delta_h^t$). *Algorithm 10 guarantees that for all $(t, h) \in [T] \times [H]$,*

$$\delta_h^t \leq \sum_{i=1}^t \alpha_t^i \delta_{h+1}^i + \text{reg}_{h+1, \mu+\nu}^t.$$

*Further, suppose that $\text{reg}_{h, \mu+\nu}^t \leq \overline{\text{reg}}_{h, \mu+\nu}^t$ for all $(h, t) \in [H] \times [T]$, where $\overline{\text{reg}}_{h, \mu+\nu}^t$ is non-increasing in $t$: $\overline{\text{reg}}_{h, \mu+\nu}^t \geq \overline{\text{reg}}_{h, \mu+\nu}^{t+1}$ for all $t \geq 1$. Then we have*

$$\delta_h^t \leq 2H \cdot \frac{1}{t} \sum_{i=1}^t \max_{h'} \overline{\text{reg}}_{h', \mu+\nu}^i.$$

*Proof.* The proof structure resembles Lemma C.2. From the definition of $\overline{Q}_h^t, \underline{Q}_h^t$, we have that

$$\overline{Q}_h^t(s,a,b) - \underline{Q}_h^t(s,a,b)$$

$$\leq \mathbb{P}_h \max_{\mu^\dagger \in \Delta_\mathcal{A}, \nu^\dagger \in \Delta_\mathcal{B}} \left\langle \mu^\dagger, \sum_{i=1}^t \alpha_t^i \overline{Q}_{h+1}^i \nu_{h+1}^i \right\rangle - \left\langle \nu^\dagger, \sum_{i=1}^t \alpha_t^i (\underline{Q}_{h+1}^i)^\top \mu_{h+1}^i \right\rangle$$

$$= \mathbb{P}_h \left[ \max_{\mu^\dagger \in \Delta_\mathcal{A}} \left\langle \mu^\dagger, \sum_{i=1}^t \alpha_t^i \overline{Q}_{h+1}^i \nu_{h+1}^i \right\rangle - \sum_{i=1}^t \alpha_t^i (\mu_{h+1}^i)^\top \overline{Q}_{h+1}^i \nu_{h+1}^i \right.$$

$$+ \max_{\nu^\dagger \in \Delta_\mathcal{B}} \sum_{i=1}^t \alpha_t^i (\mu_{h+1}^i)^\top \overline{Q}_{h+1}^i \nu_{h+1}^i - \left\langle \nu^\dagger, \sum_{i=1}^t \alpha_t^i (\underline{Q}_{h+1}^i)^\top \mu_{h+1}^i \right\rangle$$

$$+ \left. \sum_{i=1}^t \alpha_t^i (\mu_{h+1}^i)^\top \overline{Q}_{h+1}^i \nu_{h+1}^i - \sum_{i=1}^t \alpha_t^i (\mu_{h+1}^i)^\top \overline{Q}_{h+1}^i \nu_{h+1}^i \right]$$

$$\leq \mathrm{reg}_{h+1,\mu+\nu}^t + \sum_{i=1}^t \alpha_t^i \|\overline{Q}_{h+1}^i - \underline{Q}_{h+1}^i\|_\infty = \sum_{i=1}^t \alpha_t^i \delta_{h+1}^i + \mathrm{reg}_{h+1,\mu+\nu}^t.$$

Then using the same argument as Lemma C.2, we can consider an auxiliary sequence

$$\begin{cases} \Delta_h^t = \sum_{i=1}^t \alpha_t^i \Delta_{h+1}^i + \overline{\mathrm{reg}}_{h+1,\mu+\nu}^t, \\ \Delta_{H+1}^t = 0, \quad \text{for all } t. \end{cases} \tag{41}$$

Observe that $\{\Delta_h^t\}_{h,t}$ satisfies the following properties

$$\begin{cases} \Delta_h^t \geq \delta_h^t & \text{(by definition)}, \\ \Delta_h^t \leq \Delta_h^{t-1} & \text{(by Lemma A.1)}. \end{cases} \tag{42}$$

Therefore, to control $\delta_h^t$, it suffices to bound $\Delta_h^t \leq \frac{1}{t}\sum_{i=1}^t \Delta_h^i$, which follows from the standard argument in [23]:

$$\frac{1}{t}\sum_{i=1}^t \Delta_h^i = \frac{1}{t}\sum_{i=1}^t \sum_{j=1}^i \alpha_i^j \Delta_{h+1}^j + \frac{1}{t}\sum_{i=1}^t \overline{\mathrm{reg}}_{h+1,\mu+\nu}^i$$

$$\leq \frac{1}{t}\sum_{j=1}^t \left( \sum_{i=j}^t \alpha_i^j \right) \Delta_{h+1}^j + \frac{1}{t}\sum_{i=1}^t \overline{\mathrm{reg}}_{h+1,\mu+\nu}^i$$

$$\leq \left(1 + \frac{1}{H}\right) \cdot \frac{1}{t}\sum_{i=1}^t \Delta_{h+1}^i + \frac{1}{t}\sum_{i=1}^t \overline{\mathrm{reg}}_{h+1,\mu+\nu}^i$$

$$\leq \left(1 + \frac{1}{H}\right)^2 \cdot \frac{1}{t}\sum_{i=1}^t \Delta_{h+2}^i + \left(1 + \frac{1}{H}\right) \cdot \frac{1}{t}\sum_{i=1}^t \overline{\mathrm{reg}}_{h+2,\mu+\nu}^i + \frac{1}{t}\sum_{i=1}^t \overline{\mathrm{reg}}_{h+1,\mu+\nu}^i$$

$$\leq \cdots$$

$$\leq \left( \sum_{h'=h}^H \left(1 + \frac{1}{H}\right)^{h'-h} \right) \cdot \frac{1}{t}\sum_{i=1}^t \max_{1 \leq h' \leq H} \overline{\mathrm{reg}}_{h',\mu+\nu}^i$$

$$\leq (e-1)H \cdot \frac{1}{t}\sum_{i=1}^t \max_{1 \leq h' \leq H} \overline{\mathrm{reg}}_{h',\mu+\nu}^i \leq 2H \cdot \frac{1}{t}\sum_{i=1}^t \max_{1 \leq h' \leq H} \overline{\mathrm{reg}}_{h',\mu+\nu}^i.$$

which completes the proof. $\qquad\square$

**Lemma F.3** (Bound the NEGap by $\mathrm{reg}_{h,\mu+\nu}$). *Suppose that the per-state regrets (summing over the two agents) can be upper-bounded as $\mathrm{reg}_{h,\mu+\nu}^t \leq \overline{\mathrm{reg}}_{h,\mu+\nu}^t$ for all $(h,t) \in [H] \times [T]$ where $\overline{\mathrm{reg}}_{h,\mu+\nu}^t$ is non-increasing in t: $\overline{\mathrm{reg}}_{h,\mu+\nu}^t \geq \overline{\mathrm{reg}}_{h,\mu+\nu}^{t+1}$ for all $t \geq 1$. Then, the output policy $(\widehat{\mu}^T, \widehat{\nu}^T)$ of Algorithm 10 satisfies*

$$\mathrm{NEGap}(\widehat{\mu}^T, \widehat{\nu}^T) \leq 2H \max_h \overline{\mathrm{reg}}_{h,\mu+\nu}^T + 24H^2 \log T \cdot \frac{1}{T}\sum_{t=1}^T \max_h \overline{\mathrm{reg}}_{h,\mu+\nu}^t$$

*Proof.* From Lemma C.1 we have that

$$\mathrm{NEGap}(\widehat{\mu}^T, \widehat{\nu}^T) = \left(V_1^{\dagger, \widehat{\nu}^T}(s_1) - V_1^{\star}(s_1)\right) + \left(V_1^{\star}(s_1) - V_1^{\widehat{\mu}^T, \dagger}(s_1)\right)$$

$$\leq 2 \sum_{h=1}^{H} \max_s \max_{\mu^{\dagger} \in \Delta_{\mathcal{A}}, \nu^{\dagger} \in \Delta_{\mathcal{B}}} \left[(\mu^{\dagger})^{\top} Q_h^{\star} \widehat{\nu}_h^T - (\widehat{\mu}_h^T)^{\top} Q_h^{\star} \nu^{\dagger}\right](s)$$

$$= 2 \sum_{h=1}^{H} \max_s \max_{\mu^{\dagger} \in \Delta_{\mathcal{A}}, \nu^{\dagger} \in \Delta_{\mathcal{B}}} \sum_{t=1}^{T} \alpha_T^t \left[(\mu^{\dagger})^{\top} Q_h^{\star} \nu_h^t - (\mu_h^t)^{\top} Q_h^{\star} \nu^{\dagger}\right](s)$$

$$\leq 2 \sum_{h=1}^{H} \max_s \max_{\mu^{\dagger} \in \Delta_{\mathcal{A}}, \nu^{\dagger} \in \Delta_{\mathcal{B}}} \sum_{t=1}^{T} \alpha_T^t \left[(\mu^{\dagger})^{\top} \overline{Q}_h^t \nu_h^t - (\mu_h^t)^{\top} \underline{Q}_h^t \nu^{\dagger}\right](s)$$

$$\leq 2 \sum_{h=1}^{H} \max_s \left( \max_{\mu^{\dagger} \in \Delta_{\mathcal{A}}} \sum_{t=1}^{T} \alpha_T^t \left[(\mu^{\dagger})^{\top} \overline{Q}_h^t \nu_h^t - (\mu_h^t)^{\top} \overline{Q}_h^t \nu_h^t\right](s) \right.$$

$$\left. + \max_{\nu^{\dagger} \in \Delta_{\mathcal{B}}} \sum_{t=1}^{T} \alpha_T^t \left[(\mu_h^t)^{\top} \underline{Q}_h^t \nu_h^t - (\mu_h^t)^{\top} \underline{Q}_h^t \nu^{\dagger}\right](s) \right)$$

$$+ 2 \sum_{h=1}^{H} \max_s \max_{\mu^{\dagger} \in \Delta_{\mathcal{A}}, \nu^{\dagger} \in \Delta_{\mathcal{B}}} \sum_{t=1}^{T} \alpha_T^t \left[(\mu_h^t)^{\top} \overline{Q}_h^t \nu_h^t - (\mu_h^t)^{\top} \underline{Q}_h^t \nu_h^t\right](s)$$

$$\leq 2 \sum_{h=1}^{H} \overline{\mathrm{reg}}_{h,\mu+\nu}^T + 2 \sum_{h=1}^{H} \sum_{t=1}^{T} \alpha_T^t \delta_h^t$$

$$\leq 2H \max_h \overline{\mathrm{reg}}_{h,\mu+\nu}^T + 4H^2 \sum_{t=1}^{T} \alpha_T^t \frac{1}{t} \sum_{i=1}^{t} \max_h \mathrm{reg}_{h,\mu+\nu}^i \quad (\text{Lemma F.1})$$

$$\leq 2H \max_h \overline{\mathrm{reg}}_{h,\mu+\nu}^T + 4H^2 \left(\sum_{t=1}^{T} \frac{1}{t} \alpha_T^t\right) \left(\sum_{i=1}^{T} \max_h \mathrm{reg}_{h,\mu+\nu}^i\right)$$

$$\leq 2H \max_h \overline{\mathrm{reg}}_{h,\mu+\nu}^T + 24H^2 \log T \cdot \frac{1}{T} \sum_{t=1}^{T} \max_h \mathrm{reg}_{h,\mu+\nu}^t \quad (\text{Lemma A.3}),$$

$\square$

**Lemma F.4** (Bound $\mathrm{reg}_{h,\mu+\nu}^t$). *Running Algorithm 10 with $\eta = \frac{1}{16H}$ can guarantee that*

$$\mathrm{reg}_{h,\mu+\nu}^T(s) \leq \frac{36H^2 \log(A \vee B)}{T}$$

*Proof.* From Lemma B.3, substituting $g_t = w_t \overline{Q}_h^t \nu_h^t(s), M_t = w_t \overline{Q}_h^{t-1} \nu_h^{t-1}(s), \eta_t = \frac{\eta}{w_t}$, we can get that

$$\sum_{t=1}^{T} w_t \left[\left\langle \mu^{\dagger}, \overline{Q}_h^t \nu_h^t\right\rangle - \left\langle \mu_h^t, \overline{Q}_h^t \nu_h^t\right\rangle\right]$$

$$\leq \frac{w_T \log A}{\eta} + \eta \sum_{t=1}^{T} w_t \|\overline{Q}_h^t \nu_h^t(s) - \overline{Q}_h^{t-1} \nu_h^{t-1}(s)\|_{\infty}^2 - \sum_{t=2}^{T} \frac{w_t}{8\eta} \|\mu_h^t(\cdot|s) - \mu_h^{t-1}(\cdot|s)\|_1^2$$

$$\implies \mathrm{reg}_{h,\mu}^T \leq \alpha_T^1 \sum_{t=1}^{T} w_{t-1} \left\langle \mu^{\dagger}, \overline{Q}_h^t \nu_h^t\right\rangle - \left\langle \mu_h^t, \overline{Q}_h^t \nu_h^t\right\rangle$$

$$\leq \frac{\alpha_T \log A}{\eta} + \eta \sum_{t=1}^{T} \alpha_T^t \|\overline{Q}_h^t \nu_h^t(s) - \overline{Q}_h^{t-1} \nu_h^{t-1}(s)\|_{\infty}^2 - \sum_{t=2}^{T} \frac{\alpha_T^{t-1}}{8\eta} \|\mu_h^t(\cdot|s) - \mu_h^{t-1}(\cdot|s)\|_1^2.$$

Further we have that

$$\|\overline{Q}_h^t \nu_h^t(s) - \overline{Q}_h^{t-1}\nu_h^{t-1}(s)\|_\infty^2 \leq 2\|\overline{Q}_h^t - \overline{Q}_h^{t-1}\|_\infty^2 + 2\|\nu_h^t(s) - \nu_h^{t-1}(s)\|_1^2.$$

From the definition of $\overline{Q}_h^t$ we have that

$$
\begin{aligned}
\|\overline{Q}_h^t - \overline{Q}_h^{t-1}\| &\leq \left\|\sum_{i=1}^t \alpha_t^i \overline{Q}_{h+1}^i \nu_{h+1}^i - \sum_{i=1}^{t-1} \alpha_{t-1}^i \overline{Q}_{h+1}^i \nu_{h+1}^i\right\|_\infty \\
&= \left\|\alpha_t \overline{Q}_{h+1}^t \nu_{h+1}^t + (1-\alpha_t)\sum_{i=1}^{t-1}\alpha_{t-1}^i \overline{Q}_{h+1}^i \nu_{h+1}^i - \sum_{i=1}^{t-1}\alpha_{t-1}^i \overline{Q}_{h+1}^i \nu_{h+1}^i\right\|_\infty \\
&= \left\|\alpha_t \overline{Q}_{h+1}^t \nu_{h+1}^t - \alpha_t \sum_{i=1}^{t-1}\alpha_{t-1}^i \overline{Q}_{h+1}^i \nu_{h+1}^i\right\|_\infty \leq \alpha_t H.
\end{aligned}
$$

Substitute this inequality to the regret bound we have

$$\mathrm{reg}_{h,\mu}^T(s)$$

$$\leq \frac{\alpha_T \log A}{\eta} + 2\eta \sum_{t=1}^T \alpha_T^t \alpha_t^2 H^2 + 2\eta \sum_{t=1}^T \alpha_T^t \|\nu_h^t(s) - \nu_h^{t-1}(s)\|_1^2 - \sum_{t=2}^T \frac{\alpha_T^{t-1}}{8\eta}\|\mu_h^t(\cdot|s) - \mu_h^{t-1}(\cdot|s)\|_1^2$$

$$\overset{\text{(Lemma A.3)}}{\leq} \frac{\alpha_T \log A}{\eta} + \frac{8\eta H^3}{T} + 2\eta \sum_{t=1}^T \alpha_T^t \|\nu_h^t(s) - \nu_h^{t-1}(s)\|_1^2 + - \sum_{t=2}^T \frac{\alpha_T^{t-1}}{8\eta}\|\mu_h^t(\cdot|s) - \mu_h^{t-1}(\cdot|s)\|_1^2.$$

Similar bound holds for $\mathrm{reg}_{h,\nu}^T$:

$$\mathrm{reg}_{h,\nu}^T(s) \leq \frac{\alpha_T \log B}{\eta} + \frac{8\eta H^3}{T} + 2\eta \sum_{t=1}^T \alpha_T^t \|\mu_h^t(s) - \mu_h^{t-1}(s)\|_1^2 + - \sum_{t=2}^T \frac{\alpha_T^{t-1}}{8\eta}\|\nu_h^t(\cdot|s) - \nu_h^{t-1}(\cdot|s)\|_1^2.$$

Summing $\mathrm{reg}_{h,\mu}^T(s), \mathrm{reg}_{h,\nu}^T(s)$ together we get

$$
\begin{aligned}
\mathrm{reg}_{h,\mu+\nu}^T(s) \leq & \frac{2\alpha_T \log(A \vee B)}{\eta} + \frac{16\eta H^3}{T} + 16\eta \alpha_T^1 \\
& + \sum_{t=2}^T \left(2\eta \alpha_T^t - \frac{\alpha_T^{t-1}}{8\eta}\right)\left(\|\mu_h^t(\cdot|s) - \mu_h^{t-1}(\cdot|s)\|_1^2 + \|\nu_h^t(\cdot|s) - \nu_h^{t-1}(\cdot|s)\|_1^2\right).
\end{aligned}
$$

Since $\frac{\alpha_T^{t-1}}{\alpha_T^t} \geq \frac{1}{H}$ for $t \geq 2$, by setting $\eta = \frac{1}{16H}$ we can guarantee that $2\eta\alpha_T^t - \frac{\alpha_T^{t-1}}{8\eta} \leq 0$, thus

$$
\begin{aligned}
\mathrm{reg}_{h,\mu+\nu}^T(s) &\leq \frac{2\alpha_T \log(A \vee B)}{\eta} + \frac{16\eta H^3}{T} + 16\eta\alpha_T^1 \leq \frac{32H^2 \log(A \vee B)}{T} + \frac{H^2}{T} + \frac{1}{T} \\
&\leq \frac{36H^2 \log(A \vee B)}{T}
\end{aligned}
$$

$\square$

Given Lemma F.3 and F.4, we are now ready to prove Theorem F.1.

*Proof Theorem F.1.* From Lemma F.3 and F.4 we have that:

$$
\begin{aligned}
\mathrm{NEGap}(\widehat{\mu}^T, \widehat{\nu}^T) &\leq 2H \max_h \overline{\mathrm{reg}}_{h,\mu+\nu}^T + 24H^2 \log T \cdot \frac{1}{T}\sum_{t=1}^T \max_h \overline{\mathrm{reg}}_{h,\mu+\nu}^t \\
&\leq 2H \frac{36H^2 \log(A \vee B)}{T} + 24H^2 \log T \cdot \frac{1}{T}\sum_{t=1}^T \frac{36H^2 \log(A \vee B)}{t} \\
&\leq \frac{936H^4 \log(A \vee B)(\log T + 1)^2}{T},
\end{aligned}
$$

which completes the proof. $\square$

# G Optimistic policy optimization for general-sum Markov Games

## G.1 Preliminaries

Here we formally present the preliminaries for multi-player general-sum Markov games, parallel to the zero-sum setting considered in Section 2.

**Multi-player general-sum Markov games**   We consider tabular episodic (finite-horizon) $m$-player general-sum Markov games (MGs), which can be denoted as $\mathcal{M}(H, \mathcal{S}, \{\mathcal{A}_i\}_{i=1}^m, \mathbb{P}, \{r_i\}_{i=1}^m)$, where $H$ is the horizon length; $\mathcal{S}$ is the state space with $|\mathcal{S}| = S$; $\mathcal{A}_i$ is the action space of the $i$-th player, with $|\mathcal{A}_i| = A_i$. We use $\mathbf{a} = (a_1, \ldots, a_m) \in \prod_{i \in [m]} \mathcal{A}_i =: \mathcal{A}$ to denote a joint action taken by all players; $\mathbb{P} = \{\mathbb{P}_h\}_{h=1}^H$ is the transition probabilities, where each $\mathbb{P}_h(s'|s, \mathbf{a})$ gives the probability of transition to state $s'$ from state-action $(s, \mathbf{a})$; $r_i = \{r_{i,h}\}_{h=1}^H$ are the reward functions, where each $r_{i,h}(s, \mathbf{a})$ is the deterministic reward function of the $i$-th player at time step $h$ and state-action $(s, \mathbf{a})$. In each episode, the MG starts with a deterministic initial state $s_1$. Then at each time step $1 \leq h \leq H$, all players observes the state $s_h$, each player takes an action $a_{i,h} \in \mathcal{A}_i$. Then, each player receive their rewards $r_{i,h}(s_h, \mathbf{a}_h)$, and the game transitions to the next state $s_{t+1} \sim \mathbb{P}_h(\cdot|s_h, \mathbf{a}_h)$.

**Policies & value functions**   A (Markov) policy $\pi_i$ of the $i$-th player is a collection of policies $\pi_i = \{\pi_{i,h} : \mathcal{S} \to \Delta_{\mathcal{A}_i}\}_{h=1}^H$, where each $\pi_{i,h}(\cdot|s_h) \in \Delta_{\mathcal{A}_i}$ specifies the probability of taking action $a_{i,h}$ at $(h, s_h)$. We use $\pi = \{\pi_i\}_{i \in [m]}$ to denote a product policy of all players. For any joint policy $\pi$ (not necessarily a product policy), we use $V_{i,h}^\pi : \mathcal{S} \to \mathbb{R}$ and $Q_{i,h}^\pi : \mathcal{S} \times \mathcal{A} \to \mathbb{R}$ to denote the ($i$-th player's) value function and Q-function at time step $h$, respectively, i.e.

$$V_{i,h}^\pi(s) := \mathbb{E}_\pi \left[ \sum_{h=h'}^H r_{i,h'}(s_{h'}, \mathbf{a}_{h'}) \mid s_h = s \right], \tag{43}$$

$$Q_{i,h}^\pi(s, \mathbf{a}) := \mathbb{E}_\pi \left[ \sum_{h=h'}^H r_{i,h'}(s_{h'}, \mathbf{a}_{h'}) \mid s_h = s, \mathbf{a}_h = \mathbf{a} \right]. \tag{44}$$

For notational simplicity, we use the following abbreviation: $[\mathbb{P}_h V](s, \mathbf{a}) := \mathbb{E}_{s' \sim \mathbb{P}_h(\cdot|s, \mathbf{a})} V(s')$ for any value function $V$. By definition of the value functions and Q-functions, we have the following Bellman equations for all Markov product policy $\pi$ and all $(i, h, s, \mathbf{a})$:

$$Q_{i,h}^\pi(s, \mathbf{a}) = \left( r_{i,h} + \mathbb{P}_h V_{i,h+1}^\pi \right)(s, \mathbf{a}),$$

$$V_{i,h}^\pi(s, \mathbf{a}) = \mathbb{E}_{\mathbf{a} \sim \pi_h(\cdot|s)} \left[ Q_{i,h}^\pi(s, \mathbf{a}) \right] = \left\langle Q_{i,h}^\pi(s, \cdot), \pi_h(\cdot|s) \right\rangle.$$

The goal for the $i$-th player is to maximize their own value function.

**Correlated policy & best response**   A (general) correlated policy $\pi$ is any policy for which players may take actions in a history-dependent and correlated fashion. More precisely, a correlated policy $\pi$ is a mapping $\{\pi_h : \Omega \times (\mathcal{S} \times \mathcal{A})^{h-1} \times \mathcal{S} \to \Delta_{\mathcal{A}}\}$, and executes as follows. At the beginning of an episode, a random seed $w \in \Omega$ is sampled from some distribution (also denoted as $\Omega$ with slight abuse of notation). Then, at each step $h$ and state $s_h$, suppose the history so far is $(s_1, \mathbf{a}_1, \ldots, s_{h-1}, \mathbf{a}_{h-1})$. Then, $\pi$ samples a joint action $\mathbf{a}_h \sim \pi_h(\cdot|\omega, (s_1, \mathbf{a}_1, \ldots, s_{h-1}, \mathbf{a}_{h-1}); s_h)$. This formulation allows each $\pi_h(\cdot|\omega, \cdot, \cdot)$ to be a Markov product policy for any *fixed* $\omega$ while still making $\pi$ to be a correlated policy, due to the correlation introduced by $\omega$.

For any correlated policy $\pi$, let $\pi_{-i}$ denote the (marginal) policy of all but the $i$-th player. Then, the ($i$-th) player's best-response value function is

$$V_{i,1}^{\dagger, \pi_{-i}}(s_1) := \max_{\pi_i^\dagger} V_{i,1}^{\pi_i^\dagger \times \pi_{-i}}(s_1),$$

where the max is over all (potentially history-dependent) policy $\pi_i^\dagger$ for the $i$-th player.

**Coarse Correlated Equilibrium (CCE)**   For general-sum MGs, we consider learning an approximate Coarse Correlated Equilibrium [33, 49] defined as follows.

**Definition G.1** ($\varepsilon$-approximate Coarse Correlated Equilibrium). *For any $\varepsilon \geq 0$, a correlated policy $\pi$ is an $\varepsilon$-approximate Coarse Correlated Equilibrium ($\varepsilon$-CCE) if*

$$\mathrm{CCEGap}(\pi) := \max_{i \in [m]} V_{i,1}^{\dagger, \pi_{-i}}(s_1) - V_{i,1}^\pi(s_1) \leq \varepsilon.$$

---

**Algorithm 12** OFTRL for multi-player general-sum Markov games

---

1: **Initialize:** $Q_h^0(s, \mathbf{a}) \leftarrow H - h + 1$ for all $(h, s, a, b)$.
2: **for** $t = 1, \ldots, T$ **do**
3:    **for** $h = H, \ldots, 1$ **do**
4:       Update policies for all $s \in \mathcal{S}$ and $i \in [m]$ by OFTRL

$$\pi_{i,h}^t(a_i|s) \propto_{a_i} \exp\left( (\eta/w_t) \cdot \left[ \sum_{j=1}^{t-1} w_j (Q_{i,h}^j \pi_{-i,h}^j)(s, a_i) + w_t (Q_{i,h}^{t-1} \pi_{-i,h}^{t-1})(s, a_i) \right] \right).$$
(45)

5:       Update Q-value for all $(i, s, \mathbf{a}) \in [m] \times \mathcal{S} \times \mathcal{A}$:

$$Q_{i,h}^t(s, \mathbf{a}) \leftarrow (1 - \alpha_t)Q_{i,h}^{t-1}(s, \mathbf{a}) + \alpha_t(r_h + \mathbb{P}_h[Q_{i,h+1}^t \pi_{h+1}^t])(s, \mathbf{a}).$$
(46)

6: Output policy $\widehat{\pi}^T = \widehat{\pi}_1^T$, where $\widehat{\pi}_1^T$ is defined in Algorithm 13.

---

**Additional notation**    For any Q function $Q_{i,h}(s, \cdot) : \mathcal{S} \times (\prod_{i=1}^m \mathcal{A}_i) \to \mathbb{R}$ and joint policy $\pi_h(\cdot|s)$, we use $[Q_{i,h}\pi_h](s) := \langle Q_{i,h}(s, \cdot), \pi_h(\cdot|s) \rangle$ for shorthand. Similarly, for any joint policy $\pi_{-i,h}(\cdot|s)$ over all but the $i$-th player, $[Q_{i,h}\pi_{-i,h}](s, a_i) := \langle Q_{i,h}(s, a_i, \cdot), \pi_{-i,h}(\cdot|s) \rangle$.

## G.2   Algorithm and formal statement of result

---

**Algorithm 13** Policy $\widehat{\pi}_h^t$

---

**Require:** Product policies $\pi_{h'}^{t'}(\cdot|s') = \prod_{i=1}^m \pi_{i,h'}^{t'}(\cdot|s')$ for all $(h', t', s') \in [H] \times [T] \in \mathcal{S}$.
1: Sample $j \in [t]$ with probability $\mathbb{P}(j = i) = \alpha_t^i$.
2: Play policy $\pi_h^j$ at the $h$-th step of the game.
3: Play policy $\widehat{\pi}_{h+1}^j$ for step $h + 1$ onward.

---

**Theorem G.1** (Formal version of Theorem 4). *Suppose Algorithm 12 is run for $T$ rounds. Then the per-state regret can be bounded as follows for some absolute constant $C > 0$:*

$$\mathrm{reg}_h^t \leq \overline{\mathrm{reg}}_h^t := C \left[ \frac{H \log A_{\max}}{\eta t} + \frac{\eta H^3}{t} + (m-1)^2 \eta^3 H^4 \right] \quad \text{for all } (h, t) \in [H] \times [T].$$

*Further, choosing $\eta = (\log A_{\max} \log T/(H^3 T))^{1/4}(m-1)^{-1/2}$, the output policy $\widehat{\pi}^T$ achieves*

$$\begin{aligned}
\mathrm{CCEGap}(\widehat{\pi}^T) \leq &\mathcal{O}\Big( H^{11/4}(\log A_{\max} \log T)^{3/4}\sqrt{m-1} \cdot T^{-3/4} \\
&+ H^{13/4}(\log A_{\max})^{1/4}(\log T)^{5/4}(m-1)^{-1/2} \cdot T^{-5/4} \Big).
\end{aligned}$$

**Proof overview and remarks**    The proof of Theorem G.1 also follows by relating the performance of the output policy by per-state regrets via performance difference (Lemma G.2, similar as Theorem 2), and bounding per-state regrets as $\mathrm{reg}_h^t \leq \overline{\mathrm{reg}}_h^t := \widetilde{\mathcal{O}}(1/(\eta t) + \eta^3 (m-1)^2)$ which gives the theorem. The latter builds upon the fast convergence analysis of OFTRL in multi-player normal-form games [50] as well as additional handling of the changing game rewards, similar as in Theorem 3. Note that the $\widetilde{\mathcal{O}}(T^{-3/4})$ rate here is worse than $\widetilde{\mathcal{O}}(T^{-5/6})$ for the zero-sum setting in Theorem 3. This happens as the fine-grained analysis of OFTRL [9] used there relies critically on the game having two players (for translating between the iterate stabilities and loss stabilities between each other), and becomes infeasible when there are more than 2 players.

We first present some lemmas in Section G.3. The proof of Theorem G.1 is then provided in Section G.4.

### G.3 Useful lemmas

We additionally define the V-values maintained by Algorithm 12 as

$$V_{i,h}^t(s) := \sum_{j=1}^{t} \alpha_t^j \left[ Q_{i,h}^j \pi_h^j \right](s) \tag{47}$$

for all $(i, h, t, s) \in [m] \times [H] \times [T] \times \mathcal{S}$, where $Q_{i,h}^t$ and $\pi_h^t$ are the Q-functions and joint policies maintained within Algorithm 12. Note that by (46), we immediately have

$$
\begin{aligned}
Q_{i,h}^t(s, \mathbf{a}) &= \sum_{j=1}^{t} \alpha_t^j \left[ r_h + \mathbb{P}_h [Q_{i,h+1}^j \pi_{h+1}^j] \right](s, \mathbf{a}) \\
&= \left( r_h + \mathbb{P}_h \left[ \sum_{j=1}^{t} \alpha_t^j Q_{i,h+1}^j \pi_{h+1}^j \right] \right)(s, \mathbf{a}) = \left( r_h + \mathbb{P}_h V_{i,h+1}^t \right)(s, \mathbf{a}).
\end{aligned} \tag{48}
$$

We also define the value functions of $\widehat{\pi}_h^t$ and of its best response for any $(i, h, t, s)$ as (see e.g. [49, Definition C.4 & Eq.(8)]):

$$V_{i,h}^{\widehat{\pi}_h^t}(s) := \mathbb{E}_{\widehat{\pi}_h^t} \left[ \sum_{h'=h}^{H} r_{i,h'} | s_h = s \right],$$

$$V_{i,h}^{\dagger, \widehat{\pi}_{-i,h}^t}(s) := \max_{\pi_{i,h:H}} \mathbb{E}_{\pi_{i,h:H} \times \widehat{\pi}_{-i,h}^t} \left[ \sum_{h'=h}^{H} r_{i,h'} | s_h = s \right].$$

**Lemma G.1** (Equivalence of value functions)**.** *For Algorithm 12, we have for all $i \in [m]$ and all $(h, s, t) \in [H + 1] \times \mathcal{S} \times [T]$ that*

$$V_{i,h}^t(s) = V_{i,h}^{\widehat{\pi}_h^t}(s).$$

*Proof.* We prove this by backward induction over $h \in [H + 1]$. The claim trivially holds for $h = H + 1$. Suppose the claim holds for steps $h + 1$ onward and all $(s, t) \in \mathcal{S} \times [T]$. For step $h$ and any fixed $(s, t) \in \mathcal{S} \times [T]$, note that

$$V_{i,h}^t(s) = \sum_{j=1}^{t} \alpha_t^j \left[ Q_{i,h}^j \pi_h^j \right](s) \overset{(i)}{=} \sum_{j=1}^{t} \alpha_t^j \left[ (r_h + \mathbb{P}_h V_{i,h+1}^j) \pi_h^j \right](s)$$

$$\overset{(ii)}{=} \sum_{j=1}^{t} \alpha_t^j \left[ (r_h + \mathbb{P}_h V_{i,h+1}^{\widehat{\pi}_{h+1}^j}) \pi_h^j \right](s) \overset{(iii)}{=} V_{i,h}^{\widehat{\pi}_h^t}(s).$$

Above, (i) follows by (48); (ii) uses the inductive hypothesis; (iii) uses the definition of the output policy $\widehat{\pi}_h^t$ (cf. Algorithm 13), which samples $j \in [t]$ with probability $\alpha_t^j$, plays $\pi_h^j(\cdot|s)$, and plays $\widehat{\pi}_{h+1}^j$ for the rest of the game. This proves the case for step $h$ and thus the lemma. $\qquad\square$

Define the weighted per-state regrets as

$$\operatorname{reg}_{h,i}^t(s) := \max_{\pi_i^\dagger \in \Delta_{\mathcal{A}_i}} \sum_{j=1}^{t} \alpha_t^j \left\langle Q_h^j(s, \cdot), (\pi_i^\dagger \times \pi_{-i,h}^j)(\cdot|s) - \pi_h^j(\cdot|s) \right\rangle, \tag{49}$$

$$\operatorname{reg}_h^t := \max_{s \in \mathcal{S}} \max_{i \in [m]} \operatorname{reg}_{h,i}^t(s). \tag{50}$$

The following lemma bounds the difference between the values of the certified policy $\pi_h^t$ (Algorithm 13) and its best-response.

**Lemma G.2** (Recursion of best-response values)**.** *For the policy $\widehat{\pi}_h^t$ defined in Algorithm 13, we have for all $(i, h, t) \in [m] \times [H] \times [T]$ that*

$$\max_{s \in \mathcal{S}} \left( V_{i,h}^{\dagger, \widehat{\pi}_{-i,h}^t}(s) - V_{i,h}^{\widehat{\pi}_h^t}(s) \right) \le \operatorname{reg}_h^t + \sum_{j=1}^{t} \alpha_t^j \max_{s \in \mathcal{S}} \left( V_{i,h+1}^{\dagger, \widehat{\pi}_{-i,h+1}^j}(s) - V_{i,h+1}^{\widehat{\pi}_{h+1}^j}(s) \right).$$

*Proof.* Fix $(i, h, t) \in [m] \times [H] \times [T]$. We have for any $s \in \mathcal{S}$ that

$$V_{i,h}^{\dagger, \widehat{\pi}^t_{-i,h}}(s) - V_{i,h}^{\widehat{\pi}^t_h}(s)$$

$$= \max_{\pi_i^\dagger \in \Delta_{\mathcal{A}_i}} \left\langle \pi_i^\dagger, \sum_{j=1}^t \alpha_t^j \left[ (r_h + \mathbb{P}_h V_{i,h+1}^{\dagger, \widehat{\pi}^j_{-i,h+1}}) \pi_{-i,h}^j \right](s, \cdot) \right\rangle - \sum_{j=1}^t \alpha_t^j \left\langle \pi_{i,h}^j, \left[ (r_h + \mathbb{P}_h V_{i,h+1}^{\widehat{\pi}^j_{h+1}}) \pi_{-i,h}^j \right](s, \cdot) \right\rangle$$

$$\overset{(i)}{\leq} \sum_{j=1}^t \alpha_t^j \max_{s' \in \mathcal{S}} \left( V_{i,h+1}^{\dagger, \widehat{\pi}^j_{-i,h+1}}(s') - V_{i,h+1}^{\widehat{\pi}^j_{h+1}}(s') \right) + \max_{\pi_i^\dagger \in \Delta_{\mathcal{A}_i}} \sum_{j=1}^t \alpha_t^j \left\langle \pi_i^\dagger - \pi_{i,h}^j, \left[ (r_h + \mathbb{P}_h V_{i,h+1}^{\widehat{\pi}^j_{h+1}}) \pi_{-i,h}^j \right](s, \cdot) \right\rangle$$

$$\overset{(ii)}{=} \sum_{j=1}^t \alpha_t^j \max_{s' \in \mathcal{S}} \left( V_{i,h+1}^{\dagger, \widehat{\pi}^j_{-i,h+1}}(s') - V_{i,h+1}^{\widehat{\pi}^j_{h+1}}(s') \right) + \underbrace{\max_{\pi_i^\dagger \in \Delta_{\mathcal{A}_i}} \sum_{j=1}^t \alpha_t^j \left\langle \pi_i^\dagger - \pi_{i,h}^j, \left[ (r_h + \mathbb{P}_h V_{i,h+1}^j) \pi_{-i,h}^j \right](s, \cdot) \right\rangle}_{\operatorname{reg}_{i,h}^t(s)}$$

$$\leq \sum_{j=1}^t \alpha_t^j \max_{s' \in \mathcal{S}} \left( V_{i,h+1}^{\dagger, \widehat{\pi}^j_{-i,h+1}}(s') - V_{i,h+1}^{\widehat{\pi}^j_{h+1}}(s') \right) + \operatorname{reg}_h^t.$$

Above, (i) follows by substituting $V_{i,h+1}^{\dagger, \widehat{\pi}^j_{-i,h+1}}$ with $V_{i,h+1}^{\widehat{\pi}^j_{h+1}}$ and paying the additive error; (ii) follows from Lemma G.1. This proves the desired result. $\qquad \square$

**Lemma G.3** (Guarantee of Algorithm 12 via per-state regrets)**.** *Suppose that the per-state regrets* (50) *can be upper-bounded as* $\operatorname{reg}_h^t \leq \overline{\operatorname{reg}}_h^t$ *for all* $(h, t) \in [H] \times [T]$, *where* $\overline{\operatorname{reg}}_h^t$ *is non-increasing in* $t$: $\overline{\operatorname{reg}}_h^t \geq \overline{\operatorname{reg}}_h^{t+1}$ *for all* $t \geq 1$. *Then running Algorithm 12 will guarantee that*

$$\operatorname{CCEGap}(\widehat{\pi}^T) \leq CH \cdot \frac{1}{T} \sum_{t=1}^T \max_{h \in [H]} \overline{\operatorname{reg}}_h^t. \tag{51}$$

*for all* $T \geq 2$, *where* $C > 0$ *is an absolute constant.*

*Proof.* For any $(h, t) \in [H + 1] \times [T]$, define

$$\delta_h^t := \max_{i \in [m]} \max_{s \in \mathcal{S}} \left( V_{i,h+1}^{\dagger, \widehat{\pi}^j_{-i,h+1}}(s) - V_{i,h+1}^{\widehat{\pi}^j_{h+1}}(s) \right).$$

Then Lemma G.2 implies the recursive relationship

$$\delta_h^t \leq \operatorname{reg}_h^t + \sum_{j=1}^t \alpha_t^j \delta_{h+1}^j.$$

(With $\delta_{H+1}^t \equiv 0$ for all $t \in [T]$.) Therefore we can imitate the proof of Lemma C.2 and obtain that, for any $\overline{\operatorname{reg}}_h^t$ such that $\operatorname{reg}_h^t \leq \overline{\operatorname{reg}}_h^t$ and $\overline{\operatorname{reg}}_h^t \geq \overline{\operatorname{reg}}_h^{t+1}$,

$$\delta_h^t \leq H c_\beta^{H-1} \cdot \frac{1}{t} \sum_{j=1}^t \max_{h' \in [H]} \overline{\operatorname{reg}}_{h'}^j,$$

where $c_\beta = 1 + 1/H = \sup_{j \geq 1} \sum_{t=j}^\infty \alpha_t^j$ by Lemma A.2(a).

Further, by definition of the output policy $\widehat{\pi}^T = \widehat{\pi}_1^T$ (cf. Algorithm 13), we have

$$\operatorname{CCEGap}(\widehat{\pi}^T) = V_{i,1}^{\dagger, \widehat{\pi}^T_{-i,1}}(s_1) - V_{i,1}^{\widehat{\pi}_1^T}(s_1) \leq \delta_1^T \leq H c_\beta^{H-1} \cdot \frac{1}{T} \sum_{t=1}^T \max_{h \in [H]} \overline{\operatorname{reg}}_h^t$$

$$\leq CH \cdot \frac{1}{T} \sum_{t=1}^T \max_{h \in [H]} \overline{\operatorname{reg}}_h^t,$$

where $C \leq 3$ as $c_\beta^{H-1} \leq (1 + 1/H)^H \leq e \leq 3$. This is the desired result. $\qquad \square$

### G.4 Proof of Theorem G.1

**Bounding per-state regret**   We first bound $\mathrm{reg}_{i,h}^t(s)$ (cf. definition in (49)), i.e. the per-state regret for the $i$-th player, for any fixed $(i, h, s, t) \in [m] \times [H] \times [S] \times [T]$. This is the main part of this proof.

Throughout this part, we will fix $(i, h, s, t)$, and omit the subscript $(h, s)$ within the policies and Q functions, so that $\pi_{i,h}^t(\cdot|s)$ will be abbreviated as $\pi_i^t$, and $Q_{i,h}^t(s, \cdot)$ will be abbreviated as $Q_i^t$. We will also reload $T \geq 1$ to be any positive integer (instead of the fixed total number of iterations).

We first observe that the update (45) for $\pi_{i,h}^t(\cdot|s)$ is exactly equivalent to the OFTRL algorithm (Algorithm 4) with loss vectors $g_t = w_t(Q_i^t)^\top \pi_{-i}^t$ (understanding $g_0 = 0$ and $Q_i^0 = 0$), prediction vector $M_t = w_t(Q_i^{t-1})^\top \pi_{-i}^{t-1}$, and learning rate $\eta_t = \eta/w_t$. Therefore we can apply the regret bound for OFTRL in Lemma B.3 and obtain for any $T \geq 1$ that

$$
\begin{aligned}
\max_{\pi_i^\dagger \in \Delta_{\mathcal{A}_i}} \sum_{t=1}^T w_t \left\langle \pi_i^t - \pi_i^\dagger, Q_i^t \pi_{-i}^t \right\rangle &= \max_{\pi_i^\dagger \in \Delta_{\mathcal{A}_i}} \sum_{t=1}^T \left\langle \pi_i^t - \pi_i^\dagger, g_t \right\rangle \\
&\leq \frac{\log A_i}{\eta_T} + \sum_{t=1}^T \eta_t \|g_t - M_t\|_\infty^2 - \sum_{t=1}^{T-1} \frac{1}{8\eta_t} \|\pi_i^t - \pi_i^{t+1}\|_1^2 \\
&\leq \frac{\log A_i \cdot w_T}{\eta} + \sum_{t=1}^T \eta w_t \|Q_i^t \pi_{-i}^t - Q_i^{t-1} \pi_{-i}^{t-1}\|_\infty^2 \\
&\leq \frac{\log A_{\max} \cdot w_T}{\eta} + \underbrace{\sum_{t=1}^T 2\eta w_t \left\|Q_i^t - Q_i^{t-1}\right\|_\infty^2}_{\mathrm{I}} + \underbrace{\sum_{t=2}^T 2\eta w_t H^2 \left\|\pi_{-i}^t - \pi_{-i}^{t-1}\right\|_1^2}_{\mathrm{II}}.
\end{aligned}
\tag{52}
$$

Above, the last inequality uses the fact that $\left\|(Q_i^t - Q_i^{t-1})\pi_{-i}^t\right\|_\infty \leq \left\|Q_i^t - Q_i^{t-1}\right\|_\infty$ for $t \geq 1$, and $\left\|Q_i^{t-1}(\pi_{-i}^t - \pi_{-i}^{t-1})\right\|_\infty \leq H \left\|\pi_{-i}^t - \pi_{-i}^{t-1}\right\|_1$ for $t \geq 2$.

For term I, noticing that $\left\|Q_i^t - Q_i^{t-1}\right\|_\infty \leq \alpha_t H$ by (46), we have

$$
\mathrm{I} \leq \sum_{t=1}^T 2\eta w_t \alpha_t^2 H^2 = 2\eta H^2 \sum_{t=1}^T w_t \alpha_t^2.
$$

Bounding term II requires the following lemma on the stability of the iterates. The proof can be found in Section G.5.

**Lemma G.4** (Stability of iterates). *We have for any $i \in [m]$ and any $t \geq 2$ that (recall the subscripts $(h, s)$ are omitted below):*

$$
\left\|\pi_i^t - \pi_i^{t-1}\right\|_1 \leq 4\eta H.
\tag{53}
$$

*Consequently,*

$$
\left\|\pi_{-i}^t - \pi_{-i}^{t-1}\right\|_1 \leq 4(m-1)\eta H.
\tag{54}
$$

Using Lemma G.4, we have

$$
\mathrm{II} \leq \sum_{t=2}^T 2\eta w_t H^2 \cdot 16(m-1)^2 \eta^2 H^2 = 32\eta^3 H^4 (m-1)^2 \sum_{t=2}^T w_t.
$$

Plugging the preceding bounds into (52) and using $\alpha_T^1 \cdot w_t = \alpha_T^t$ yields that

$$
\mathrm{reg}_{i,h}^T(s) = \max_{\pi_i^\dagger \in \Delta_{\mathcal{A}_i}} \sum_{t=1}^T \underbrace{\alpha_T^t}_{\alpha_T^1 \cdot w_t} \left\langle \pi_i^t - \pi_i^\dagger, Q_i^t \pi_{-i}^t \right\rangle = \alpha_T^1 \max_{\pi_i^\dagger \in \Delta_{\mathcal{A}_i}} \sum_{t=1}^T w_t \left\langle \pi_i^t - \pi_i^\dagger, Q_i^t \pi_{-i}^t \right\rangle
$$

$$\leq \frac{\log A_{\max} \cdot \alpha_T^T}{\eta} + 2\eta H^2 \sum_{t=1}^{T} \alpha_T^t \alpha_t^2 + 32\eta^3 H^4 (m-1)^2 \underbrace{\sum_{t=2}^{T} \alpha_T^t}_{\leq 1}$$

$$\overset{(i)}{\leq} \frac{2H \log A_{\max}}{\eta T} + \frac{8\eta H^3}{T} + 32\eta^3 H^4 (m-1)^2$$

$$\leq C \left[ \frac{H \log A_{\max}}{\eta T} + \frac{\eta H^3}{T} + \eta^3 H^4 (m-1)^2 \right] =: \overline{\mathrm{reg}}_h^T.$$

Above, (i) used the fact that $\alpha_T^T = \alpha_T = (H+1)/(H+T) \leq 2H/T$, and $\sum_{t=1}^{T} \alpha_T^t \alpha_t^2 \leq 4H/T$ by Lemma A.3(c), and $C \leq 32$ is an absolute constant. This proves the per-state regret bounds claimed in Theorem G.1.

**Overall policy guarantee** Plugging the above per-state regret bounds into Lemma G.3 yields that, the output policy $\widehat{\pi}^T$ of Algorithm 12 achieves

$$\mathrm{CCEGap}(\widehat{\pi}^T) \leq CH \cdot \frac{1}{T} \sum_{t=1}^{T} \max_{h \in [H]} \overline{\mathrm{reg}}_h^T$$

$$\leq \mathcal{O}\left( \frac{H^2 \log A_{\max} \log T}{\eta T} + \frac{\eta H^4 \log T}{T} + \eta^3 H^5 (m-1)^2 \right).$$

Choosing $\eta = (\log A_{\max} \log T/(H^3 T))^{1/4}(m-1)^{-1/2}$, the above can be upper bounded as

$$\mathcal{O}\left( H^{11/4} (\log A_{\max} \log T)^{3/4} \sqrt{m-1} \cdot T^{-3/4} + H^{13/4} (\log A_{\max})^{1/4} (\log T)^{5/4} (m-1)^{-1/2} \cdot T^{-5/4} \right),$$

which is the desired result. $\qquad\square$

### G.5 Proof of Lemma G.4

We first prove (53). By the OFTRL update (45) and the smoothness of exponential weights (Lemma A.5), we have for any $t \geq 2$ that

$$\left\| \pi_i^t - \pi_i^{t-1} \right\|_1 \leq 2 \left\| G^t - G^{t-1} \right\|_\infty,$$

where $G^t, G^{t-1}$ are the (weighted) total losses in (45):

$$G^t := \frac{\eta}{w_t} \left[ \sum_{j=1}^{t-1} w_j Q_i^j \pi_{-i}^j + w_t Q_i^{t-1} \pi_{-i}^{t-1} \right],$$

$$G^{t-1} := \frac{\eta}{w_{t-1}} \left[ \sum_{j=1}^{t-2} w_j Q_i^j \pi_{-i}^j + w_{t-1} Q_i^{t-2} \pi_{-i}^{t-2} \right].$$

Therefore we have

$$\left\| \pi_i^t - \pi_i^{t-1} \right\|_1 \leq 2 \left\| G^t - G^{t-1} \right\|_\infty$$

$$\leq 2\eta \left\| \left( \frac{1}{w_t} - \frac{1}{w_{t-1}} \right) \sum_{j=1}^{t-1} w_j Q_i^j \pi_{-i}^j \right\|_\infty + 2\eta \left\| 2Q_i^{t-1} \pi_{-i}^{t-1} - Q_i^{t-2} \pi_{-i}^{t-2} \right\|_\infty$$

$$\overset{(i)}{\leq} 2\eta H \cdot \underbrace{\left( \frac{1}{w_{t-1}} - \frac{1}{w_t} \right) \sum_{j=1}^{t-1} w_j}_{\overset{(ii)}{=} H/(H+1) \leq 1} + 2\eta H \leq 4\eta H.$$

Above, (i) uses the fact that $Q_i^j \pi_{-i}^j \in [0, H]$ entry-wise for all $j \geq 1$, and (ii) uses Lemma A.4(b). This proves (53).

The above directly implies (54) by the following bound on the TV distance (or $L_1$ norm) of product distributions [50]:

$$\left\| \pi_{-i}^t - \pi_{-i}^{t-1} \right\|_1 \leq \sum_{j \neq i} \left\| \pi_j^t - \pi_j^{t-1} \right\|_1.$$

This completes the proof. $\qquad\qquad\qquad\qquad\qquad\qquad\qquad\qquad\qquad\qquad\qquad\qquad\square$

# H  Experimental details and additional studies

## H.1  Experimental details for Section 5

**Details about the game**  The simulations in Section 5 is performed on the following two-player-zero-sum Markov game with $H = 2$. The state space at $h = 1$ only consists of a single state $\mathcal{S}_1 = \{s_0\}$. The state space at $h = 2$ consists of four different states $\mathcal{S}_2 = \{s_{11}, s_{12}, s_{21}, s_{22}\}$. The action spaces are the same for every state, namely $\mathcal{A} = \{a_1, a_2\}, \mathcal{B} = \{b_1, b_2\}$, i.e. each player has two actions. The transition from $\mathcal{S}_1 \times \mathcal{A} \times \mathcal{B} \to \mathcal{S}_2$ is deterministic, which takes the following form:

$$s_0 \times a_i \times b_j \to s_{ij}, \quad 1 \leq i, j \leq 2.$$

The instantaneous reward $r_h$ depends only on the action (and not the state), i.e., $r_h : \mathcal{A} \times \mathcal{B} \to [0, 1]$, which takes values as (scaled) identity matrices:

$$r_1(\cdot, \cdot) = \begin{bmatrix} 0.1 & 0 \\ 0 & 0.1 \end{bmatrix}, \quad r_2(\cdot, \cdot) = \begin{bmatrix} 1 & 0 \\ 0 & 1 \end{bmatrix}. \tag{55}$$

Direct calculation yields that the Nash values and policies for this game is given by

$$V_2^\star(s) = 0.5 \text{ for } s \in \{s_{11}, s_{12}, s_{21}, s_{22}\},$$

$$Q_1^\star(s_0, \cdot, \cdot) = \begin{bmatrix} 0.6 & 0.5 \\ 0.5 & 0.6 \end{bmatrix}, \quad \mu_1^\star(\cdot|s_0) = \nu_1^\star(\cdot|s_0) = \begin{bmatrix} 0.5 \\ 0.5 \end{bmatrix}, \quad V_1^\star(s_0) = 0.55. \tag{56}$$

**Initialization**  All algorithms in Figure 1 use the following initialization $(\mu^0, \nu^0)$:

At $h = 1$: $\mu_1^0(a_1|s_0) = 0.3$, $\quad\mu_1^0(a_2|s_0) = 0.7$, $\quad\nu_1^0(b_1|s_0) = 0.7$, $\quad\nu_1^0(b_2|s_0) = 0.3$;

At $h = 2$: $\mu_2^0(a_1|s_{11}) = 0.248, \mu_2^0(a_2|s_{11}) = 0.752, \nu_2^0(b_1|s_{11}) = 0.248, \nu_2^0(b_2|s_{11}) = 0.752;$

$\qquad\qquad\mu_2^0(a_1|s_{12}) = 0.500, \mu_2^0(a_2|s_{12}) = 0.500, \nu_2^0(b_1|s_{12}) = 0.168, \nu_2^0(b_2|s_{12}) = 0.832;$

$\qquad\qquad\mu_2^0(a_1|s_{21}) = 0.500, \mu_2^0(a_2|s_{21}) = 0.500, \nu_2^0(b_1|s_{21}) = 0.168, \nu_2^0(b_2|s_{21}) = 0.832;$

$\qquad\qquad\mu_2^0(a_1|s_{22}) = 0.752, \mu_2^0(a_2|s_{22}) = 0.248, \nu_2^0(b_1|s_{22}) = 0.248, \nu_2^0(b_2|s_{22}) = 0.752.$

Standard FTRL (Algorithm 3) and OFTRL (Algorithm 4) by default uses the uniform distribution as the initialization, as it minimizes the (neg)entropy $\Phi(\cdot)$. To make them initialized at $\mu^0$, we change the regularizers for $\mu_h(\cdot|s)$ to be $\mathrm{KL}(\cdot\|\mu_h^0(\cdot|s))$ for the max-player. (And similarly $\mathrm{KL}(\cdot\|\nu_h^0(\cdot|s))$ for the min-player.) Note that our actual initialization above satisfies the property that all its values are bounded within the interval $[0.15, 0.85]$. In particular, $\mathrm{KL}(\mu'\|\mu_h^0(\cdot|s))$ for any other $\mu' \in \Delta_{\mathcal{A}}$ is bounded by $O(\log(1/0.15)) = O(1)$, and thus all the convergence theorems will still hold with this modified regularizer, with at most a larger (multiplicative) constant than with the $\Phi(\cdot)$ regularizer.

**Remark on runtime**  Running all our experiments takes approximately 6.46 hours CPU running time (Intel(R) Core(TM) i5-8250U CPU).[5]

## H.2  Additional visualizations for the INPG algorithm

Figure 1 shows that the INPG algorithm (with $\eta = 1/\sqrt{T}$) appears to converge much slower than $O(T^{-1/2})$ (which is the rate for FTRL with $\eta = 1/\sqrt{T}$). Here we present some further understandings of this phenomenon by visualizing the optimization trajectories of the INPG algorithm.

---

[5]Code: https://github.com/DianYu420376/NeurIPS2022-Policy-Optimization-MGs

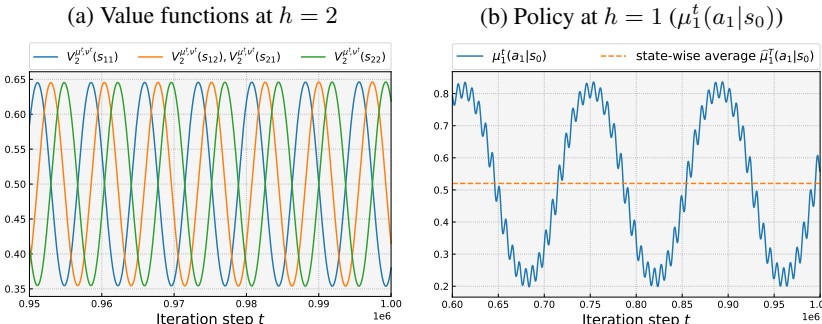

(a) Value functions at $h = 2$      (b) Policy at $h = 1$ ($\mu_1^t(a_1|s_0)$)

Figure 2: Visualizations of the optimization trajectory of the INPG algorithm **along a single run** with $T = 10^6$ and $\eta = 1/\sqrt{T}$. **(a)** Value functions in the second layer $V_2^{\mu^t, \nu^t}(s)$ for all four states $s \in \{s_{11}, s_{12}, s_{21}, s_{22}\}$ over the last $5 \times 10^4$ steps. **(b)** Policy in the first layer, i.e. $\mu_1^t(a_1|s_0)$ over the last $4 \times 10^5$ steps. The horizontal line plots the value of the final averaged policy $\widehat{\mu}_1^T(a_1|s_0)$ (where the averaging is over the **entire run** $t \in [T]$).

Figure 2a shows the evolution of the value functions at $h = 2$ over iteration step $t$, for the last $5 \times 10^4$ steps. For all four states, the policy optimization is equivalent to Hedge on the matrix game with identity reward matrix 55, and thus exhibits an expected cyclic behavior and leads to the sinusoidal-like curves shown in Figure 2a. However, due to the choice of our specific initialization $(\mu^0, \nu^0)$, the four curves behave like the same periodic curves with different *"phases"*.

Figure 2b shows the evolution of the policy at $h = 1$ (specifically, $\mu_1^t(a_1|s_0)$ which is the probability of the max-player taking action $a_1$) over $t$, for the last $4 \times 10^5$ steps. (The result for the min-player is similar.) The curve also behaves periodically, and appears to be a superposition of two waves, one *main waive* with larger magnitude and period, and another *oscillation* with smaller magnitude and period. Qualitatively, the main wave is caused by the intrinsic cyclic behavior of learning with respect to the (fixed) reward at the $h = 1$, while the oscillation is caused by the changing reward that is backed-up from $h = 2$. Further, as the reward in the second layer has much higher magnitude than the first layer in this game, the oscillation has a non-negligible magnitude.

The horizontal line in Figure 2b plots the final output policy $\widehat{\mu}_1^T(a_1|s_0) \approx 0.52$, which we recall is the average of $\mu_1^t(a_1|s_0)$ over the entire run $t \in [T]$ (cf. Section 5). Note that the unique Nash equilibrium satisfies $\mu_1^\star(a_1|s_0) = 0.5$ (56), and the error $\widehat{\mu}_1^T(a_1|s_0) - \mu_1^\star(a_1|s_0) \approx 0.02$. We suspect that this may be an intrinsic bias caused by the aforementioned correlation between the two layers' learning processes (in particular, the different "phases" of the second-layer's learning over the four states), and may also be the cause of the slow convergence for INPG shown in Figure 1.

### H.3     Additional theoretical justifications

**INPG as an instantiation of Algorithm 1**    Here we show why the instantiation of Algorithm 1 with $\beta_t = 1$ and

$$\mu_h^t(a|s) \propto_a \mu_h^{t-1}(a|s) \exp\big(\eta \big[Q_h^{t-1} \nu_h^{t-1}\big](s)\big), \;\; \nu_h^t(b|s) \propto_b \nu_h^{t-1}(b|s) \exp\Big(-\eta \Big[\big(Q_h^{t-1}\big)^\top \mu_h^{t-1}\Big](s)\Big).$$

considered in Section 5 is equivalent to the Independent Natural Policy Gradient (INPG) algorithm. Indeed, choosing $\beta_t = 1$ in Algorithm 1 ensures that $Q_h^t = Q_h^{\mu^t, \nu^t}$ (the true value function of $(\mu^t, \nu^t)$). Therefore, the above update is equivalent to

$$\mu_h^t(a|s) \propto_a \mu_h^{t-1}(a|s) \exp\Big(\eta \Big[Q_h^{\mu^{t-1}, \nu^{t-1}} \nu_h^{t-1}\Big](s)\Big), \;\; \nu_h^t(b|s) \propto_b \nu_h^{t-1}(b|s) \exp\Big(-\eta \Big[\big(Q_h^{\mu^{t-1}, \nu^{t-1}}\big)^\top \mu_h^{t-1}\Big](s)\Big).$$

This is exactly an independent two-player version of the Natural Policy Gradient algorithm (e.g. [1]), where each player plays an NPG algorithm as if they are facing their own Markov Decision Process, with the opponent fixed.

NEGap-Layer-1 **lower bounds** NEGap    Here we show NEGap-Layer-1$(\mu, \nu) \le$ NEGap$(\mu, \nu)$ for any $(\mu, \nu)$. From the definition of $V_h^\star$ we have that

$$V_h^\star(s) = \inf_\nu V_h^{\dagger, \nu}(s) = \sup_\mu V_h^{\mu, \dagger},$$

$$\implies \ V_h^{\mu,\dagger} \le V_h^\star(s) \le V_h^{\dagger,\nu}(s), \ \ \forall \mu, \nu.$$

Thus

$$Q_h^\star(s,a,b) = \left[ r_h + \mathbb{P}_h V_{h+1}^\star \right](s,a,b) \le \left[ r_h + \mathbb{P}_h V_{h+1}^{\dagger,\nu} \right](s,a,b) = Q_h^{\dagger,\nu}(s,a,b),$$

$$Q_h^\star(s,a,b) = \left[ r_h + \mathbb{P}_h V_{h+1}^\star \right](s,a,b) \ge \left[ r_h + \mathbb{P}_h V_{h+1}^{\mu,\dagger} \right](s,a,b) = Q_h^{\mu,\dagger}(s,a,b)$$

$$\implies \ Q_h^{\mu,\dagger}(s,a,b) \le Q_h^\star(s,a,b) \le Q_h^{\dagger,\nu}(s,a,b).$$

Thus for our example

$$\begin{aligned}
\text{NEGap-Layer-1}(\mu,\nu) &= \max_{\mu_1^\dagger} \left[ (\mu_1^\dagger)^\top Q_1^\star \nu_1 \right](s_0) - \min_{\nu_1^\dagger} \left[ \mu_1^\top Q_1^\star \nu_1^\dagger \right](s_0) \\
&\le \max_{\mu_1^\dagger} \left[ (\mu_1^\dagger)^\top Q_1^{\dagger,\nu} \nu_1 \right](s_0) - \min_{\nu_1^\dagger} \left[ \mu_1^\top Q_1^{\mu,\dagger} \nu_1^\dagger \right](s_0) \\
&= V_1^{\dagger,\nu}(s_0) - V_1^{\mu,\dagger}(s_0) = \text{NEGap}(\mu,\nu).
\end{aligned}$$

# I   Additional details

## I.1   Learning setting

In this paper we consider the full-information setting for learning Markov Games formally defined via the following oracle: Given any $h \in [H]$, policies $\mu_{h+1} = \{\mu_{h+1}(\cdot|s) \in \Delta_\mathcal{A}\}_{s \in \mathcal{S}}$, $\nu_{h+1} = \{\nu_{h+1}(\cdot|s) \in \Delta_\mathcal{B}\}_{s \in \mathcal{S}}$, and $V$ function $V_{h+1} \in \mathbb{R}^\mathcal{S}$, we can query the exact value of

$$r_h + \mathbb{P}_h V_{h+1} \in \mathbb{R}^{\mathcal{S} \times \mathcal{A} \times \mathcal{B}}. \tag{57}$$

Note that our algorithm framework (Algorithm 1) and all its subsequent instantiations can execute under the above oracle: Each iteration of Algorithm 1 makes $H$ queries to this oracle, one for each $h \in [H]$ with value function $V_{h+1} = (\mu_{h+1}^t)^\top Q_{h+1}^t \nu_{h+1}^t$ (cf. (4)).

For the purpose of comparing policy optimization type algorithms, we also consider the following weaker oracle: Given any $h \in [H]$, policies $\mu_h = \{\mu_h(\cdot|s) \in \Delta_\mathcal{A}\}_{s \in \mathcal{S}}$, $\nu_h = \{\nu_h(\cdot|s) \in \Delta_\mathcal{B}\}_{s \in \mathcal{S}}$ and $V$ function $V_{h+1} \in \mathbb{R}^\mathcal{S}$, we can query the expected value of its one-step backup under each player's policy:

$$\begin{cases} (r_h + \mathbb{P}_h V_{h+1}) \nu_h \in \mathbb{R}^{\mathcal{S} \times \mathcal{A}}, \\ (r_h + \mathbb{P}_h V_{h+1})^\top \mu_h \in \mathbb{R}^{\mathcal{S} \times \mathcal{B}}. \end{cases} \tag{58}$$

This oracle is also considered in [53, Section 3]. It is clear that one query to (57) can implement one query to (58), and thus (58) is a weaker oracle.

## I.2   Comparison of algorithms

We first remark that all algorithms considered in Section 3.2, 4, & 5 fall into the unified framework of Algorithm 1 and thus are implementable using the full-information oracle (57).

Second, most algorithms we consider: Nash V-Learning (Example 1), GDA (Example 2), OFTRL (Algorithm 4 & Theorem 3), as well as INPG (Section 5) are policy optimization type algorithms which can be implemented using the weaker oracle (58) (see implementation details in Algorithm 5,6,9). In this paper, we are interested in comparing the rates of the above algorithms, where OFTRL achieves a faster $\widetilde{O}(T^{-5/6})$ rate, and Nash V-Learning and GDA achieve the standard $\widetilde{O}(T^{-1/2})$ rate. (The convergence rate of INPG in theory is still open.)

By contrast, Nash Q-Learning (Example 3) and Nash Policy Iteration (Example 4) are not typically considered as policy optimization algorithms, and they cannot be implemented using (58). Our unified framework (Algorithm 1) allows using (57) and thus encapsulates these two algorithms, but due to the usage of a stronger oracle, their convergence rates are not comparable with the aforementioned policy optimization algorithms.