# OpenReview forum: "Policy Optimization for Markov Games: Unified Framework and Faster Convergence"
_NeurIPS.cc/2022/Conference — NeurIPS 2022 Accept_

### Official Review · Reviewer_S7zG · 2022-07-07

**Rating:** 6
**Confidence:** 2
**Soundness:** 2 fair
**Presentation:** 3 good
**Contribution:** 2 fair

**Summary:**

The authors study a unifying meta-algorithm for two-player zero-sum Markov games in the full information setting. The proposed framework aims to generalize some specific algorithms that perform symmetric updates among agents, and the simultaneous learning of policies and their values: relying on upper bound assumptions on the regrets of the employed sub-routines, the authors give a general theoretical result on the convergence of the framework to approximated Nash Equilibria. Beyond providing some examples of how existing algorithms can fit in the proposed framework, the authors use a particular policy update scheme and obtain an overall new algorithm with an improved convergence rate.


**Questions:**

1) On line 267, the authors make consideration of an improved rate for OFTRL which requires two-sided regret bounds.
This consideration seems important, but the authors relegate it to a quick citation.
What is the difference between one-sided and two-sided regret?
The authors say that two-sided regret does not imply one-sided regrets, but the opposite hold?

**Strengths And Weaknesses:**

The paper is well written, and it is mathematically sound. The required background is nicely summarized, the theoretical results are well described, and their requirements clearly stated.
The attempt to put some order in the huge body of literature on RL algorithm for Markov Games is admirable and welcome.

On the other side, the strong theoretical limitations required by this work (full-information setting, tabular representation) raise some concern about his impact and importance (either theoretical or, above all, practical).

---

> ### Author Response · Authors · 2022-08-01
> **Response to Reviewer S7zG**
>
> Thank you for your valuable feedback to our paper! We respond to the questions as follows.
>
> — “*On line 267, the authors make consideration of an improved rate for OFTRL which requires two-sided regret bounds. … The authors say that two-sided regret does not imply one-sided regrets, but the opposite hold*?”
>
> By two-sided regret, we meant the summation of the regrets of the two players, whereas one-sided regret refers to the two individual regrets. Note that the one-sided regret cannot be upper bounded by the two-sided regret (since the regrets may be possibly negative). The opposite holds though, as one-sided regrets directly bound the two-sided regret by summing the bounds.
>
> To avoid possible confusion, we have revised Line 267-270 and renamed the terms as “summation of the regrets” and “individual regret” in our revision.

---

### Official Review · Reviewer_N8fq · 2022-07-09

**Rating:** 6
**Confidence:** 2
**Soundness:** 3 good
**Presentation:** 3 good
**Contribution:** 3 good

**Summary:**

This paper provides a generic framework for two-player zero-sum Markov Games in the full-information setting. The framework has a guarantee on Nash Equilibrium approximation gaps, depending on the per-state weighted regrets provided by different Matrix Game solvers chosen in the framework. Specifically, this paper instantiates the solver to be Nash V-Learning, GDA-Critic, Nash Q-Learning, and Nash-PI, then proves their detailed NE gap guarantees. Further, this paper shows that optimistic FTRL gives a $\tilde{O} (T^{-5/6})$ NE gap, which is the state-of-the-art for symmetric, policy optimization type algorithms in two-player zero-sum MGs. The results can be extended to multi-player general-sum MGs for a Coarse Correlated Equilibrium. Some simulations corroborate the results and incur interesting new problems.

**Questions:**

Basically, the question is about (1) and (2) in Weaknesses.
* What is the difference between policy optimization type/like algorithms?
* Are there any other taxonomic standards which create barrier or gaps in the NE convergence result?

The authors can add more details to Section 3.2, Section 4, and Section 5, like a clear summary of different algorithms and their traits, whether they are policy optimization -type or -like, etc.

**Limitations:**

As the authors stated that it is an open question on generalization to sample-based setting, there are no other fundamental limitations.

**Strengths And Weaknesses:**

Strengths:
1. The framework (Algorithm 1) and its general guarantee (Theorem 2) are useful for future works. Researchers may focus on the per-state regrets and then easily get a NE gap using the framework and guarantee.
2. The $\tilde{O} (T^{-5/6})$ NE gap for OFTRL is currently the best result.
3. The simulation results raise new open problems, which are interesting in my view.


Weaknesses:
1. Some of the concepts need to be provided in the main text. For example, full-information learning, policy optimization type/like algorithms, layer-wise learning, one/two-sided regrets, etc.
2. There lack of comparison between algorithms used in Section 3.2. There are many theoretical results, and the authors did not make it clear whether they are comparable.
3. In practice we cannot apply full information learning. So there is a need for generalization to sample-based learning.

---

> ### Author Response · Authors · 2022-08-01
> **Response to Reviewer N8fq**
>
> Thank you for your valuable feedback on our paper! We respond to the questions as follows.
>
> —“*What is the difference between policy optimization type/like algorithms*?”
>
> By “policy optimization-like”, we were referring to algorithms by (Cen et al. 2021, Zhao et al. 2021; Line 255) where we meant that while their algorithms do use gradient-based updates to learn the policies, they also incorporate some tweaks in other aspects, where either the matrix game at each state needs to be learnt to a sufficient precision before the backup (Cen et al. 2021), or that asymmetric policy update needs to be applied (Zhao et al. 2021).
>
> All other appearances are “policy optimization-type” algorithms, where we meant more vanilla policy optimization algorithms with simultaneous value and policy learning at all layers, and symmetric updates (i.e. without the above tweaks).
> To avoid possible confusion, we have changed the one appearance of "policy optimization-like algorithms” in Line 255 to "algorithms with optimistic gradient-based policy updates" in our revision.
>
> —“*Some of the concepts need to be provided in the main text. For example, full-information learning, policy optimization type/like algorithms, layer-wise learning, one/two-sided regrets, etc*.”
>
> Thanks for the suggestion. To clarify these concepts, we have revised our paper as follows. For “full-information learning”, we have added a formal definition in Appendix H.1. For “layer-wise learning”, we have added some explanations in Line 79-80 and Line 256-257. For “one/two-sided regrets”, we have renamed “one-sided regret” to “individual regret for each player” and “two-sided regret” to “summed regret over the two players” for clarification (Line 268-270).
>
> For “policy optimization type/like algorithms”, please find our explanations in the above question.
>
> —“*Lack of comparison between algorithms used in Section 3.2…clear summary of different algorithms and their traits… Are there any other taxonomic standards which create barrier or gaps in the NE convergence result*?”
>
> We appreciate the suggestion on clarifying the comparisons. We have added a more detailed comparison of the various algorithms in Appendix H.2 in our revision and will make sure to incorporate them into the main text (where appropriate) in the final version.
>
> As a quick summary, all algorithms considered in this paper fall into the unified framework of Algorithm 1 which can be implemented using a full-information feedback oracle (cf. Eq(52)). Further, most algorithms we consider: Nash V-Learning (Example 1), GDA (Example 2), OFTRL (Theorem 3), as well as INPG (Section 5) can also be implemented using a weaker version of the full-information oracle, where the players only observe the expected value of a value backup under the opponent’s current policy (cf. Eq(53)). The convergence rates of these algorithms are comparable. Among these, our proposed OFTRL achieves the fastest $\widetilde{O}(T^{-5/6})$ rate.
>
> By contrast, Nash Q-Learning (Example 3) and Nash Policy Iteration (Example 4) cannot be implemented by the weaker oracle. Our unified framework (Algorithm 1) was designed intentionally to be general enough to encapsulate these two algorithms. Therefore, their convergence rates are not comparable with the aforementioned policy optimization algorithms.

---

> > ### Comment · Reviewer_N8fq · 2022-08-05
> > **post rebuttal response**
> >
> > The authors' response answers my concern. I decide to keep the score unchanged.

---

### Official Review · Reviewer_GbK4 · 2022-07-11

**Rating:** 7
**Confidence:** 4
**Soundness:** 4 excellent
**Presentation:** 4 excellent
**Contribution:** 3 good

**Summary:**

This paper proposes a unified framework (algorithm 1) for Markov games, with symmetric updates and the simultaneous learning of the value functions and the policies. The main theorem (theorem 2) guarantees that the per-state weighted average policies converge to the $\epsilon$-NE (or CCE for the multi-player general-sum setting). The authors show some examples that can be subsumed into the unified framework. Finally, the numerical experiments demonstrate the theoretical results.

**Questions:**

(1) How does state-wise weight average implement? Is the average policy derived as the weighted sum (Line 7 in Algo. 1) on each state separately?

(2) Are the results of INPG the average or other form of policies?

(3) INPG appears to learn faster than FTRL on layer $h=2$ in Fig.1(b) and the NEGap is always lower than the FTRL (FRTL seems to plateau). Can you show the results with more iterations?

(4) Can the framework readily apply in extensive-form games?

**Limitations:**

The current version only establishes the convergence of the average policy, while averaging multiple or even infinite policies in Markov games is non-trivial. The results will be consolidated with the last-iterate property.

**Strengths And Weaknesses:**

**Strengths**:

This general framework can capture many relative algorithms and the authors prove the same convergence rate. The framework consists of two main steps: (1) a policy update by a matrix game solver and (2) a (soft/smooth) value update. However, the update is symmetric and simultaneous, so obtaining the same convergence rate seems to be non-trivial and requires some adaptation. In the numerical experiments, the authors verify the theoretical results first (the results essentially match the order) and show that an eager value update can cause the oscillation issue.


**Weaknesses**:

The state-wise average of policies in Markov games is non-trivial, so the last-iterate convergence is very appealing. Although the current proof only seems to be suitable for the average case, the last-iterate convergence property can still be investigated in experiments. Besides, some variants of OFTRL (e.g., OMWU) can achieve last-iterate convergence with specific update rules, the last-iterate convergence property may be obtained with some adaption.

---

> ### Author Response · Authors · 2022-08-01
> **Response to Reviewer GbK4**
>
> Thank you for the valuable reviews and the positive feedback on our paper! We respond to the questions as follows.
>
> —“*The state-wise average of policies in Markov games is non-trivial,* … *the last-iterate convergence property may be obtained with some adaptation*...*The results will be consolidated with the last-iterate property*.”
>
> Indeed, optimistic algorithms such as OGDA have been shown to also achieve last-iterate convergence in Markov Games with standard $\widetilde{O}(T^{-1/2})$ rate (Wei et al. 2021). However, we emphasize that averaging is required for proving faster rates, such as the $\widetilde{O}(T^{-5/6})$ we show in Theorem 3. This is also the case in normal-form games (e.g. Daskalakis et al. 2021). Also, last iterate convergence for optimistic algorithms on certain problems has been shown to be provably slower than the average (Golowich et al. 2020). Thus, we believe our faster convergence results are complementary to last-iterate convergence results in the literature.
>
> —“*Is the average policy derived as the weighted sum (Line 7 in Algo. 1) on each state separately*?”
>
> Yes, the average is a weighted sum on each state separately.
>
> —“*The current version only establishes the convergence of the average policy, while averaging multiple or even infinite policies in Markov games is non-trivial*… *How does state-wise weight average implement*?”
>
> Our state-wise averaging (Line 7, Algorithm 1) can be directly implemented in a computation and storage efficient manner via moving averages. First, as we average on each state separately, the averaging is over a probability simplex and thus can be obtained by averaging the probability vectors directly (instead of sampling). Second, the weights $\beta_T^t$ correspond exactly to the weights for the moving average associated with learning rates $\{\beta_t\}$ (cf. Eq(6)). Therefore, we can use standard moving average tricks to keep track of the averages, which introduces almost no overhead.
>
> For clarification, we have added a description of this implementation in Appendix C.1 in our revision.
>
> —“*Are the results of INPG the average or other form of policies*?”
>
> Yes, it is the naive (unweighted) average policy (cf. Line 307).
>
>
> —“*INPG appears to learn faster than FTRL on layer h=2 in Fig.1(b) and the NEGap is always lower than the FTRL*?”
>
> Yes, INPG is slightly faster on layer 2, which is expected in theory: On this game, layer 2 is the last layer and is equivalent to a standard matrix game, for which both FTRL and INPG achieve the $O(1/\sqrt{T})$ rate. INPG has a slightly better leading constant due to using standard unweighted FTRL (Line 305), whereas our FTRL uses a weighted version with increasing weights $\alpha_T^t$ (Line 297-299), which causes the regret upper bound to have a slightly worse leading constant (Lemma B.2).
>
> However, we emphasize that while both algorithms are reasonable for layer 2, INPG is much slower on layer 1 (Figure 1(c)) due to its eager value updates compared with FTRL, which causes a slow down of its overall progress reported in Figure 1(a).
>
> —“*Can the framework readily apply in extensive-form games*?”
>
> Our framework can cover Extensive-Form Games with perfect information, as those can be expressed as a Markov Game. However, for the more commonly considered Extensive-Form Games with imperfect information, our results are not readily extendable since there the two players only observe their own infoset rather than the underlying state, and thus the game is not directly expressible as a Markov Game.

---

> > ### Comment · Reviewer_GbK4 · 2022-08-04
> > **post-rebuttal response**
> >
> > Thanks for the detailed clarifications, especially on the average and last-iterate convergence comparisons. I think it wouldl be a much nicer work in the future if this framework can be generalized to the sample-based setting or even in the function approximation (non-tabular representation) case.

---

> > > ### Public Comment · ~Eric_Hodge1 · 2022-11-19
> > > **Agreed**
> > >
> > > Yeah, I agreed with you. And I found 1 minimum deposit casino gaming in Canada via a google search. The website's URL was https://playcasinoscanada.com/casino-bonuses/1-minimum-deposit-casinos-in-canada/. I like to play online real money games through which I can earn money and I don't have to go anywhere now to earn real money.

---

### Author Response · Authors · 2022-08-01
**Revision Updated**

We thank all reviewers for their valuable feedback on our paper! We have updated our paper to incorporate the reviewers’ suggestions. For clarity, all changes are marked in blue.

---

### Meta-Review · Area_Chair_qUhM · 2022-08-26

**Recommendation:** Accept
**Confidence:** Certain

**Metareview:**

The paper makes interesting progress on issues related to multi-agent reinforcement learning providing fast convergence guarantees as well as a unified framework. This is definitely a hot topic of research and it would make for a nice NeurIPS contribution.

**Award:**

No

---

### Decision · Program_Chairs · 2022-09-14

Accept